# Replication-IDentifier links epigenetic and metabolic pathways to the replication stress response

Sophie C. van der Horst ●[1,4], Leonie Kollenstart ●[1,3,4], Amandine Batté ●[1], Sander Keizer[1], Kees Vreeken[1], Praveen Pandey[2], Andrei Chabes ●[2] & Haico van Attikum ●[1] ✉

Perturbation of DNA replication, for instance by hydroxyurea-dependent dNTP exhaustion, often leads to stalling or collapse of replication forks. This triggers a replication stress response that stabilizes these forks, activates cell cycle checkpoints, and induces expression of DNA damage response genes. While several factors are known to act in this response, the full repertoire of proteins involved remains largely elusive. Here, we develop Replication-IDentifier (Repli-ID), which allows for genome-wide identification of regulators of DNA replication in *Saccharomyces cerevisiae*. During Repli-ID, the replicative polymerase epsilon (Pol ε) is tracked at a barcoded origin of replication by chromatin immunoprecipitation (ChIP) coupled to next-generation sequencing of the barcode in thousands of hydroxyurea-treated yeast mutants. Using this approach, 423 genes that promote Pol ε binding at replication forks were uncovered, including *LGE1* and *ROX1*. Mechanistically, we show that Lge1 affects replication initiation and/or fork stability by promoting Bre1-dependent H2B mono-ubiquitylation. Rox1 affects replication fork progression by regulating S-phase entry and checkpoint activation, hinging on cellular ceramide levels via transcriptional repression of *SUR2*. Thus, Repli-ID provides a unique resource for the identification and further characterization of factors and pathways involved in the cellular response to DNA replication perturbation.

Faithful DNA replication is essential for cell survival and genome stability. However, DNA replication faces continual challenges from DNA damage, as well as intrinsic and extrinsic stressors. For example, DNA adducts and interstrand crosslinks can impede replication fork progression, whereas drugs such as hydroxyurea (HU) deplete the cellular dNTP pools, causing replication forks to stall. Failure to restart stalled replication forks can lead to a collapse of these structures and result in the formation of DNA double-strand breaks, which, if left unresolved can lead to genome instability and the onset of cancer[1]. To counteract replication fork stalling and collapse, cells activate a replication stress response that induces cell cycle arrest, upregulates DNA damage response genes, and stabilizes and repairs stalled or collapsed forks. Additionally, homologous recombination (HR) is inhibited to prevent inappropriate recombination at damaged DNA sites, while late origins of replication are suppressed to limit the replication of damaged DNA. These origins can then be reactivated after the stalled or collapsed forks have been repaired[1].

When replication forks stall, the intra-S-phase checkpoint response is activated. Single-stranded DNA (ssDNA) stretches are exposed due to uncoupling of the MCM helicase from the replicative

[1]Department of Human Genetics, Leiden University Medical Center, Leiden, The Netherlands. [2]Department of Medical Biochemistry and Biophysics, Umeå University, Umeå, Sweden. [3]Present address: Novo Nordisk Foundation Center for Protein Research (CPR), University of Copenhagen, Copenhagen, Denmark. [4]These authors contributed equally: Sophie C. van der Horst, Leonie Kollenstart. ✉e-mail: h.van.attikum@lumc.nl

DNA polymerases (Pol α, Pol δ, and Pol ε). These ssDNA regions become coated with the ssDNA-binding protein complex RPA (Rfa1-Rfa2-Rfa3)[2]. The binding of RPA to ssDNA acts as a signal for the recruitment and activation of the Mec1 kinase (the yeast equivalent of human ATR) through its co-factor Ddc2 (hATRIP). Full activation of Mec1 requires additional factors, including the Ddc1 subunit of the PCNA-like clamp complex 9-1-1, Dbp11 (hTopBP1), and Dna2[3,4]. Once activated, Mec1 phosphorylates Rad53 (analog of human CHK1/2), a process aided by the phosphorylation of Mrc1 (hCLASPIN), which moves with the replication fork[5]. Moreover, other components of the replisome, such as the Sgs1 helicase (the homolog of human WRN helicase), also contribute to Rad53 activation[6], enhancing amplification of the intra-S-phase checkpoint response[7]. Together, these events are essential for stabilizing replication forks and allowing the resumption of DNA synthesis after fork arrest. Further research showed that Mec1 functions not only in the intra-S-phase checkpoint, but also acts as a central orchestrator in pathways dedicated to mitigating replication fork stalling, encompassing chromatin remodeling, transcription and proteotoxic stress control, for instance[8]. However, the intricacies of these pathways remain largely elusive, primarily due to a lack of comprehensive knowledge regarding the functions and interplay between the factors involved.

High-throughput functional genomics and phenotypic screens have been employed to pinpoint factors implicated in preventing and resolving replication fork stalling and collapse[9–11]. However, most of these approaches were based on indirectly measuring replication stress responses through cell survival, while methods to directly measure replication fork progression or stability under stress conditions, particularly in a genome-wide manner, were lacking. Therefore, we developed Replication-IDentifier (Repli-ID), which quantitatively measures replication fork integrity and progression in thousands of yeast mutants in parallel. Using Repli-ID, we identified 423 genes whose loss decreases the accumulation of the replicative polymerase Pol ε near an origin of replication under conditions of HU-induced replication stress. Functional assays unveiled the regulatory roles of *LGE1* and *ROX1*, two genes identified by Repli-ID, in distinct epigenetic and metabolic pathways linked to the replication stress response. Thus, Repli-ID provides a robust strategy to directly investigate replisome stability and progression in replication stress conditions in a high-throughput manner, providing insights into the factors and pathways involved.

## Results

### Towards Repli-ID: a method to identify regulators of replication fork stability/progression in yeast

To identify genes that affect the cellular response to DNA replication perturbation, we aimed at developing a tool that measures replication fork progression and stability in the presence of HU in a genome-wide collection of yeast deletion mutants. This method, which we termed Replication-IDentifier (Repli-ID) builds upon the chromatin immuno-precipitation (ChIP)-based barcode screening method Epi-ID[12], and classical ChIP-qPCR experiments, which track the binding and progression of Pol2, the catalytic subunit of the leading strand DNA polymerase ε (Pol ε), near replication origins[6,13]. To refine these approaches, we utilized the established barcoder library, consisting of yeast strains in which barcodes were integrated at the HO locus, 53 bp downstream of the replication origin ARS404 (Fig. 1a), allowing us to measure Pol ε binding near this origin. The barcodes were introduced into the yeast knockout (KO) strain collection using Synthetic Genetic Array (SGA) technology, so each mutant carries a unique barcode (Fig. 1a). ChIP will be performed on the pool of yeast mutants to monitor Pol ε binding at the barcode next to ARS404 (Fig. 1b, c), following next-generation sequencing of the barcodes (Fig. 1d). The amount of Pol ε in a given yeast mutant is reflected by the abundance of its corresponding barcode relative to all other barcodes (Fig. 1d).

This serves as an indicator of replisome progression or stability in that mutant[14], although Pol ε levels may also be influenced by replication forks moving away from sites of DNA damage[15], or by impaired S-phase entry[16].

To study the firing of ARS404, we first investigated the binding of Pol ε, functionally tagged with 9xMyc[6], to ARS404. The well-studied early firing origin ARS607 served as a control (Supplementary Fig. 1a)[6,17]. Cells were synchronized in G1 by α-factor treatment and released into S-phase in medium containing 200 mM HU (Supplementary Fig. 1a). Binding of Pol ε at ARS607 was observed 40 min after release (Supplementary Fig. 1b), agreeing with earlier reports[6,17]. In contrast, binding of Pol ε to ARS404 was not observed, suggesting that ARS404 is a late firing origin, as previously shown by others[18].

In the presence of HU, late origin firing is repressed by Rad53-dependent phosphorylation of Sld3 and Dbf4[19]. To circumvent the lack of firing of ARS404, we overexpressed mutant forms of Sld3 and Dbf4 (*sld3-38A dbf4-4A*), which cannot be phosphorylated by Rad53, in the strain expressing Myc-tagged Pol ε (Fig. 1b). These mutants are fully competent in essential functions in replication initiation and checkpoint activation[19]. Accordingly, the *sld3-38A dbf4-4A* mutant showed little to no HU sensitivity (Supplementary Fig. 1c). However, to limit possible DNA damage from previous cell cycles caused by the overexpression of *sld3-38A dbf4-4A*, we overexpressed these mutants from a galactose-inducible promotor for only 30 min. Under these conditions, efficient binding of Pol ε was observed after 40 and 80 min in S-phase at both ARS607 and ARS404 (Supplementary Fig. 1d).

Having established ARS404 firing in S-phase, yeast strains and conditions for the Repli-ID screen were established. To this end, the *Pol ε-9xMyc sld3-38A dbf4-4A* strain was crossed with the barcoder KO library using SGA technology (Fig. 1b). The resulting Repli-ID library contained around ~4500 barcoded mutants and ~1100 barcoded wild-types as control, all carrying the *Pol ε-9xMyc sld3-38A dbf4-4A* alleles (in the S288C strain background). To validate the Repli-ID library and our experimental approach, we obtained the barcoded *sgs1Δ* mutant from the library and assessed Pol ε binding after G1 arrest and release in HU. In the absence of Sgs1, Pol ε binding was reduced in S-phase in the presence of HU (Supplementary Fig. 1e), consistent with a previous study showing that Sgs1 promotes replisome stability[6].

To study the effect of pooled gene deletions on Pol ε stability, we examined Pol ε binding by ChIP-qPCR in a pool of ~1100 barcoded mutants and in a pool of ~1000 barcoded wild-type strains. Pol ε levels at ARS404 were equal in both pools (Supplementary Fig. 1f), suggesting no apparent effect of individual mutants in the pool on the overall recruitment of Pol ε to the barcodes. The strongest binding of Pol ε at ARS404 was observed 40 min after release in S-phase (Supplementary Fig. 1f). After 80 min, Pol ε declined at ARS404, suggesting progression of the replication fork away from the origin into the neighboring chromatin. As decreased Pol ε at ARS404 at both timepoints is more likely to reflect an unstable fork, including both timepoints enables us to distinguish slower traveling or delayed forks (e.g. from delayed origin firing) from unstable forks. Therefore, these timepoints were chosen for sample collection during Repli-ID screens.

### Repli-ID identifies known regulators of replication fork stability/progression

Having established the Repli-ID conditions, we performed two independent Repli-ID screens. The data of the two replicates strongly correlated (R = 0.90 for t = 40 min and R = 0.91 for t = 80 min; Supplementary Fig. 2a, b), indicating a high-quality dataset. Ultimately, Repli-ID outcomes were obtained for 2905 mutants (Supplementary Data 1). Several common and strong outliers could be detected over the background at 40 and 80 min after release in S-phase (Fig. 2a). In total, we identified 423 mutants in which Pol ε levels were decreased (log2(fold change) < −1.25), likely due to impaired replication fork stability/progression and/or reduced S-phase entry. Furthermore, we

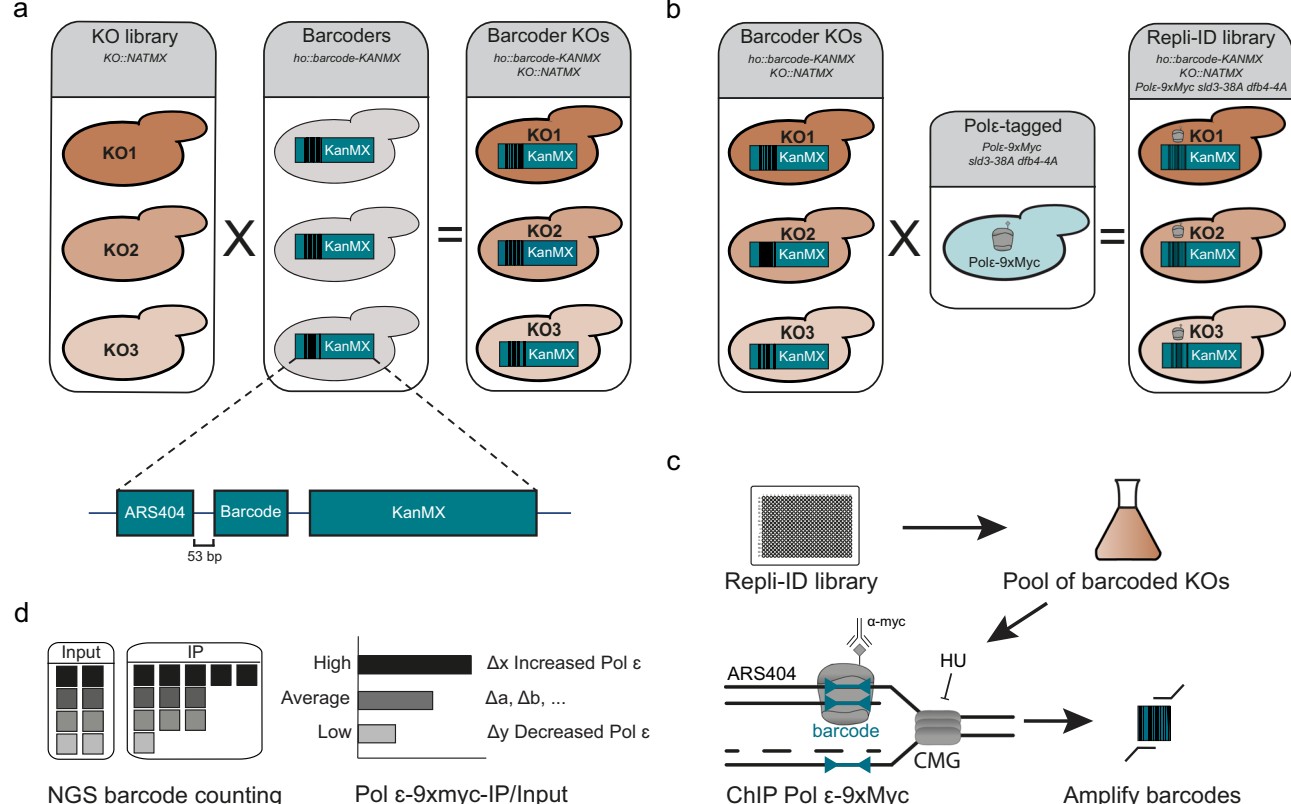

**Fig. 1 | Repli-ID, an approach to identify regulators of replication fork progression/stability. a** Construction of the Repli-ID library. Knockout (KO) libraries of yeast mutants were crossed to a barcoder library of yeast strains using SGA technology. Each strain in the barcoder library contains a KanMX selection gene flanked by a unique 20 bp barcode integrated at the HO locus, which is located adjacent to an origin of replication (ARS404). The cross between the KO libraries and the barcoder library produced yeast strains in which each knockout contains a unique barcode. **b** The strain containing Myc-tagged Pol ε (Pol2-9xMyc) and the galactose-inducible *sld3-38A dbf4-4A* construct[19], which was integrated at the BAR1 locus, was crossed with the barcoder KO library to generate the Repli-ID library. **c** Outline of the Repli-ID procedure. Strains from the Repli-ID library were pooled and grown in liquid medium. Pools of cells were arrested in G1 using alpha-factor treatment, after which galactose was added to induce the expression of the *sld3-38A dbf4-4A* mutants, thereby activating all origins during the release into S-phase. The cells were released in S-phase in the presence of 200 mM HU for 40 or 80 min. Next, cells were subjected to chromatin immunoprecipitation (ChIP) of Pol ε-9xMyc using anti-Myc antibody. The barcodes were amplified from ChIP and input DNA. **d** Next-generation sequencing of barcodes. Barcodes were counted in immunoprecipitated (IP) and input samples. Abundance of Pol ε was measured by adjusting barcode levels in IP to those in the input. This approach enabled the measurement of DNA polymerase levels at the barcode in each individual mutant, allowing for a direct comparison with wild-type levels.

found 128 mutants in which Pol ε levels were increased (log2(fold change) > 1.25) at both timepoints, possibly due to persistent replication fork stalling. Moreover, 154 mutants showed reduced Pol ε levels at 40 min (log2(fold change) < −1.25), but not at 80 min, possibly reflecting impaired fork progression, defective origin firing or cell cycle problems. Lastly, 202 mutants showed normal Pol ε levels at 40 min, but reduced Pol ε levels at 80 min (log2(fold change) < −1.25), likely due to enhanced fork progression.

To study whether the 423 hits with decreased Pol ε levels are a result of a defective entry into S-phase, we overlapped these hits with a previously published list of mutants with increased levels of G1 cells (Supplementary Fig. 2c)[15]. This analysis revealed that 5.0% of our hits (21 out of 423 mutants) were classified as having a high percentage of G1 cells, compared to a genome-wide rate of 3.5% (152 out of 4342 mutants), suggesting that the hits from the Repli-ID are only slightly enriched for factors that are unable to properly enter S-phase and start DNA replication. Furthermore, the Repli-ID hits were enriched in the GO Slim term Chromatin organization (Fig. 2b)[20–22], which is in agreement with several reports showing that alterations in chromatin organization impact DNA replication[23,24], as well as in the GO Slim terms Vacuolar transport and Endosomal transport. The latter may be explained by defects in vacuole biogenesis, resulting in cell cycle misregulation[25]. Finally, among the strongest hits were mutants of Pol32 (a subunit of replicative Pol δ)[26], the replication factor Mgs1[27] and

the DNA helicase Pif1[28], all of which are known to play a role in DNA replication (Fig. 2a). Thus, Repli-ID interrogated Pol ε abundance near an origin of replication in nearly 4500 strains simultaneously, identifying several known regulators of replication fork stability/progression.

## Repli-ID identifies new factors that affect replication fork stability/progression

In addition to known replication stress factors, our Repli-ID screens identified several mutants of factors whose role in the response to replication stress remains unclear (Fig. 2a). These included the *tma7Δ*, *yta7Δ*, *tpa1Δ*, *cse2Δ*, *rec8Δ*, *nsr1Δ*, *fpr4Δ*, *efg1Δ*, *pml1Δ*, *apt1Δ*, *rox1Δ*, and *lge1Δ* mutants, all of which exhibited reduced Pol ε levels at both 40 and 80 min (log2(fold change) < −1.25). To validate these Repli-ID results, we examined Pol ε binding in these randomly selected mutants of the Repli-ID library, which express *sld3-38A dbf4-4A*, near ARS404 and ARS607 using ChIP-qPCR. Although in the *tma7Δ* and *yta7Δ* mutants the Pol ε levels were not much reduced at ARS404 and ARS607 at 40 min, we found that in the *tpa1Δ*, *cse2Δ*, *rec8Δ*, *nsr1Δ*, *fpr4Δ*, *efg1Δ*, *pml1Δ*, *apt1Δ*, *rox1Δ*, and *lge1Δ* mutants, the Pol ε levels were decreased to the same extent as observed in *pol32Δ*, which served as a positive control (Fig. 2c), validating results from the Repli-ID. To determine whether replication fork progression/stability could be affected independently of replication stress in these mutants, we

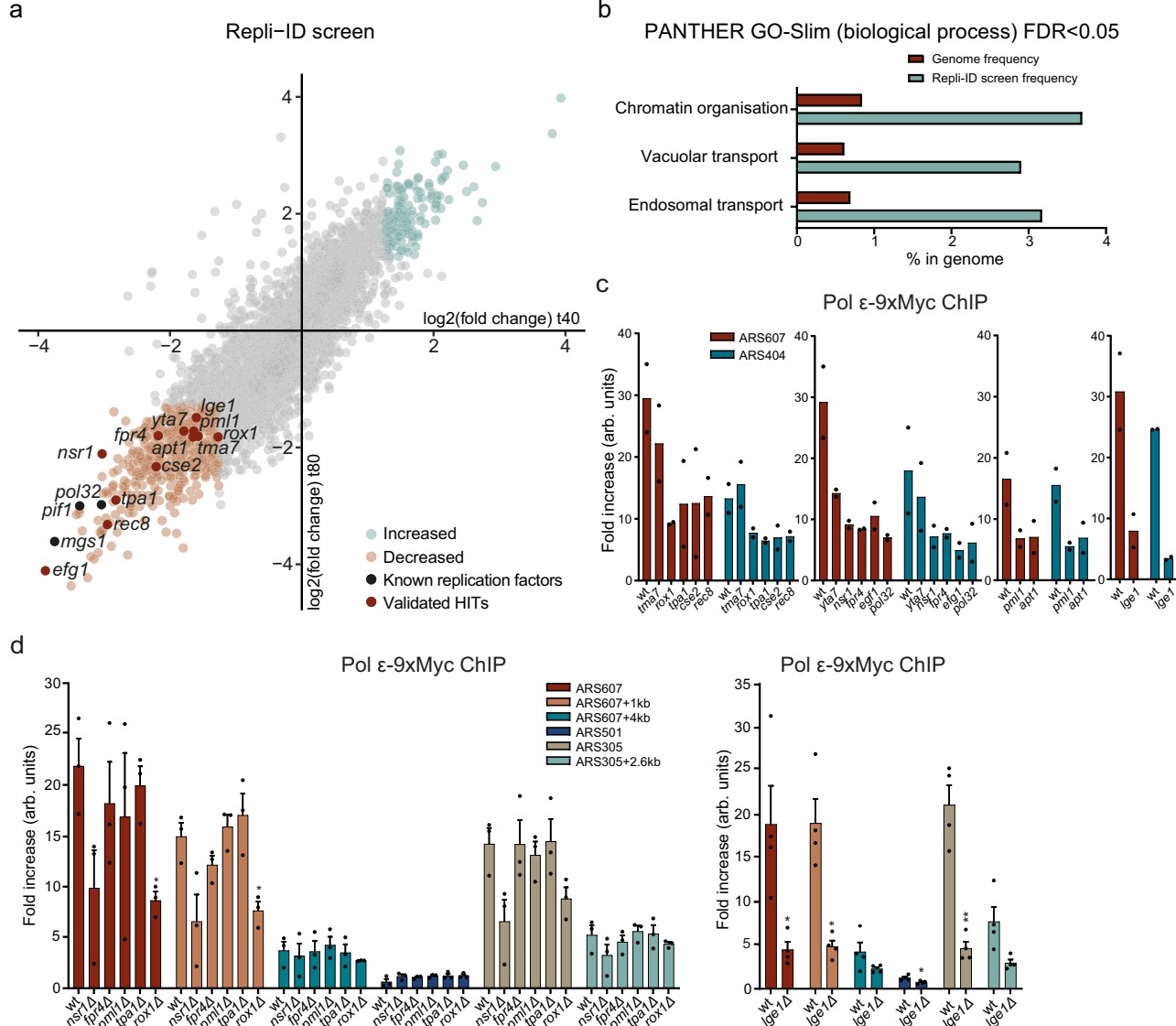

**Fig. 2 | Repli-ID identifies known and new regulators of DNA replication fork progression/stability. a** Outcome of Repli-ID for Pol ε-9xMyc in 2905 yeast mutants (Supplementary Data 1). Scatter plot shows IP/input ratios at 40 and 80 min after G1 arrest and release in 200 mM HU. The mean of $n = 2$ independent Repli-ID screens is shown. Each dot represents a single mutant strain. Increased (light blue) and decreased factors (light orange), known replication factors (black) and validated factors (red) are highlighted. **b** GO Slim biological process analysis (false discovery rate (FDR) < 0.05) of the top 423 mutants showing depletion of Pol ε (< −1.25 log2(fold change) for $t = 40$ min and $t = 80$ min). Genome frequency is depicted in red and frequency within Repli-ID in blue. **c** ChIP-qPCR analysis of Pol ε-9xMyc at ARS607 and ARS404 in a selection of yeast mutants from the Repli-ID screens at 40 min after G1 arrest and release in S-phase in 200 mM HU. Data

represent the mean relative fold enrichment of Myc signal over beads only signal of $n = 2$ independent experiments. Values were normalized to a non-replicated region (ARS607 + 14 kb). **d** ChIP-qPCR analysis of Pol ε-9xMyc in selected hits from the Repli-ID screens, for which the corresponding mutants were generated de novo in the W303 strain background, at the indicated origins 20 min after G1 arrest and release in S-phase in 200 mM HU. Data represent the mean relative fold enrichment + SEM of Myc signal over beads only signal in $n = 3$ or $n = 4$ independent experiments. Values were normalized to a non-replicated region (ARS607 + 14 kb). Statistical significance compared to the wild type was calculated using the two-tailed unpaired Student's $t$ test, assuming unequal variances, *$p < 0.05$, **$p < 0.01$. Source data are provided as a Source Data file.

performed ChIP-qPCR to examine Pol ε binding at ARS404 in the absence of HU in *rec8Δ*, *fpr4Δ*, and *apt1Δ* mutants, all of which did not show increased levels of G1 cells[16]. While in *rec8Δ* cells Pol ε levels were comparable to that in wild-type cells, they were reduced in *fpr4Δ* and *apt1Δ* cells (Supplementary Fig. 2d). Thus, Repli-ID identified mutants that affect replication forks both independently of replication stress (e.g. *fpr4Δ* and *apt1Δ*) and specifically under replication stress conditions (e.g. *rec8Δ*).

Given that yeast knockout (YKO) libraries can contain incorrect or contaminated mutants[29], and that overexpression of *sld3-38A dbf4-4A* might influence the DNA damage response[30], we next examined whether replication fork stability is also affected in *nsr1Δ*, *fpr4Δ*, *pml1Δ*,

*tpa1Δ*, *rox1Δ*, and *lge1Δ* mutants, all of which were hits in the Repli-ID, when generated de novo in the W303 background. Loss of Fpr4, Pml1, and Tpa1 did not show any decrease in Pol ε accumulation at both early firing origins ARS607 and ARS305 after 20 min in S-phase in the presence of HU (Fig. 2d), a timepoint at which the strongest enrichment was observed in W303 cells (Fig. 3b). This may suggest that decreased Pol ε levels in the Repli-ID screens were not due to loss of these factors, or that their effect on Pol ε is background-dependent. In contrast, loss of Nsr1, Rox1, and Lge1 resulted in decreased levels of Pol ε. Furthermore, late firing origin ARS501 was not activated in any of these mutants, as evidenced by the lack of Pol ε binding, validating inhibition of late firing origins (Fig. 2d). Since the Repli-ID screens were

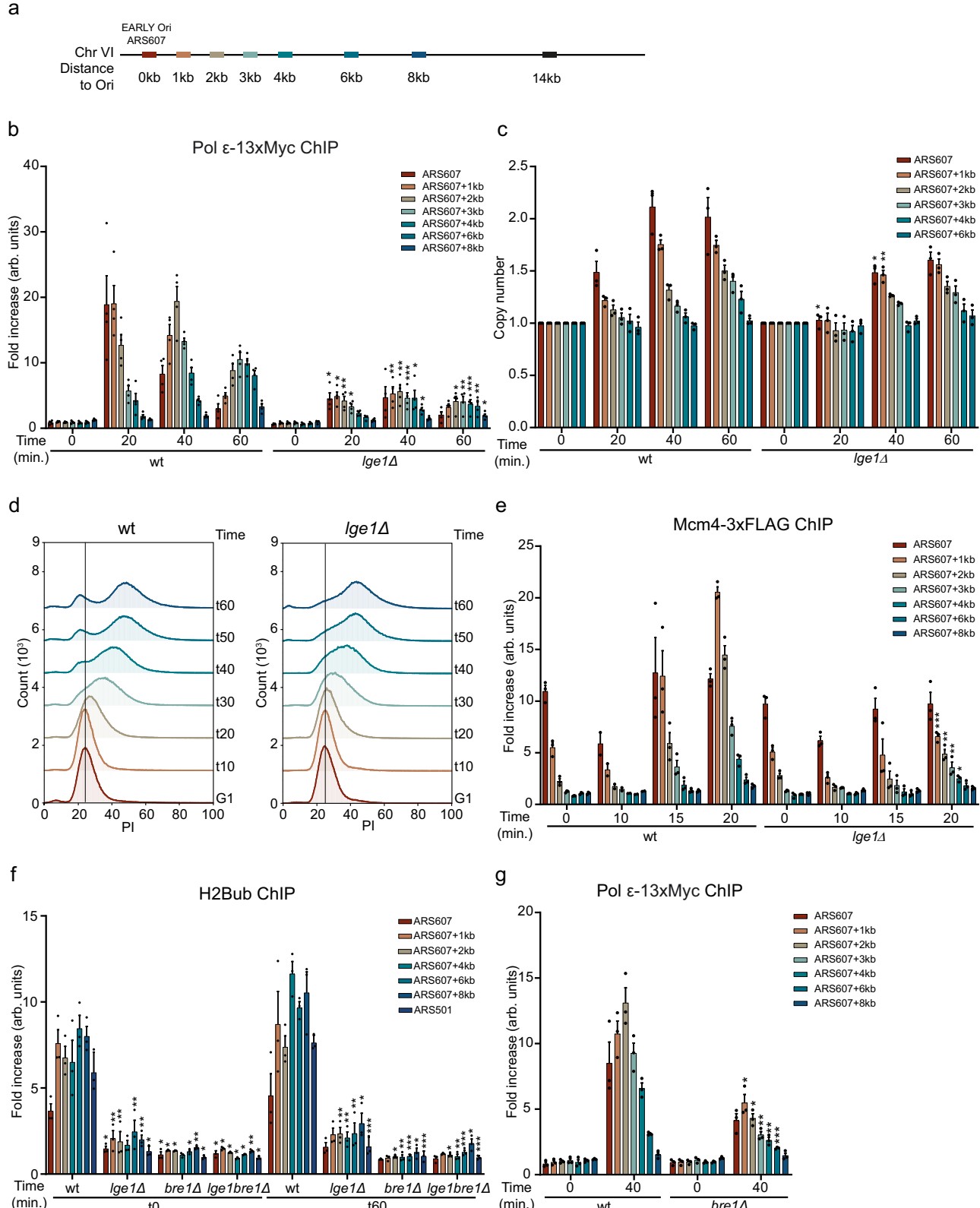

performed in presence of HU, the HU sensitivity of these deletion mutants was examined by a spot dilution test (Supplementary Fig. 2e). While *rox1Δ* and *lge1Δ* showed HU sensitivity, *nsr1Δ* was not HU-sensitive, suggesting that in this mutant the decreased Pol ε accumulation does not confer HU sensitivity. We therefore decided to further investigate the role of Lge1 and Rox1 in the response to replication stress. In conclusion, these results demonstrate that Repli-ID provides

a robust and valuable resource for modulators of replication fork stability/progression.

## Lge1 affects DNA replication under stress conditions by promoting H2B mono-ubiquitylation

Large 1 (Lge1) is a co-factor of Bre1, which is an E3 ligase that acts in concert with the E2 conjugase Rad6 to mono-ubiquitylate histone H2B

**Fig. 3 | Lge1 controls fork progression/stability via H2B ubiquitylation.**
**a** Overview of qPCR amplicons around ARS607 used in ChIP-qPCR experiments.
**b** ChIP-qPCR analysis of Pol ε-13xMyc at ARS607 in the indicated strains at different timepoints after release from G1 in S-phase in 200 mM HU. Data represent the mean relative fold enrichment + SEM of Myc signal over beads only signal of $n = 4$ independent experiments. Values were normalized to a non-replicated region (ARS607 + 14 kb). **c** Dynamics of ARS607 duplication assayed by DNA copy number analysis using qPCR in the indicated strains at different timepoints after release from G1 in S-phase in 200 mM HU. Data represent the mean DNA quantity + SEM of $n = 3$ independent experiments. Values were normalized to a non-replicated region (ARS607 + 14 kb) and further normalized to the ratios of the samples in G1, which were set to 1. **d** Cell cycle profiling of the indicated strains. Cells were grown, arrested in G1, and released in 10 mM HU. DNA content was determined by propidium iodide (PI) staining and flow cytometry. A repeat of this experiment is shown in Supplementary Fig. 3c. **e** As in (**b**), except for Mcm4-3xFLAG. Data represent the mean relative fold enrichment + SEM of FLAG signal over beads only signal of $n = 3$ independent experiments. Values were normalized to a non-replicated region (ARS607 + 14 kb). **f** As in (**b**), except for H2B ubiquitylation at Lysine 123 (H2Bub). Data represent the mean relative fold enrichment + SEM of H2Bub signal over input signal of $n = 3$ independent experiments. Values were normalized to a telomere region (TELVI-R), 0.5 kb away from the telomere on the right arm of chromosome 6 at which no binding of Bre1 is expected due to a lack of histones. **g** As in (**b**), expect for *bre1Δ*. Statistical significance compared to wt was calculated using the two-tailed unpaired Student's *t* test, assuming unequal variances, *$p < 0.05$, **$p < 0.01$, ***$p < 0.001$. Source data are provided as a Source Data file.

at Lysine 123 (H2Bub)[31]. Indeed, we observed impaired H2B ubiquitylation in the *lge1Δ* mutant, which could be rescued by ectopic expression of *LGE1* (Supplementary Fig. 3a). H2Bub is a histone modification that is required for transcription elongation and the recovery of cells from replication stress[32–34]. Moreover, Lge1 and H2Bub have been reported to play roles in establishing sister chromatid cohesion during replication and premeiotic DNA synthesis[35]. However, while the role of Bre1 in HU-induced replication stress via H2Bub is well established[32], the involvement of Lge1 in the replication stress response remained unexplored. To investigate this, we first validated the effect of Lge1 on DNA replication as observed in Repli-ID (Fig. 2a, c, d) through a more extensive analysis of Pol ε progression near ARS607 (Fig. 3a). In wild-type cells, Pol ε accumulated at the origin at early timepoints (20 min) and gradually decreased, while its presence increased further away from the origin at later timepoints (40 and 60 min). Loss of Lge1 led to a strong decrease in Pol ε enrichment after 20, 40, and 60 min in HU (Fig. 3b). To assess whether the decrease in Pol ε binding results in reduced DNA synthesis, we performed DNA copy number analysis. This demonstrated that DNA synthesis was already detectable in wild-type cells after 20 min in HU, while it only became detectable in *lge1Δ* after 40 min in HU. Moreover, after 40 min in HU, DNA levels were reduced in *lge1Δ* cells when compared to that in wild-type (Fig. 3c). Collectively, these findings point to defects in replication initiation and/or fork stability in the absence of Lge1. Corroborating these findings, we found that the budding index of cells lacking Lge1 was comparable to that of wild-type cells (Supplementary Fig. 3b), suggesting the initiation of replication and not the progression through START affects DNA synthesis. Moreover, flow cytometry-based cell cycle analysis showed impaired entry and progression into S-phase in the presence of HU (Fig. 3d and Supplementary Fig. 3c).

In the absence of HU, we also observed a significant decrease in Pol ε enrichment after 25 min in S-phase (Supplementary Fig. 3d). The loading of the MCM helicase was not affected as demonstrated by Mcm4-3xFLAG ChIP-qPCR, indicating proper licensing, consistent with previous findings in *htb-K123R* mutant cells in which H2B cannot be ubiquitylated at Lysine 123[33]. In contrast, the progression of Mcm4 was impaired (Fig. 3e), correlating with the decrease in Pol ε binding (Fig. 3b). Together, these results suggest that Lge1 regulates DNA replication both in unperturbed and stress conditions, likely by affecting replication initiation and/or fork stability.

It remains unclear whether Lge1 impacts DNA replication indirectly, such as by regulating dNTP levels or through HU-induced production of reactive oxygen species (ROS). To study this, we first examined dNTP levels in *lge1Δ*. We found the dNTP levels to be increased in the absence of Lge1 (Supplementary Fig. 3e), similar to what has been previously described for H2Bub-deficient (*htb-K123R*) cells[32]. Reduced rather than increased dNTP levels have been suggested to impair replication fork stability/progression[36], suggesting that altered dNTP levels are unlikely the cause of reduced Pol ε levels in *lge1Δ* (Figs. 2a, 3b and Supplementary Fig. 3e). Second, we tested the HU sensitivity of *lge1Δ* in the presence of a ROS scavenger *N*-acetyl-L-

cysteine (NAC) to demonstrate that the decrease in Pol ε binding observed in this mutant was not due to ROS production (Supplementary Fig. 3f).

Given that Lge1 promotes H2B ubiquitylation (Supplementary Fig. 3a)[31], we next investigated whether it facilitates this process near origins of replication, thereby regulating DNA replication. In wild-type cells, high levels of H2Bub were detected in G1-phase (*t0*), which remained largely unchanged at 20, 40 and 60 min after release in HU (Supplementary Fig. 3g). Loss of Lge1 resulted in strongly decreased levels of H2Bub near ARS607, which were similar in G1-phase (*t0*) and at 20, 40 and 60 min after release in HU (Supplementary Fig. 3g). Loss of only Bre1 or both Lge1 and Bre1 showed a similar decrease of H2Bub levels near ARS607 in G1 and at 60 min after release in HU (Fig. 3f), suggesting that Lge1 and Bre1 act epistatically during H2B ubiquitylation. Furthermore, after 40 min in HU, Pol ε binding in the absence of Bre1 was comparable to the levels observed without Lge1 (Fig. 3b, g). This aligns with previous findings at ARS305 in *htb-K123R* mutant cells[33], confirming that the lack of H2Bub associates with loss of Pol ε under HU conditions. Taken together, we conclude that Lge1 promotes Bre1-dependent H2B ubiquitylation in both G1 and S-phase near origins of replication, thereby likely affecting replication initiation and/or fork stability.

## Bre1 recruitment to stalled replication forks is partially Lge1-dependent
Bre1 is recruited to promoters and open reading frames in a manner independent of Lge1[31]. Whether Lge1 is also dispensable for Bre1 recruitment to stalled replication forks remains unclear. To this end, we monitored the binding of N-terminally FLAG-tagged Bre1 near ARS607 in presence and absence of Lge1 (Fig. 4a). In wild-type cells, Bre1 was enriched near ARS607 and the constitutively transcribed *PMA1* gene (positive control) both in the absence and presence of HU, while it was not enriched at an intergenic region on chromosome V (negative control) (Supplementary Fig. 4a)[37]. Additionally, Bre1 was not detected at the late firing origin ARS501 at 20, 40, and 60 min after release into S-phase in the presence of HU. Surprisingly, in the absence of Lge1, recruitment of Bre1 near ARS607 was only partially lost (Fig. 4a), while a complete loss of H2Bub was observed under these conditions (Fig. 3f). In the absence of HU, Bre1 was not recruited near ARS607 (Supplementary Fig. 4b), showing Bre1 recruitment is a replication stress-dependent process.

During transcription, Lge1 prevents recruitment of the deubiquitylating enzyme (DUB) Ubp8[37,38]. Based on these observations, we infer that in the absence of Lge1, Ubp8 is responsible for removing the H2Bub marks formed by residual Bre1 protein present at stalled replication forks. We tested this hypothesis by performing H2Bub ChIP-qPCR experiments in wild type, *lge1Δ*, *ubp8Δ*, and *lge1Δubp8Δ* (Supplementary Fig. 4c). We did not observe an increase in the H2Bub levels in *lge1Δubp8Δ* compared to *lge1Δ*, indicating that during replication Lge1 is not responsible for preventing the recruitment of Ubp8. Since we demonstrated that Bre1 can still be partially recruited in the

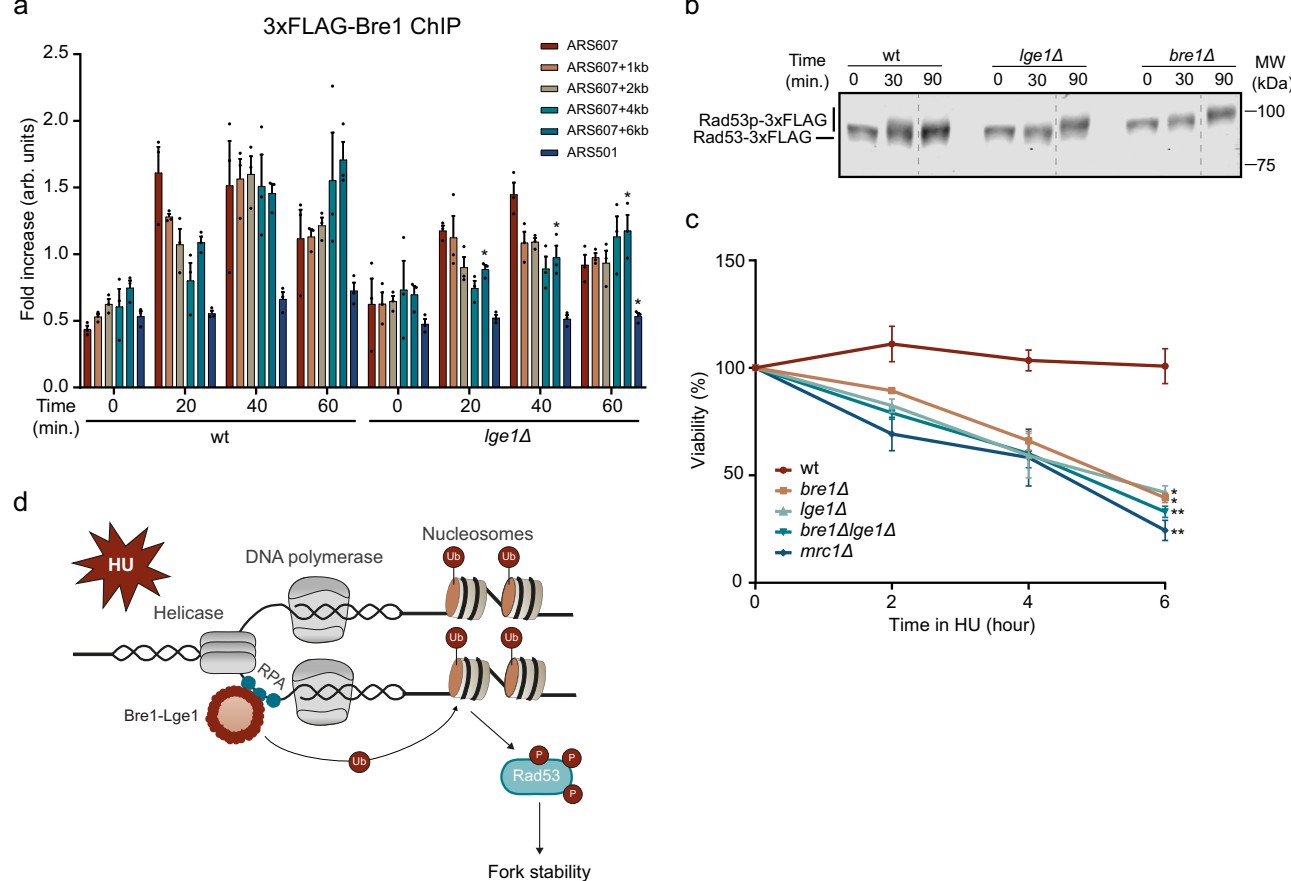

**Fig. 4 | Lge1 controls the intra-S-checkpoint and fork recovery via H2B ubiquitylation. a** ChIP-qPCR analysis of 3xFLAG-Bre1 at ARS607 and ARS501 in the indicated strains at different timepoints after release from G1 in S-phase in 200 mM HU. Data represent the mean relative fold enrichment + SEM of FLAG signal over beads only signal of *n* = 3 independent experiments. Values were normalized to a telomere region (TELVI-R). **b** Representative western blot analysis (*n* = 2) of Rad53-3xFLAG phosphorylation in the indicated strains at different timepoints after G1 arrest and release in S-phase in 200 mM HU. Dotted line indicates a cropped blot. A repeat of this blot is shown in Supplementary Fig. 4d. **c** Clonogenic survival of the indicated strains following exposure to 300 mM HU for the indicated time periods. Data represent the mean + SEM of *n* = 3 independent experiments. **d** Model showing the role of Bre1-Lge1 in response to replication stress (see text for details). Statistical significance compared to wt was calculated using the two-tailed unpaired Student's *t* test, assuming unequal variances, *p < 0.05, **p < 0.01. Source data are provided as a Source Data file.

---

absence of Lge1, while observing a complete loss of H2B ubiquitylation, we conclude that Bre1 cannot ubiquitylate H2B without its binding partner Lge1, likely due to the impaired functionality of Bre1.

## Lge1 controls the intra-S-checkpoint and fork recovery via H2B ubiquitylation

*Htb-K123R* mutants, in which H2B cannot be ubiquitylated at Lysine 123, show a delay in checkpoint response[32,33]. Given that Lge1 promotes H2B ubiquitylation, we next studied the impact of Lge1 loss on the intra-S-checkpoint by examining Rad53 phosphorylation status after HU (Fig. 4b and Supplementary Fig. 4d). While in wild-type cells Rad53 is activated after 30 min of HU exposure, the activation of Rad53 in *lge1Δ* or *bre1Δ* cells only became apparent after 90 min, indicative of a delayed checkpoint response. Thus, Lge1 promotes efficient Rad53-dependent checkpoint activation following HU.

Timely activation of the S-phase checkpoint is critical for the inhibition of late firing origins[19]. While the early origin ARS607 was fired as evidenced by Pol ε binding, we did not observe firing of the late origins ARS501, ARS404, and ARS316 in *lge1Δ* (Supplementary Fig. 4e), suggesting that the delay in Rad53 activation did not impact late origin firing. Furthermore, efficient checkpoint activation is critical for the recovery of cells from HU-induced replication fork stalling[39]. Therefore, we examined whether the delayed Rad53 activation in the absence of Lge1 could impact fork recovery. Loss of Lge1 or Bre1 impaired fork recovery after transient exposure of cells to HU (Fig. 4c), which is in line with previous work showing decreased fork recovery when H2B ubiquitylation was impaired[32]. Moreover, the combined loss of Lge1 and Bre1 led to a similar impairment in fork recovery, indicating that Lge1 and Bre1 act epistatically during this process. Collectively, these findings suggest that Lge1 sustains Bre1-dependent H2B mono-ubiquitylation near origins of replication both by stabilizing Bre1 and enhancing its activity, thereby facilitating an efficient intra-S-checkpoint response crucial for replication fork initiation, stability, and recovery following replication stress (Fig. 4d).

## Rox1 does not impact DNA replication by affecting cellular dNTP levels

Regulation by oxygen 1 (Rox1) is a highly conserved transcriptional repressor of hypoxic genes containing a nuclear HMG-box domain[40,41]. Rox1 also regulates the ribonucleotide reductase (RNR) genes, which are necessary for the biosynthesis of dNTPs[42]. To further validate the effect of Rox1 on DNA replication as observed in Repli-ID (Fig. 2a, c, d), we first performed a more extensive analysis of Pol ε progression near ARS607 at various timepoints after HU (Fig. 5a). In the absence of Rox1, like for Lge1, Pol ε levels were strikingly decreased at all timepoints in HU (Fig. 5a). To rule out the possibility that the decreased Pol ε levels were due to impaired

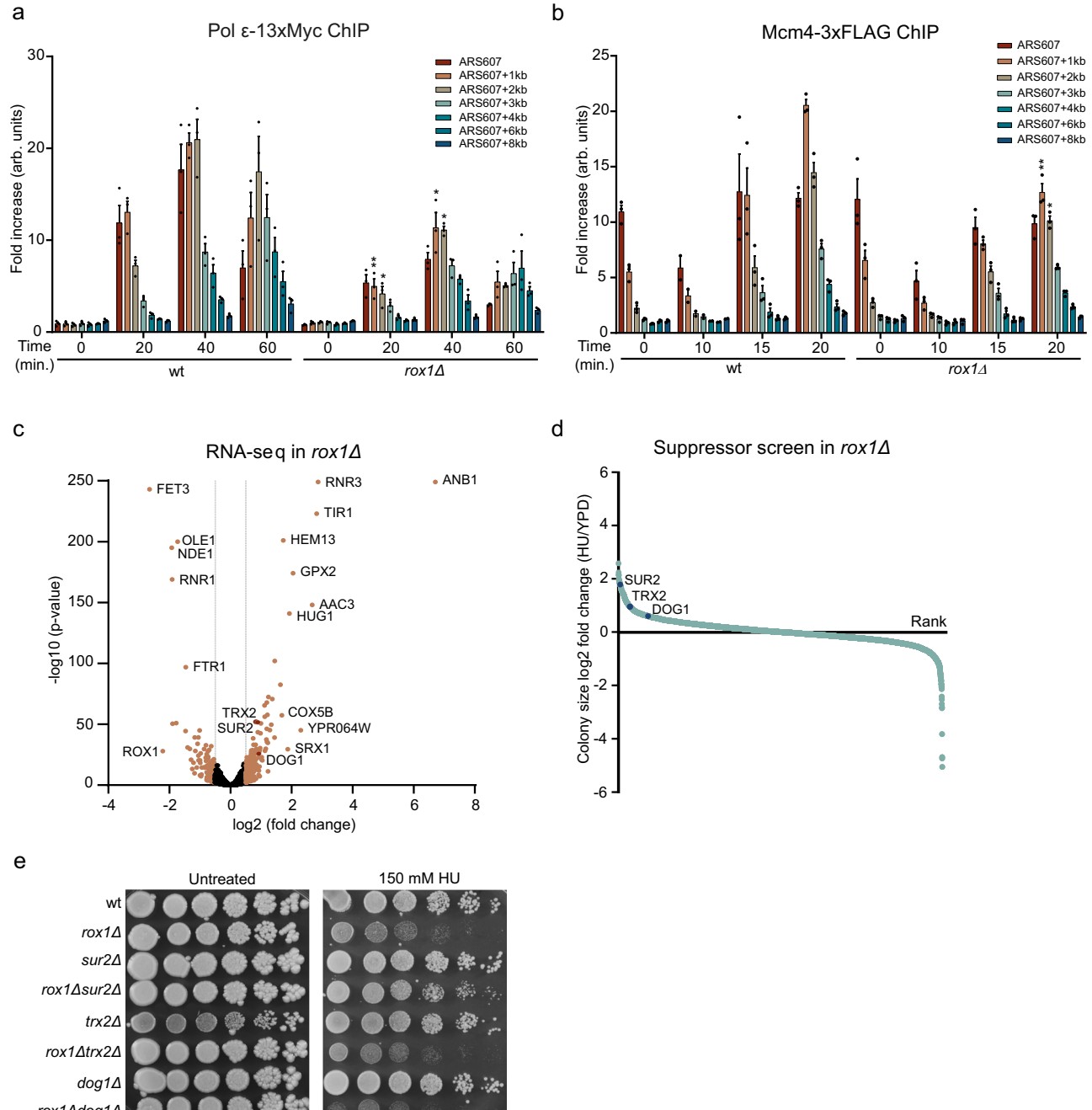

**Fig. 5 | Rox1 controls the response to replication stress by repressing Sur2.**
**a** ChIP-qPCR analysis of Pol ε-13xMyc at ARS607 in the indicated strains at different timepoints after release from G1 arrest in S-phase in 200 mM HU. Data represent the mean relative fold enrichment + SEM of Myc signal over beads only signal in $n = 3$ independent experiments. Values were normalized to a non-replicated region (ARS607 + 14 kb). Statistical significance compared to wt was calculated using the two-tailed unpaired Student's $t$ test, assuming unequal variances, *$p < 0.05$, **$p < 0.01$. **b** As in (**a**), except for Mcm4-3xFLAG. Data represent the mean relative fold enrichment + SEM of FLAG signal over beads only signal of $n = 3$ independent experiments. Values were normalized to a non-replicated region (ARS607 + 14 kb).

**c** Volcano plot of up- and downregulated genes identified by RNA-seq analysis in *rox1Δ* compared to wt cells following G1 arrest and release in S-phase in 200 mM HU. Data represent the mean of $n = 3$ independent experiments. $p$ values were calculated using the Wald test. Notably, for RNR3 and ANB1, the -log10 (p-value) exceeded 250, indicating that the p-value was effectively close to zero **d** Waterfall plot of colony size changes in a suppressor screen of *rox1Δ* double mutants grown on YPD plates with and without 300 mM HU. Data represent $n = 1$ experiment. **e** Spot dilution assay with the indicated strains which were generated de novo in the W303 background. Fivefold serial dilutions were spotted on medium without or with the indicated concentrations of HU. Source data are provided as a Source Data file.

replication licensing by the MCM helicase, we performed Mcm4-3xFLAG ChIP-qPCR experiments. Mcm4 was properly loaded in G1 ($t = 0$ min) in the absence of Rox1, and progressed similarly to that in wild-type cells (Fig. 5b), although its levels were slightly lower after 20 min. To confirm that the decrease in Pol ε levels results in reduced DNA synthesis, DNA copy number analyses were conducted. These

revealed that less DNA was synthesized in *rox1Δ* (Supplementary Fig. 5a), corroborating that Rox1 regulates DNA replication under HU-induced stress. However, how Rox1 impacts DNA replication under these conditions remains elusive.

In vitro, Rox1 represses expression of the RNR genes *RNR2*, *RNR3*, and *RNR4* by binding to their promoters[42]. Reverse transcriptase

quantitative PCR (RT-qPCR) analysis revealed that in G1-phase *RNR1* expression decreased twofold, while *RNR4* expression slightly, but significantly increased in *rox1Δ* (Supplementary Fig. 5b). After release into S-phase with HU, *RNR3* levels were strongly increased in wild-type cells (Supplementary Fig. 5c), agreeing with previous reports[43,44]. However, loss of Rox1 resulted in an additional tenfold increase in *RNR3* expression and a fivefold reduction in *RNR1* expression under these conditions (Supplementary Fig. 5b). Despite reduced *RNR1* mRNA levels, the Rnr1 protein levels did not decrease, whereas elevated Rnr3 protein levels were observed under these conditions (Supplementary Fig. 5d). The RNR complex comprises Rnr1/Rnr3 homodimers or heterodimers and Rnr2/Rnr4 heterodimers. Increased Rnr3 protein levels may alter RNR complex composition, affecting dNTP production[45]. However, dNTP levels did not change significantly in the absence of Rox1 during both G1 and S-phase with HU (Supplementary Fig. 5e), suggesting that the impact on replication fork progression cannot be explained by alterations in RNR protein expression or dNTP levels.

## Rox1 controls the response to replication stress by repressing Sur2

Since Rox1 is a transcriptional repressor, we speculated that it may affect replication fork progression/stability by regulating genes involved in DNA replication and repair, either directly or indirectly. Although it was previously shown by microarray analysis that Rox1 regulates genes involved in mitochondrial respiration/oxidative phosphorylation and heme, sterol and unsaturated fatty acid synthesis[46,47], its genome-wide targets are largely unknown. We therefore conducted RNA-sequencing (RNA-seq), which revealed 137 genes repressed and 89 genes induced by Rox1 (Fold change > 1.5 and $p < 0.05$) (Fig. 5c). Since Rox1 is known as a transcriptional repressor, we focused on the genes it downregulates. *ANB1*, *AAC3*, *HEM13*, and *COX5B* were among the strongest downregulated genes, which were also found by microarray analysis in earlier reports[46,47], validating the RNA-seq analysis. Using a PANTHER GO Slim analysis[20–22], we identified significantly upregulated biological processes such as response to oxidative stress, consistent with findings from Kwast et al.[46] and Ter Linde and Steensma[47], as well as homeostatic processes (Supplementary Fig. 6a). However, no connections to DNA replication and repair were found.

Next, we aimed to determine if any upregulated genes identified by RNA-seq in *rox1Δ* could account for the HU sensitivity observed for this mutant. To this end, we performed a suppressor screen to identify genes whose loss would alleviate the HU sensitivity of *rox1Δ* (Fig. 5d). A genome-wide collection of *rox1Δ* double mutants was generated by crossing the *rox1Δ* query strain to an array of ~4700 deletion mutants using SGA technology[48]. Subsequently, colony growth of the double mutants was measured both in the absence and presence of HU and analyzed using SGAtools[49]. In total, 431 gene deletions were identified that suppressed the lack of growth on HU in *rox1Δ* (Fold change > 1.5) (Fig. 5d). No significantly enriched biological functions were revealed by PANTHER GO Slim analysis[20–22]. To identify upregulated genes in *rox1Δ* that contribute to the HU sensitivity observed in this mutant, we compared the genes identified by the suppressor screen with those from the RNA-seq analysis. *SUR2*, *TRX2,* and *DOG1* (FC > 1.5 for both RNA-seq and suppressor screen) were among the top hits in the suppressor screen (Fig. 5d), whose expression was upregulated in the RNA-seq analysis (Fig. 5c). RT-qPCR analysis confirmed that *SUR2*, *TRX2*, and *DOG1* were overexpressed in *rox1Δ* cells (Supplementary Fig. 6b). Importantly, while the loss of *SUR2*, *TRX2*, and *DOG1* rescued the HU sensitivity of *rox1Δ* cells in spot dilution assays in the S288C background (Supplementary Fig. 6c), which was used in the suppressor screen, we observed this phenotype for *SUR2*, but not *TRX2* or *DOG1*, in the W303 background (Fig. 5e), indicating background-specificity. Thus, Rox1 controls the cellular response to HU by repressing Sur2 expression.

## Rox1 regulates ceramide levels via Sur2 to control the response to replication stress

*SUR2* encodes for the sphinganine C4-hydroxylase, which is required for the production of sphingolipids, a major component of the plasma membrane[50]. Specifically, Sur2 catalyzes the hydroxylation of dihydrosphingosine to phytosphingosine, which in turn is converted into phytoceramide[51]. Ceramides play a role in activating PP2A phosphatase[52], which counteracts the DNA damage response[53]. Based on this, we hypothesized that the HU sensitivity in the absence of Rox1 could be the result of increased, Sur2-dependent ceramide production. To test this, we examined how the addition of cell-permeable C2 ceramide, known to activate PP2A[52], could impact the HU sensitivity of *rox1Δ, sur2Δ,* and *rox1Δsur2Δ* using spot dilution assays (Fig. 6a). Again, we observed that the HU sensitivity of *rox1Δ* was rescued by Sur2 loss in the absence of ceramide. In the presence of ceramide, we found an overall increase in HU sensitivity. Notably, the HU sensitivity of *rox1Δ* could not be rescued by *SUR2* deletion when ceramide was present, suggesting that Rox1 regulates ceramide levels through Sur2, thereby affecting the cellular response to HU-induced replication stress. To rule out that the HU phenotype is not caused by the production of ROS, we tested the HU sensitivity of *rox1Δ* and *rox1Δsur2Δ* in the presence of NAC (Supplementary Fig. 6d). Although the HU sensitivity of *ccs1Δ* (positive control) was alleviated by reducing ROS levels with NAC addition, neither *rox1Δ* nor *rox1Δsur2Δ* showed any rescue. This indicates that the HU phenotype of *rox1Δ* is caused by the depletion of dNTP pools and the stalling of replication forks.

## Rox1 affects DNA replication by regulating checkpoint activation and S-phase entry through Sur2/ceramide control

Ceramides are involved in activating PP2A phosphatase, which impacts the intra-S-checkpoint by Rad53 dephosphorylation, and S-phase entry by regulating the G1/S transition[52,53]. To examine whether altered Rad53 phosphorylation levels may underly the impaired HU response in *rox1Δ*, we assessed Rad53 phosphorylation status by western blot analysis (Fig. 6b and Supplementary Fig. 6e). Increased levels of Rad53 phosphorylation were observed in HU-treated wild-type cells, which could be suppressed by the addition of extracellular C2 ceramide, agreeing with a previous report[53]. Importantly and also in line with high ceramide levels suppressing Rad53 phosphorylation, we found that the Rad53 phosphorylation levels were reduced in *rox1Δ* when compared to that in wild-type cells. The addition of ceramide to *rox1Δ* cells further impaired Rad53 phosphorylation, potentially by exacerbating the already elevated ceramide levels in these cells. Thus, Rox1 controls the Rad53-induced intra-S-phase checkpoint by suppressing Sur2/ceramide-dependent PP2A activation.

Next, we assessed whether Rox1 also impacts S-phase entry by regulating the G1/S transition. Budding index analysis of wild-type cells showed an increase in budded cells at 30 min after release in HU, while *rox1Δ* cells only showed an increase at 40 min after release in HU, suggesting a delay in S-phase entry (Fig. 6c). This delay is HU-dependent as no pronounced impact on S-phase entry was observed in unperturbed conditions (Supplementary Fig. 6f). Deletion of *SUR2* in *rox1Δ* restored S-phase entry to wild-type levels, suggesting the delay in S-phase entry is Sur2/ceramide-dependent (Fig. 6c). Indeed, supplementing *rox1Δsur2Δ* cells with extracellular C2 ceramide impaired S-phase entry, decreasing budding index levels to *rox1Δ* levels (Fig. 6d). Of note, the addition of ceramide to *rox1Δ* cells further delayed S-phase entry, potentially by exacerbating the already elevated ceramide levels in these cells. Collectively these results show that Rox1 promotes S-phase entry by regulating the Sur2/ceramide-dependent G1/S transition.

Rox1 was identified in the Repli-ID screens as a factor that impacts Pol ε progression. We therefore investigated whether this is associated with ceramide control through Sur2 repression. Strikingly, ChIP-qPCR

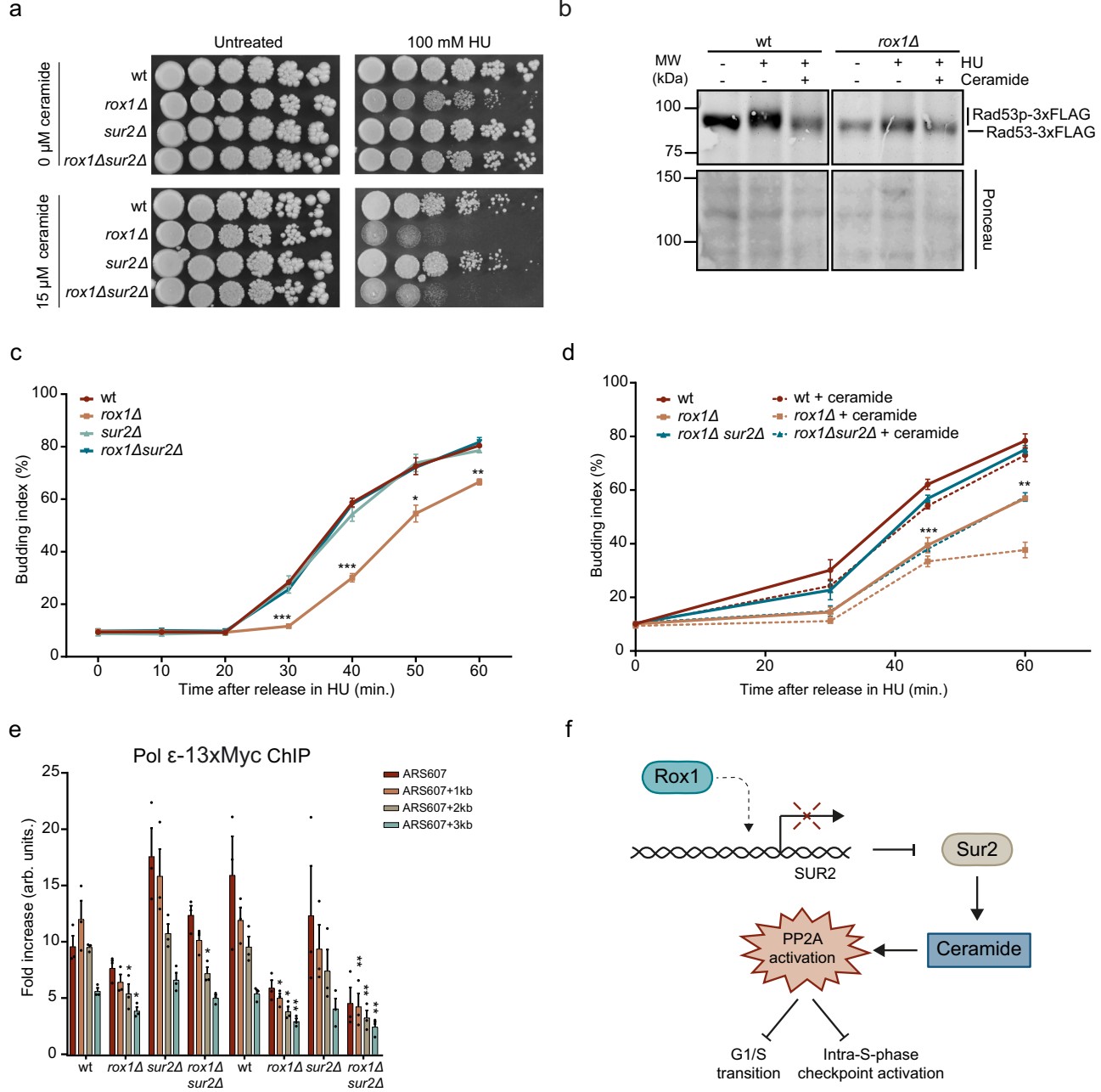

**Fig. 6 | Rox1 affects DNA replication by regulating checkpoint activation and S-phase entry through Sur2/ceramide control. a** Spot dilution assay with the indicated strains. Fivefold serial dilutions were spotted on medium without or with 100 mM HU and/or 15 μM ceramide. **b** Representative western blot analysis (*n* = 2) of Rad53-3xFLAG phosphorylation in the indicated strains after G1 arrest and a 60 min release in S-phase in 200 mM HU and/or 15 μM ceramide. Total protein staining by Ponceau is a loading control. A repeat of this blot is shown in Supplementary Fig. 6e. **c** Budding index analysis of the indicated strains following G1 arrest and release in S-phase in 200 mM HU. Data represent the mean + SEM of *n* = 3 independent experiments. Statistical significance between *rox1Δ* compared to wt is shown. **d** As in (**c**), except for the indicated strains in 200 mM HU and/or 15 μM

ceramide. Data represent the mean + SEM of *n* = 3 independent experiments. Statistical significance between *rox1Δsur2Δ* compared to *rox1Δsur2Δ* + ceramide is shown. **e** ChIP-qPCR analysis of Pol ε-13xMyc at ARS607 in the indicated strains after G1 arrest and a 20 min release in S-phase in 200 mM HU and/or 15 μM ceramide. Data represent the mean relative fold enrichment + SEM of Myc signal over beads only signal in *n* = 3 independent experiments. Values were normalized to a non-replicated region (ARS607 + 14 kb). **f** Model showing the role of Rox1 and Sur2 during the replication stress response (see text for details). Statistical significance compared to wt was calculated using the two-tailed unpaired Student's *t* test, assuming unequal variances, *\*p* < 0.05, *\*\*p* < 0.01, *\*\*\*p* < 0.001. Source data are provided as a Source Data file.

revealed that the reduced Pol ε enrichment near ARS607 in *rox1Δ* was rescued in *rox1Δsur2Δ* cells (Fig. 6e). Moreover, this rescue was negated when *rox1Δsur2Δ* cells were treated with C2 ceramide. Together, this suggests that Rox1 plays a crucial role in ensuring proper Pol ε levels under replication stress conditions by promoting S-phase entry and activating the intra-S-phase checkpoint. This regulatory function is

dependent on cellular ceramide levels, which are controlled by Sur2 repression (Fig. 6f).

## Discussion
Here we present an approach, termed Repli-ID, to study replication fork stability and progression in thousands of yeast mutants

simultaneously. By examining the enrichment of Pol ε at a barcoded origin of replication, ARS404, in the genome-wide collection of yeast deletion strains, we uncovered 423 known and new regulators that promote Pol ε binding at stalled replication forks. Mechanistic follow-up studies on two regulators from the screen, Lge1 and Rox1, provided insights into how epigenetic and metabolic pathways control DNA replication under stress conditions. While Lge1 drives DNA replication by promoting Bre1-dependent H2B ubiquitylation directly at perturbed replication forks, Rox1 controls DNA replication by repression of Sur2-dependent ceramide production, which impairs G1/S transition and checkpoint activation. Thus, Repli-ID generated a comprehensive overview of factors affecting Pol ε in yeast and provides a resource for follow-up studies that aim to understand the mechanistic role of these factors in DNA replication.

Using Repli-ID, we identified 423 mutants in which Pol ε levels were reduced both at 40 and 80 min after release in S-phase (Fig. 2a), suggesting a defect in fork stability or fork progression. However, a decrease in Pol ε levels after HU treatment may also indicate that intact replisomes moved away from sites of DNA synthesis, as demonstrated in checkpoint-deficient mutants[15]. Additionally, since our barcode is located 53 bp from the origin of replication (ARS404), a lack of Pol ε could also result from insufficient origin firing when mutants are impaired in origin loading, replication licensing by the MCM helicase, or the G1 to S-phase transition (Supplementary Fig. 2c). Future research using genome-wide analysis of Pol ε and/or MCM binding in combination with flow cytometry-based cell cycle profiling may identify the mutants with impaired origin firing and/or cell cycle progression.

The Repli-ID screens also revealed mutants in which Pol ε levels were decreased only at 40 min or only at 80 min after release in S-phase, indicating that replication forks may have progressed faster or slower in these mutants, respectively, or have defects in origin firing. Additionally, the Repli-ID screens led to the identification of 128 mutants in which Pol ε levels were enhanced at 40 and 80 min after release in S-phase (Fig. 2a). In the current study, we focused on mutants in which Pol ε levels were reduced at both timepoints as these may most likely suffer from replication fork instability. However, enhanced Pol ε levels may also be indicative of replication fork stalling, but not unstable replication forks, warranting future mechanistic studies of mutants showing this phenotype. Collectively, these findings further support the comprehensiveness of the Repli-ID screens, revealing a wealth of factors that could be further studied to better understand how Pol ε is mechanistically regulated to prevent replication fork instability and/or promote replication fork progression.

Repli-ID was conducted using the freely available barcoder library of yeast strains, in which the barcodes were integrated at the HO locus, located downstream of the late firing origin ARS404[54] (Fig. 1). Consequently, Repli-ID had to be performed in mutants expressing the *sld3-38A dbf4-4A* mutants to allow firing of ARS404 in early S-phase. This method leads to the simultaneous activation of all origins, including neighboring origins of ARS404, which could potentially influence Pol ε levels at this origin. Foss et al. reported the identification of 1600 origins, including several sites outside of known origins, referred to as alternate origins, from which replication can be initiated[55]. Within a 16 kb region, an estimated maximum distance that forks could travel in 80 min under HU conditions[56,57], five alternate origins were found[55]. Although we do not know if replication is initiated from these alternate origins in the mutants with overexpressed *sld3-38A dbf4-4A* identified in our screen, we cannot exclude the possibility that this overexpression indirectly affected Pol ε levels near ARS404 in these mutants. Furthermore, Morafraile et al. demonstrated that increased replication initiation can lead to the accumulation of DNA damage markers, such as γH2A and Rad52, due to topological stress[30]. Given that we examined relatively early timepoints in S-phase ($t = 40$ and $t = 80$) and monitored Pol ε binding in close proximity of the origin of replication ARS404, which was far away from other origins (30.4 kb

(ARS403) and 39.7 kb (ARS405)), topological stress is likely minimized. Moreover, although we cannot entirely rule out the possibility that the expression of *sld3-38A dbf4-4A* may have caused additional replication stress and/or DNA damage during the Repli-ID screens, the brief duration of expression for these mutants (30 min) may have mitigated this effect.

Repli-ID can be performed in other configurations, for instance, by using tagged Rad51 recombinase, to identify mutants in which loss of Pol ε leads to replication fork collapse and a failure to repair replication-associated DNA breaks by HR[58]. Alternatively, interrogating the Rad18 status at stalled replication forks during Repli-ID may reveal regulators of DNA damage tolerance pathways[59]. Finally, by examining the loading of Cdc45-Mcm2-7-GINS (CMG) helicase at origins of replication, Repli-ID may reveal factors that impact replication licensing[60,61]. These examples underscore the versatility of the Repli-ID approach and its potential for various applications.

The Repli-ID screens, in combination with mechanistic follow-up studies, showed that Lge1 drives Bre1-dependent H2B mono-ubiquitylation at Lysine 123, which promotes replication fork initiation, stability and recovery after replication stress by both stabilizing Bre1 at the fork and enhancing Bre1 activity (Fig. 4d). While this suggests that Lge1 is a key constituent of the Bre1 complex, not only during transcription regulation, but also during the replication stress response[31–34], the structure-function relationship between Lge1 and Bre1 during this latter response remains elusive. Biochemical reconstitution and liquid–liquid phase separation experiments showed that, following binding of Bre1 to Lge1, Lge1 acts as a scaffold protein whose intrinsically disordered region phase separates[62]. The resulting condensates consist of a core of Lge1 proteins encapsulated by an outer shell of Bre1 protein. Subsequently, Rad6 is recruited to this layered condensate, ultimately facilitating H2B ubiquitination along gene bodies. It would be of interest to determine whether this layered Bre1/Lge1/Rad6-containing condensate also augments catalytic activity toward H2B at stalled replication forks to promote their recovery. To this end, it is worth mentioning that Bre1 recruitment to stalled forks has been shown to occur in a manner dependent on RPA[63], which itself also forms condensates when bound to ssDNA at telomeres[64]. Whether these RPA condensates also form around ssDNA at stalled forks and have functional relevance for Bre1 activity at these structures awaits further investigation. Finally, how loss of Bre1-dependent H2B ubiquitylation affects DNA replication under stress conditions remains poorly understood. One possible explanation could be that H2Bub present at the origins of replication is necessary for efficient replication initiation, while H2Bub at stalled forks promotes checkpoint activation and fork stability. It has been suggested that Bre1-dependent H2B ubiquitylation recruits specific proteins to distinct chromatin contexts, promoting structural changes by chromatin remodeling to enhance DNA more accessibility[65], or facilitating histone deposition and stabilizing nucleosomes behind the initiating/advancing fork[33] However, whether these processes also promote replication fork initiation, stability and the recovery of stalled forks remains to be studied.

WAC is the human homolog of Lge1, which ubiquitylates H2B together with the E3 ligase proteins RNF20/40 (hBre1)[66]. Similar to yeast, impaired H2B ubiquitylation and RNF20/40 deficiency have been shown to cause replication stress in human cells[67]. Given these phenotypic similarities, we hypothesize that WAC, similar to Lge1 in yeast, may play a role in regulating DNA replication during stress conditions. Interestingly, WAC loss-of-function mutations are known to result in the so-called DeSanto–Shinawi syndrome (DESSH)[68]. This disease is characterized by developmental delay, intellectual disability, behavioral problems, and dysmorphic features[68,69], which are clinical manifestations also found in replication diseases[70,71]. Hence, investigating whether genetic defects in WAC result in replication abnormalities in cells of DESSH patients is crucial to determine whether DESSH could be categorized as a replication-related disorder.

In addition to Lge1, the Repli-ID screens and mechanistic follow-up studies also revealed an unanticipated role for the hypoxic repressor Rox1 in promoting replication fork stability/progression[41]. Mechanistically, we found that Rox1 represses Sur2, thereby protecting cells against replication stress. The transcriptional repressor Rox1 is known to bind via its HMG box to the conserved binding motif YYYATTGTTCTC[40]. Interestingly, a putative Rox1 binding motif (CCTATTGTCTTA) in the promoter of SUR2 was identified[47], suggesting Rox1 represses Sur2 directly by binding on its promoter.

Previously, it was shown that in the absence of Sur2 the conversion of dihydrosphingosine and dihydroceramide to phytosphingosine and phytoceramide is impaired[72]. In the absence of Rox1, Sur2 is over-expressed and the balance of ceramides is expected to shift toward high amounts of phytoceramides and low amounts of dihydroceramides. Although the precise nature of the ceramides being increased in the absence of Rox1 is unknown, we showed that intra-S-phase checkpoint activation, G1/S transition and Pol ε progression can be modulated by SUR2 deletion or the addition of extracellular C2 ceramides (Fig. 6). This provides evidence that Rox1 indeed regulates the replication stress response by impacting Sur2-dependent ceramide regulation. Metabolomics studies will, however, be required to disentangle which type of ceramides are dysregulated in the absence of Rox1. Such studies may also help to further elucidate the ceramides that activate PP2A phosphatase, a process that is driven by metabolic circuits involving Irc21, a putative cytochrome b5 reductase that promotes dihydroceramide production. Importantly, loss-of-function mutations in IRC21 rescued mec1 mutant phenotypes coupling metabolic pathways to the DNA damage response[53]. Here we extend these findings by integrating Rox1/Sur2-dependent ceramide regulation into this PP2A/Mec1-dependent response (Fig. 6f). Further research is needed to determine whether Irc21 functions as an integral component of the Rox1/Sur2-dependent ceramide pathway that regulates the G1/S transition and checkpoint activation in response to replication stress.

While Sur2 does not have a mammalian orthologue[73], the HMG box of Rox1 shows similarities with the HMG DNA binding domain of mammalian transcription factors SRY/Sox proteins[74]. These proteins have been shown to be upregulated in different cancer types and have been found to be associated with poor prognosis and therapy resistance[75]. However, it is unknown whether the SRY/Sox proteins have a similar function as Rox1 in the production of ceramides and PP2A activation in human cells, where this metabolic pathway is important for tumor suppression by mediating apoptosis or growth inhibition[76,77]. Unraveling the potential connection between SRY/Sox proteins, ceramide production, and PP2A activation in cancer will deepen our understanding of cancer development, offering potential for refining therapy strategies.

## Methods

### Media and yeast strains
Yeast strains, which are listed in Supplementary Table 1, were grown in YPAD or synthetic complete medium (SC). The strains that were used in the initial Repli-ID validation studies were taken from the YKO library[48] and verified by PCR and phenotypic analysis. Yeast libraries were crossed by SGA technology[48] using the RoToR (Singer Instruments). For follow-up research, the yeast strains used were derivatives of W303-1A (Supplementary Table 1). Gene deletions and epitope tags on endogenous genes were generated by PCR-based gene targeting[78]. DNA cloning design and sequence alignment were carried out using SnapGene (7.1.1). Yeast strains were generated using lithium acetate-based transformations[79].

### Antibodies
Antibodies used were anti-Myc 9B11 (#2276; Cell Signaling; ChIP: 2 μl/40 μl beads), anti-FLAG (F1804; Sigma; WB: 1:5000; ChIP: 2 μl/40 μl beads), anti-Pgk1 (#459250; Invitrogen; WB: 1:5000), anti-H2Bub (obtained from Fred van Leeuwen, NKI, Amsterdam, the Netherlands, described in van Welsem et al.[80]; WB: 1:5000; ChIP: 0.2 μl/40 μl beads), anti-RNR1 (AS214608; Agrisera; WB: 1:5000), anti-RNR3 (AS09574; Agrisera; WB: 1:1000), anti-tubulin (T6199; Sigma; WB: 1:1000).

### Repli-ID screen
Repli-ID is based on the Epi-ID approach from Vlaming et al.[12]. Briefly, the collection of barcoded yeast mutants was generated by crossing a set of 1140 barcoded strains (obtained from Corey Nislow, University of British Colombia, Vancouver, Canada)[54] to the MATα NatMX knockout collection (YSC1053, Eurogentec) of ~4700 yeast mutants[48] using a ROTOR (Singer Instruments). Deletion mutants were divided over five subsets to provide each mutant a unique barcode in each subset and a wild-type barcode set as a control. Next, in a second crossing, the strain containing Myc-tagged Pol ε (Pol2-9xMyc) and the galactose-inducible sld3-38A dbf4-4A construct (adapted from Zegerman and Diffley[19]), which was integrated at the BAR1 locus, was crossed into barcoded KO libraries to generate the final Repli-ID library using a ROTOR. To enable galactose-induced expression of mutant forms of Sld3 and Dbf4 (sld3-38A dbf4-4A) during the screen, libraries were grown overnight on YPA plates containing raffinose at 30 °C. The next day, they were transferred and diluted to 0.1 OD and grown for 3 h in liquid YPA containing raffinose at 30 °C. To arrest the cells in G1, α-factor (Zymo research; Y1001) was added for 2.5 h with extra additions after 1 h and after 2 h. During the last 30 min of α-factor arrest, galactose was added to induce expression of the Sld3 and Dbf4 mutants (sld3-38A dbf4-4A) and allow firing of all origins during release into S-phase. α-factor was removed by washing the cells twice in YPAD prior to their release in YPAD containing 200 mM HU (Sigma-Aldrich; H8627) and 50 μg/ml pronase (Merck Millipore; 53702). Cells were crosslinked using 1% formaldehyde (Sigma; 47608) after 40 and 80 min. ChIP was performed as in Vlaming et al.[12], except that anti-Myc antibody was used. DNA was isolated and the barcode closest to ARS404 was PCR amplified using Downtag primers as described in Vlaming et al.[12]. PCR products were deep-sequenced on a single-end flow-cell Illumina Hi-Seq2500.

### Repli-ID analysis
Barcodes were counted, and depletion or enrichment of each barcode was determined as described by Vlaming et al.[12] using xcalibr (https://github.com/NKI-GCF/xcalibr) and RStudio (2023.09.1), with the exception that barcodes with read numbers below 0.0025% of the total reads in an input sample were excluded as they were considered absent from the plate. Additionally, median-normalized barcode scores from the ChIPs were divided by the corresponding input scores. The ratios from two replicate screens were averaged for t = 40 min and t = 80 min, and overlapping ORFs were identified. This analysis produced Repli-ID results for 2905 mutants (Supplementary Data 1).

### GO Slim analysis
The GO Slim (biological process) analysis was executed using the PANTHER GO enrichment analysis[21] (https://geneontology.org)[20,22].

### ChIP-qPCR
For Repli-ID validation experiments, the ChIP protocol from the Repli-ID screens was used. All other ChIP experiments were performed as previously described[81] in W303 background, which is commonly used for replication ChIP assays[6,17], yet is distinct from the S288C-derived BY background used in the Repli-ID screens and initial validation experiments[82]. Briefly, cells were grown for 3 h, treated with α-factor for 2 h, washed once in YPAD medium and released in YPAD containing 200 mM HU. Samples were collected at 0, 20, 40, and 60 min after release and fixed with 1% formaldehyde. Input and immunoprecipitated DNA was purified and analyzed by qPCR using a CFX384 Touch Real-Time PCR detection system (BioRad). Data analysis

was performed using BioRad CFX Manager Software 3.1. Relative enrichment was determined by 2−ΔΔCt method. Signal for Dynabeads (M-280 Sheep anti-mouse; Sigma; 11202D) alone was used to correct for background. Primers used are listed in Supplementary Table 2.

### Spot dilution test
Cells were grown overnight in rich media (YPAD) and set to OD 0.5 (~7 × 10⁶ cells/ml). Cells were spotted in fivefold serial dilutions on YPAD plates without HU and YPAD plates containing 50, 100, 150, or 200 mM HU, and were grown for 3 days at 30 °C before images were taken.

### Budding index
Cells were grown for 3 h and treated with α-factor for 2 h, washed once in YPAD medium and released in YPAD without and YPAD containing 200 mM HU at, respectively, room temperature or 30 °C. Samples were taken every 10 or 15 min for 2 h and fixed in 4% paraformaldehyde at room temperature for 15 min, washed and resuspended in KPO4/Sorbitol solution (10 mM KPO4, 1.2 M Sorbitol, pH = 7.5). Brightfield images were captured with a AxioImager M2 widefield fluorescence microscope (Zeiss) equipped with 100x PLAN APO (1.4 NA) oil-immersion objectives (Zeiss). Image analysis was performed using ZEN 2012 and Image J (1.48v) software.

### RT-qPCR
Cells were grown to 5 × 10⁶ cells/ml in YPAD and synchronized with α-factor for 2 h (G1/t0 sample). Cells were washed with YPAD medium and released for 60 min in YPAD containing 200 mM HU (t60 sample). 1.5 × 10⁷ cells were harvested and RNA was isolated with the RNeasy Mini kit according to the manufacturer's protocol (Qiagen; 74104). Genomic DNA was digested using the RNase-Free DNase set (Qiagen; 79254). Subsequently, RNA was purified using the RNeasy Mini kit. cDNA was prepared using the GoScript Reverse Transcriptase System (Promega; A5001). The expression levels of the *RNR* genes were quantified by qPCR using a CFX384 Touch Real-Time PCR detection system (BioRad) and normalized to the TAF10 locus. Data analysis was performed using BioRad CFX Manager Software 3.1. Primers used are listed in Supplementary Table 2.

### Western blot analysis
Whole cell extracts were prepared from 10 ml culture in log phase (~1–2 × 10⁷ cells/ml). Cell pellets were precipitated with 20% trichloracetic acid (TCA) and disrupted by bead beating with the bioruptor. Samples were dissolved and boiled in 2X Laemmli buffer. Proteins were resolved in 4–12% polyacrylamide gels (NuPAGE; NP0321) and transferred onto PDVF membranes (Millipore; IPFL00010). Membranes were blocked with blocking buffer (Rockland; MB-070-010) or 5% milk (Campina) in PBS followed by overnight incubation with primary antibody in blocking buffer or 2% milk in PBS at 4 °C. Membranes were washed with 0.1% Tween20 in PBS. Secondary antibody incubations were performed for 1 h in blocking buffer or 2% milk in PBS at room temperature. Membranes were subsequently scanned on a LI-COR Odyssey® V3.0 IR Imager (Biosciences) and analyzed using ImageStudio Lite 5.2.5. Uncropped blots are provided in the Source Data file.

For Rad53-3xFLAG phosphorylation analysis, cells were grown for 3 h, synchronized with α-factor for 2 h, washed with YPAD and released in YPAD containing 200 mM HU. Samples were collected at 0, 60, and 90 min after release, or only at 60 min after release. Whole cell extracts were prepared by TCA precipitation as described above except that a 10% Tris-glycine gel was used and blocking was performed in 2% milk.

### dNTP measurements
dNTP quantification was performed as previously described[83]. Briefly, cells were grown for 3 h and treated with α-factor for 2 h, washed once

in YPAD medium and released in YPAD containing 200 mM HU at 30 °C. At timepoints 0 and 60 min, ~3.7 × 10⁸ cells were collected onto a 0.8 µm nitrocellulose filter, resuspended immediately in ice-cold lysis solution and snap frozen in liquid nitrogen. Nucleotides were extracted from the cells and dNTPs were separated from rNTPs using a boronate column. Both dNTPs and rNTPs were analyzed by HPLC, and dNTP levels were normalized to total rNTP levels in each sample.

### RNA-seq analysis
Three independent yeast colonies were used for inoculation. Overnight cultures were diluted in fresh medium. Cells were grown for 3 h, treated with α-factor for 2 h, washed once in YPAD medium and released in YPAD containing 200 mM HU. Samples were collected after α-factor synchronization (t0) 60 min after release in HU (t60). Total RNA was isolated using the RNeasy Mini kit and treated with the RNase-Free DNase Set to remove any contaminating genomic DNA. Three independent biological replicates for each condition were subjected to sequencing on an Illumina NovaSeq6000 PE150. Sequencing data were analyzed using www.usegalaxy.org[84]. Adaptors were removed using TRIMMOMATIC, reads were aligned to the UCSC SacCer3 reference genome (April 2011)[85] using HISAT2, reads were counted using FEATURECOUNTS and the differential expression was calculated via DESeq2. The Wald test was used to calculate *p* values. The 137 strongest hits (log2(fold change) > 1.25) were used for GO Slim analysis. The GO Slim (biological process) analysis was executed using the PANTHER GO enrichment analysis[21] (https://geneontology.org)[20,22].

### Suppressor screen
To create *rox1Δ* double mutants, a *rox1Δ* query strain was crossed with an array of ~4700 deletion mutants from the YKO library (YSC1053) using a ROTOR (Singer Instruments)[48]. Colony growth of the double mutants was measured in the absence and presence of 300 mM HU and analyzed with SGAtools (http://sgatools.ccbr.utoronto.ca/)[49]. The colony size was normalized to the median colony size on the plate and a score per mutant was calculated by dividing the colony size on the HU-treated plate by the colony size on the YPD plate. Data were removed for normalized colony sizes which were 0.

### Copy number assay
Cells were grown for 3 h in YPAD, synchronized with α-factor for 2 h, washed in YPAD, and released in YPAD containing 200 mM HU at 30 °C. Five milliliters of culture was collected at 0, 20, 40, and 60 min after release from G1, fixed with 0.2% Na-azide for 10 min and washed with 10 mM Tris, 50 mM EDTA. For genomic DNA extraction, cells were digested in 1 M Sorbitol, 0.1 M Sodium citrate pH 7.0, 60 mM EDTA, 8 mg/ml β-Mercaptoethanol, 2 mg/ml Zymolyase 20T (MP Biomedicals, ref 8320921) for 45 min, and DNA was isolated using the DNeasy Blood and Tissue kit (Qiagen; 69504) following the manufacturer instructions. The amount of genomic DNA at ARS607 and downstream loci was quantified by qPCR using a CFX384 Touch Real-Time PCR detection system (BioRad). The ratio of DNA in HU-arrested cells to that in G1 was calculated and normalized to the ARS607 + 14 kb locus. Data analysis was performed using BioRad CFX Manager Software 3.1. Primers used are listed in Supplementary Table 2.

### Flow cytometry
Cells were grown for 3 h in YPAD, synchronized with α-factor for 2 h, washed in YPAD, and released in YPAD containing 10 mM HU at 30 °C. Cells were then collected and fixed in 70% cold ethanol at specified time intervals. After overnight fixation, cells were incubated with 200 µg/ml RNase A at 37 °C for 3 h. Following resuspension of the cells, propidium iodide (PI) was added to a final concentration of 10 µg/ml. Finally, cells were briefly sonicated, after

which 250,000 events were recorded on a Novocyte (ACEA Biosciences, Inc.) and analyzed with NovoExpress software.

### Fork recovery assay

Cells were grown for 3 h and treated with α-factor for 2 h, washed once in YPAD medium, and released in YPAD containing 300 mM HU at 30 °C. Samples were taken after 0, 2, 4, and 6 h, plated on YPAD plates, and grown for 2 or 3 days at 30 °C. Colonies were counted and viability was calculated by normalizing to timepoint $t = 0$ h for each condition.

### Statistics

Statistical significance was calculated using the two-tailed unpaired Student's $t$ test using Excel (Microsoft), assuming unequal variances, $*p < 0.05$, $**p < 0.01$, $***p < 0.001$, except for the Repli-ID and RNA-seq analyses (see "Repli-ID analysis" and "RNA-seq analysis" subsections in the "Methods" section). Graphs were created using GraphPad Prism 6.

### Reporting summary

Further information on research design is available in the Nature Portfolio Reporting Summary linked to this article.

### Data availability

The Repli-ID screen data generated in this study have been deposited in the NCBI Sequence Read Archive (SRA) under accession code PRJNA1079539. The RNA-seq data generated in this study have been deposited in the NCBI SRA under accession codes PRJNA1076696. All other data generated in this study are provided in the main manuscript and its Supplementary Information files. Source data are provided with this paper.

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

## Acknowledgements

We would like to thank Fred van Leeuwen and Hanneke Vlaming for discussions and generously providing reagents, Robin van Schendel and Diana van den Heuvel for bioinformatics help, and Sushma Sharma for critical reading of the manuscript. This work was financially supported by grants from the Swedish Cancer Society (22 2377 Pj to A.C.) and the Swedish Research Council (2022–00675_VR to A.C.), as well as the Dutch Research Council (NWO-Vici 182.052 to H.v.A.).

## Author contributions

S.C.v.d.H. performed ChIP-qPCR, spot dilution tests, budding index, RT-qPCR, western blots, RNA-seq, suppressor screen, fork recovery assay, copy number experiments, flow cytometry analysis, and wrote the paper. L.K. performed the Repli-ID screens and analysis, ChIP-qPCR, and wrote the paper. A.B. performed the Repli-ID screens and ChIP-qPCR. S.K. performed RT-qPCR, ChIP-qPCR, flow cytometry analysis, and western blots. K.V. performed western blots and suppressor screen. P.P. and A.C. performed dNTP analysis. H.v.A. supervised the project and wrote the paper.

## Competing interests

The authors declare no competing interests.
