## [Transparent Peer Review file · Nature Communications]

Replication-Identifier links epigenetic and metabolic pathways to the replication stress response

Corresponding Author: Professor Haico van Attikum

Version 0:

Reviewer comments:

Reviewer #1

(Remarks to the Author)

Replication-Identifier links epigenetic and metabolic pathways to the replication stress response
Horst, Kollenstart, Batte, Keizer, Vreeken, Pandey, Chabes, Attikum

In this study, the authors describe a new strategy (termed Repli-ID) designed to identify regulators of replication stress responses genome-wide in budding yeast. They go on to characterize two hits, LGE1 and ROX1. Repli-ID is based on Epi-ID and involves crossing a strain library of knockouts to a strain containing a cassette with ARS404 plus the KanMX gene flanked by two barcodes (one immediately adjacent to origin, one 1.6 kb from origin as cassette is integrated into the HO locus), an epitope tagged Pol epsilon plus sld3-38A dbf4-4A to enable ARS404 to "escape" intra-S phase checkpoint regulation so it can fire prior to a hydroxyurea arrest instead of later in the cell cycle. Investigators perform ChIPseq for an epitope tagged Pol epsilon in the presence of DNA replication stress mediated by hydroxyurea after release from a G1 arrest. Overall, Repli-ID appears to be a powerful strategy to identify genes that influence DNA replication either directly or indirectly, the data is of high quality and analyzed appropriately. The study will be of high interest to others in the field, many of whom will be interested in adopting various versions of this approach. However, in its current form, the manuscript has a few issues, mostly at the level of more detail is needed in explanation of experiments plus care with "semantics" - the reader needs to be able to really understand the experimental designs so they can more easily draw their own conclusions from the data.

1. To facilitate readers' understanding of this study and how their findings relate to published studies, Fig. 1/legend or text should provide more detail on experimental method.
2. Throughout manuscript, clarification is needed as to what site is being assessed. Is the screen based on the efficiency of localization of Pol epsilon to the UPTag/0 kb from origin only? If so, Fig. 1a is displaying a compilation of mutants that potentially cause differences in either (non-catalytic activity dependent) role of Pol epsilon in functioning in the preloading complex, a role of Pol epsilon in converting the CMG to a competent form for origin firing, exchange of Pol alpha for Pol epsilon during initiation, or the presence of Pol epsilon in forks that initiated at a different origin, but move through this region that then "de-license" ARS404. Several of these roles of Pol epsilon occur prior to the existence of a fork that is "progressing" as emphasized by the authors and therefore (many?) identified mutants may instead be affecting "assembly" of a fork, not "stability" as emphasized by the others. But this is not made clear, and it is not clear if those possibilities have been fully considered by the authors when presenting/interpreting the data. Although it might be "simpler" to tell a story in terms of fork stability/progression, it would be helpful for the authors to also discuss their findings with these possibilities in mind.
3. To facilitate readers' understanding of this study and how their findings relates to published studies, Oligo table should note original names and source of design of oligonucleotides (e.g. that ARS 404 fw ARS404 rv correspond to Fred van Leeuwen's group's primers for the UPTag barcode). Distance from initiation site should be noted. This was a bugger to sort out.
4. Presumably ARS404's timing and efficiency was originally mapped by Raguhuraman plus Bonita Brewer and others. What is known about it's timing as well as the location of other origins that are documented, but not annotated (e.g. not in

SGD), and in the vicinity of ARS? e.g. Foss elife 88987.4 2024. Presumably changes in the Pol epsilon ChIP signals could alternately be influenced by mutants that affect adjacent origin firing, so it would be helpful for the authors to also discuss their findings in this context. If a mutant enables alternate origins to start firing, this may show up in Fig. 1a upper right quadrant.

5. Add the Pol2 myc tagged sld3-38A dbf4-4A strain to strain table.

6. p 6. "Moreover, 154 mutants showed reduced Pol levels at 40 minutes (Log_2 (fold change) < -1.25), but not at 80 minutes, possibly reflecting impaired fork progression, whereas 202 mutants showed this effect at 80 minutes (Log_2 (fold change) < -1.25), but not at 40 minutes, likely due to enhanced fork progression"

Please explain logic in more detail and which site is being referred to (0 kb versus distal site primers or both). The logic for the interpretation is not clear. For the first case, if authors are analyzing 0 kb from origin, this could instead reflect a timing delay in origin firing/ recruitment of Pol epsilon to origin in the mutant vs wt, independent of the presence of sld3-38A dbf4-4A and equal Pol epsilon at 80 min could simply reflect the origin fired later in the mutant and have nothing to do with "progression".

7. One thing to consider for "...a large fraction of mutants with impaired RNAPII transcription was found to have low levels of Pol epsilon.." Immediately upstream of ARS404 are the overlapping genes on opposite strands SSB1/YDL228C. If the transcription mutants are affecting SSB1 vs. YDL228C promoter usage (YDL228C's promoter overlaps the ARS), this has the potential to influence a competition between the TXN machinery and Pol epsilon (or chromatin accessibility facilitated by txn machinery) at the site assessed by the screen if 0 kb from origin was used. Testing the SSB1 vs. YDL228C transcript ratios in the transcription mutants should clarify this possibility.

8. Flow cytometry or additional budding index data should be provided to confirm discussed mutant (e.g. highlighted in Fig. 1) phenotypes do not reflect a defect in release from alpha factor G1 arrest, which would indirectly lead to reduced origin firing/association of Pol epsilon at time of cell harvest, but be unrelated to DNA replication per se. This appears to be the case for the rox1 line of inquiry in Fig. 5, not (noted in end of results section) a "new factor that impacts Pol epsilon progression". An alternate interpretation is the reduction of Pol epsilon ChIP signal in Fig 5e is probably primarily reflecting that only a subset of cells released from the G1 arrest during the experiment. Would Pol epsilon normally be recruited to origins in cells still in G1? This is very different than "ensuring proper progression and stability of Pol epsilon", which implies a fully assembled fork has left an origin after START. Care with/Change of phrasing is recommended. In the long run, outside of the scope of this manuscript, it may be useful to include a secondary screen with Repli-ID to differentiate between factors that influence "release from G1 arrest" and those that act at the level of replication.

9. For the sup. figures in excel, more detail/a key is need for worksheets. e.g. Sup Fig S3c. Readers would be helped with a definition of the unit for the unitless numbers. The number of significant digits shown should reflect the whether the assay can really be accurately measured to four significant digits. Or not.

Reviewer #2

(Remarks to the Author)

In this article, SC van der Horst, L Kollenstart and colleagues present a new technical approach, termed Repli-ID, to screen for mutants affecting the stabilization of DNA polymerase epsilon on chromatin around replication origins after depleting dNTP pools with hydroxyurea. This technique is based on Pol2 ChIP at a barcoded origin of replication followed by next-generation sequencing. They identified mutants affecting the binding of Pol2 to chromatin upon HU treatment and undertook the mechanistic characterization of two candidates, Lge1 and Rox1. They conclude that Lge1 and Rox1 are indirectly required for the efficient activation of the S-phase checkpoint through epigenetic and metabolic pathways, respectively, and that Repli-ID could be used in future studies to better understand the response to replication stress.

Overall, the reviewers felt that, although this technique appears of interest for the discovery of new players in the replication stress response, the manuscript lacks clarity in many aspects. The introductory and discussion parts need a better inclusion of already published data. The choices of the technical approach parameters are not well justified or not well described. There are flaws in the data analysis and the conclusions are not always supported by the data.

Thus, the reviewers think that a major revision is required and have the following recommendations to improve the manuscript and promote its publication:

(There is no page numbering in the manuscript, rendering the review process very tedious).

Global improvement of the introduction part:

1. Better describe the functions of the DDR.

What is the purpose of increasing the transcription of RNR genes? Are other genes' transcription induced? Why suppressing late origin firing? What means "late"? Other functions?

Better describe the mechanistic activation of the DDR.

How ssDNA stretches are exposed? Current models?

Why the authors cite REF#5 Cobb et al., 2003 if they do not describe the role of Sgs1 in facilitating the activation of Rad53?

2. The authors claim in the abstract that Repli-ID has been elaborated to identify new regulators of the replication stress

response. However, they describe in the introduction that Repli-ID is meant to measure replication fork progression or stability, which is more accurate considering that they immunoprecipitated the DNA polymerase epsilon. To identify new regulators of the DDR, one could expect a more direct ChIP readout as the recruitment to chromatin of Mec1, Ddc2, or Rad53. Please correct the abstract.

3. Concerning the choice to ChIP Pol2 around replication origins under HU, there is a complete lack of justification of this choice and a lack of description of the known literature. For example, REF#5, Katou et al., 2003, De Piccoli et al., 2012 and others.

Interpretation of Pol2 ChIP-qPCR data under HU has been under debate for years and Pol2 ChIP-seq data have identified disparities between the binding pattern of Pol2 around very early replication origins like ARS607 used in this work and other origins (De Piccoli et al., 2012). These data may prevent the generalization of the conclusions made in this manuscript.

Results section, screen validation and results:

4. The authors do not justify why they choose to study the binding of Pol2 near ARS404, which, as such, seems irrelevant since this replication origin is inhibited under HU. They describe this result in Supplementary Figure S1b, which also describe the binding of Pol2 at the early origin ARS607. However, the data of Pol2 ChIP at ARS607 is not consistent with what is expected (progression along time from ARS607 to ARS607 + 4kb) and with similar data presented in Figures 3 and 4.

All over the manuscript, the reviewers noticed huge variations in the “fold increase” of Pol2 ChIP in wild-type cells between experiments. They believe that the double normalization method used for ChIP experiments (first normalization over the beads only signal (Figure 2) or IgG signal (Supplementary Figure S1) and second normalization over a non-replicated region ARS607+14kb) is critically impacting the enrichment of the proteins of interest, thus affecting conclusions. Raw IP/input ratios (without normalization) may be a better way to present their data or use only one normalization parameter.

Moreover, Pol2 ChIP data analysis would benefit a lot from the parallel analysis of the copy number increase by using the qPCR data of input samples after release in S phase normalized by the qPCR data of input samples in G1 (Time 0). This would allow a parallel assessment of the progression of replications forks under the experimental conditions used.

5. The use of ARS404 in Repli-ID forces the authors to perform the screen in the *sld3-38A dbf4-4A* genetic background. It is not clear why they decided to overexpress the mutant alleles in the presence of wild-type alleles rather than using a strain only expressing the mutant alleles constitutively. It seems to complicate the experimental procedure by working in a culture medium with raffinose supplemented with galactose to overexpress the mutants. Please explain.

Moreover, they authors should indicate that DNA replication in the *sld3-38A dbf4-4A* genetic background has detrimental consequences: global lower efficiency of origin firing, spontaneous activation of the DDR (gamma-H2A signal), increased DNA damage in response to HU (Rad52 foci formation), increase in topological stress. The authors should cite the relevant literature, for example, Morafraille et al., 2019, and discuss the implications of using the *sld3-38A dbf4-4A* mutant background in Repli-ID screens.

6. The reviewers understood that the authors justify the choice of using the 40min and 80min time points in HU for the screen because they observed maximum Pol2 ChIP enrichment values at the replication origin at 40min and away from the origin at 80min (Supplementary Figure S1c). However, they claim in their conclusion that the 80min time point is ideal for sample collection during Repli-ID screens because Pol2 ChIP enrichment values “decline” at ARS404 at 80min (Supplementary Figure S1d). Please clarify your justification.

Moreover, the data should be better described. For example, Pol2 signals accumulate at the origin at early time points, then progressively decrease while increasing away from the origin at later time points, indicating replication fork progression.

7. The authors describe that they built the Repli-ID library by crossing the Pol2-9Myc *sld3-38A dbf4-4A* with the barcode mutant library (Figure 1a) but this is not depicted in the Figure 1a.

The distance from the ARS404 at which the barcode sequence is inserted is not indicated. Thus, it is difficult to understand if Repli-ID is assessing the binding of Pol2 at the replication origin or away from the origin. The fact that Pol2 hardly accumulated away from ARS404 in the experimental conditions used (low Pol2 enrichment at ARS404+1.6 kb) questions the ability of the Repli-ID technique to assess fork progression.

8. Having performed the screen, the authors identified 423 deletion mutants in which Pol2 levels were decreased a both time points. They conclude that these mutants are likely affected in fork stability/progression under HU stress. They cannot exclude that these mutants can be affected in replication initiation independently of replication stress and should test, at least in some candidates, if Pol2 binding to the barcode sequence is affected in the absence of HU.

They also identified 128 mutants in which Pol2 levels were increased a both time points. They have not interpreted these data.

They also identified 154 mutants in which Pol2 levels were decreased at 40min but not at 80min, suggesting an impaired fork progression, and 202 mutants in which Pol2 levels were not decreased at 40min but decreased at 80min, suggesting enhanced fork progression.

The authors should discuss further these data and propose alternative interpretations.

For instance, decreased signals at a replication origin at both time points are reminiscent of what has been observed in *mec1* and *rad53* mutants, defective in checkpoint activation. Pol2 signals were detected only 15 kb away from some very early replication origins in these mutants (De Piccoli et al., 2012), reflecting replication fork progression before dNTP

depletion.

In light of the checkpoint activation defect observed in both *lge1Δ* and *rox1Δ* mutants, the authors should reconsider their data interpretation.

9. The strongest hits showing Pol2 decrease at ARS404 are mutants of factors involved in DNA replication (Pol32, Mgs1 and Pif1) but none of them has been shown to impact the stability of Pol2 under HU. On the contrary, the *sgs1Δ* mutant did not come out as a strong hit but has already been shown to decrease Pol2 levels at replication origins under HU by ChIP-qPCR (Cobb et al., 2003). This would have validated the new findings of the screen.

The authors should propose an explanation for the absence of *sgs1Δ* in their strongest hits, although it is present in the Repli-ID library and test it individually by ChIP-qPCR at ARS404 and ARS607 in the *sld3-38A dbf4-4A* genetic background.

10. Among the candidate mutants showing reduced Pol2 levels at both time points, the authors curated the results by assessing Pol2 levels in individual mutants by ChIP-qPCR. The authors should clearly describe that the first step of curation was performed in the *sld3-38A dbf4-4A* background (Figure 2c) and the second step in mutants generated de novo in wild-type W303 background (Supplementary Figure S2c). The latter panel S2c should be included in the main Figure 2. In this experiment, the authors must justify why they decided to assess Pol2 levels at the time point 20min and not 40min or 80min as in the Repli-ID screen. They also have to explain to the readers why they decided to include ARS305 and ARS501.

11. The authors observed that *rox1Δ* and *lge1Δ* mutants are sensitive to high doses of HU and use this result to justify further characterization of these mutants. Exposure to high doses of HU not only depletes dNTPs but also leads to the production of reactive oxygen species (ROS; Huang et al., 2016), which can explain the mutants' sensitivity independently of a replication defect. The authors should thus test HU sensitivity in the presence of a ROS scavenger (N-acetyl-L-cysteine 10mM or alpha-ketoglutarate 5mM) to demonstrate that *rox1Δ* and *lge1Δ* mutants have defects related to the depletion of dNTP pools and the stalling of replication forks by HU.

Results section, *lge1Δ* characterization:

12. The authors state that a role for Lge1 in the response to replication stress has not been described yet. This is maybe true, but they should also clearly indicate that a role for Bre1, co-factor of Lge1 for histone H2B ubiquitylation, has been deeply investigated in response to HU-induced replication stress. They cited the relevant literature (REF#24) without recalling the main phenotypes caused by the absence of Bre1 or the ubiquitylation of histone H2B (*htb-K123R* mutant) on DNA replication.

The authors should acknowledge these previous findings and interpret their results in light of these previous findings, which also showed that the absence of H2B ubiquitylation led to a checkpoint activation defect.

13. The authors found that *lge1Δ* and *bre1Δ* mutants have increased dNTP levels but argue that this cannot impair replication progression and explain the low levels of Pol2 at replication origins under HU. However, mutants with high dNTP levels have been described to progress further away from origins than wild-type cells in HU because of a delayed exhaustion of dNTPs. A consequence of this extended progression could result in the dilution of the ChIP signal, explaining why Pol2 signals are lower in *lge1Δ* than in wild-type cells. This would also indicate that ChIP-qPCR values should not be normalized by ARS607+14kb.

Here again, exploiting the DNA copy number increase data from the ChIP data should indicate the degree of replication progression independently of Pol2 binding to chromatin. This readout is much more informative than the budding index.

14. The loss of Lge1 or Bre1 resulted in strongly decreased levels of H2Bub in G1 and S phase +HU. The authors concluded that Lge1- and Bre1-dependent H2B ubiquitylation facilitate DNA replication at HU-stalled forks. This conclusion is not justified by the data. H2Bub levels are high in all regions close to ARS607 without DNA replication or without HU in G1-arrested wild-type cells, likely due to the presence of transcribed genes. After the release of cells in S phase +HU, the pattern of H2Bub does not change along time, whereas Pol2 progressively travel from ARS607 to ARS607+8kb. Thus, the progression of DNA replication under HU cannot be linked to H2B ubiquitylation.

In *lge1Δ* cells, H2Bub levels are globally lost (Figures 3c, 3d and Supplementary Figure S3a) in G1-arrested cells and S phase +HU. Whether the absence of H2Bub impacts the stability of Pol2 under HU is just a correlation. It could be confirmed by measuring Pol2 levels in *bre1Δ* or *htb-K123R*.

15. The efficiency of Bre1 ChIP is doubtful (Figure 3e). Normalized fold increase values are very low (around 0.5 above the background) in wild-type cells. A positive control validating the ChIP efficiency would have been welcome. The figure legend indicates that normalization was done on the beads only signal but source data indicate TELVI (?). Raw IP/input ratios may be a better indicator of the efficiency of Bre1 ChIP.

The authors propose that the loss of H2Bub in *lge1Δ* could be due to an upregulated recruitment of Ubp8. However, they do not test this hypothesis experimentally in *ubp8Δ* cells. This may be worth mentioning only in the discussion.

16. The authors study the kinetics of Rad53 phosphorylation as an indicator of the checkpoint response (Figure 3f). The source data indicate that the western blot picture has been cut to select time points of interest, which may explain the presence of dotted lines on the picture. What means "X" on the source picture? How many times this experiment has been performed?

These data indicate that *lge1Δ* cells have a delayed checkpoint activation, as previously observed in *htb-K123R* mutant (REF#24). As stated by the authors, efficient checkpoint activation is critical for the recovery of cells from HU-induced replication stalling and tested it in *lge1Δ* and *bre1Δ* mutants. However, their data did not indicate that the checkpoint

activation was not “efficient” but “delayed”.

Timely activation of the S-phase checkpoint is critical for the inhibition of late origin firing. The authors should therefore investigate if late origins are fired in the *Ige1Δ* mutant as this may be related to the loss of viability. They observed that Pol2 was not recruited at ARS501 in *Ige1Δ* (Supplementary Figure 2c) but late origins may fire later than 20min after release in S phase+HU.

They also observed an impairment in recovery from HU arrest in *Ige1Δ* and *bre1Δ* mutants to the same extent as in *mrc1Δ* (not described in the results). *mrc1Δ* mutant is a canonical example of a checkpoint deficient mutant that has lost the ability to inhibit late origin firing in HU.

Results section, *rox1Δ* characterization:

17. As indicated earlier, HU exposure also leads to ROS and *rox1Δ* phenotypes may be more linked to ROS accumulation than to dNTP depletion. Therefore, *rox1Δ* and *rox1Δ sur2Δ* sensitivity to HU should be re-evaluated in the presence of ROS scavengers.

18. When looking at RNR genes expression, the authors should indicate that the increase of RNR3 expression is expected in response to HU because it is a checkpoint response-induced gene.

19. In Figure 5a; the authors test how the addition of ceramide impacts the sensitivity of *rox1Δ* and *sur2Δ* mutants to HU. The description of the results is incomplete. It is worth mentioning that ceramide addition increased the sensitivity to HU of wild-type and *rox1Δ* cells, before describing that ceramide addition prevented the suppression of *rox1Δ* HU sensitivity by *sur2Δ*. Ceramides activate PP2A through *Irc21* (REF#42). Thus, it would be worth checking if the addition of ceramides in the absence of *IRC21* still prevents the suppression and if *irc21Δ* could suppress the HU sensitivity and the defect in Rad53 phosphorylation in response to HU of *rox1Δ* cells.

20. In Figure 5c, the authors use the budding index as an indicator of the G1/S transition. It may not be a good indicator because the lipid metabolism is affected in *rox1Δ* cells and HU could also lead to lipids oxidation. Both could impede a normal bud growth independently of the G1/S transition.

21. When assessing Pol2 levels in *rox1Δ* and *sur2Δ* mutants (Figure 5e), the authors chose the time point 60min without any justification. Enrichment values are very different from those at the same positions in wild-type cells at 60min (Figure 3b), questioning the validity of the data.

Moreover, Pol2 levels are globally higher in *sur2Δ* than in wild-type cells, raising the possibility that the rescue in *rox1Δ sur2Δ* may be a compensatory effect.

Reviewer #3

(Remarks to the Author)

Reviewer #4

(Remarks to the Author)

Van der Horst et al. 2024

The authors establish a method they call Repli-ID (derived from Epi-ID) to enable high-throughput screening of barcoded mutant collections for genes affecting replication fork stability. The method uses ChIP of Pol e, the leading strand replicative DNA polymerase, near a barcoded origin as a proxy for replication fork stability in hydroxyurea (HU), which depletes dNTPs and slows/stalls DNA Polymerase. Relative dissociation (or accumulation) of Pol e from replicating chromatin is inferred to indicate some defect in fork stabilization/checkpoint signaling. Enriched or diminished number of specific barcodes indicates potential gene(s) involved.

While the overall concept of analyzing Pol e stability on DNA as a measure of fork stability in different mutants makes sense, a strangely baroque and incompletely explained aspect of this screen is the use of ARS404 as the barcoded locus for analysis. Presumably this is to use an existing barcoder strain library, but ARS404 doesn't fire in HU in this context (late/dormant), so they overexpress *Dbf4* and *Sld3* mutants to bypass the intra-S checkpoint. This raises some concern given their intended purpose to identify regulators of fork stability, which is dependent on the intra-S checkpoint. Bypassing the checkpoint can create replication stress, which might sensitize the assay and lead to identification of genes with indirect effects. Why not barcode ARS607?

Nevertheless, the results of a screen of ~2900 (or 4500?) knock-out strains resulted in identification of 423 strains with reduced Pol e association and 128 strains with increased signal, though stringency appears low. The 423 genes represented in the strains with reduced Pol e are said to be highly enriched for certain GO annotations like DNA repair, but statistical analysis appears absent. It also seems that certain categories not expected to be related to fork stability like “transmembrane transport” are similarly enriched so the significance of this analysis as validation is not established. Most replication genes are essential and thus are not represented in the knock-out collection; however, three non-essential replication/repair factors were identified, as might be anticipated, suggesting at least nominal functionality of the screen.

They validated the accuracy of the screen by re-testing 12 knockout strains by Pol e ChIP and most (10) were reproducible (i.e., reduced Pol e association), supporting the ability of the screen to detect reduced Pol e in HU. It is not clear how these 12 were selected for further testing, which is important for validation and inferring the overall usefulness of the screen. It is implied that these have been linked in some way to replication/genome stability but the connection is vague and possibly indirect. If this is true, the remaining ~400 genes have no previous link to replication, and while these might represent bona fide new regulators, no further analysis of those is performed. Is this correct? Why?

Moreover, reconstruction of six of the 12 knockout strains in a different strain background only validated three (LGE1, ROX1, and NSR1), suggesting significant strain-dependent effects or questionable strain identities in the knockout collection. Because *lge1Δ* and *rox1Δ* were HU-sensitive, they focused further analysis on these two.

Lge1 is a known co-factor of *Bre1* that together are required for ubiquitylation of H2B, which is reported to play a role in establishment of cohesion during DNA replication (Zhang et al 2017 not cited) and premeiotic DNA synthesis (Jordan et al 2007 not cited). Whereas H2Bub is known to be required for recovery from replication stress, *Lge1* has not been directly implicated previously, but would seemingly be strongly implicated by association.

Analysis of *lge1Δ* cells by ChIP of Pol e shows a striking decrease in Pol e loading onto ARS607, suggesting an initiation defect. There is no analysis without HU so there's no distinction here between an initiation defect versus fork stability defect due to HU. I think analysis without HU is required. Also, they should demonstrate that reduced replication is not due to a defect in licensing by analyzing MCM levels.

In WT cells H2Bub is present on chromatin prior to replication (and generally higher at origin distal regions) and changes little during S-phase in HU. In *lge1Δ*, levels of H2Bub are lower across the region, throughout the time course; similar in *bre1Δ* and *bre1Δ lge1Δ*. While deletion of LGE1 reduces H2Bub and reduces Pol e loading or stability, there is no functional connection between these effects. A simple interpretation is that H2Bub present at origin is required for efficient initiation, but this is not demonstrated. Elimination of *Lge1* and/or *Bre1* function through a degron approach may enable more rigorous test of *Lge1*-*Bre1* requirement in replication. Analysis of bulk replication by flow cytometry or other approaches is sorely lacking.

Though there is no obvious change in H2Bub levels linked to replication across the region during the time-course. *Bre1* association with the origin region appears to increase (~2-fold) during replication, possibly even anticipating the increased Pol e association that occurs during the time-course, which seems inconsistent with its recruitment to stalled forks. This increase in *Bre1* signal is partially dependent on LGE1. If this increased association is functionally significant, it remains unclear whether this is a normal or replication stress response.

They show reduced Rad53 phosphorylation/activation. Given the possible evidence of reduced replication initiation, the reduced Rad53 activation may simply reflect lower initiation levels, rather than a direct role in replication stress signaling. Viability assays show sensitivity to HU in the *lge1Δ*, *bre1Δ* and *bre1Δ lge1Δ* cells; mechanistic significance is unclear.

Overall, the identification of LGE1 is seen as validation of the screen given prior evidence of its role in replication as well as the well-established role of *Bre1* in replication fork stability; however, the additional analysis of *lge1Δ* does not provide compelling new evidence of a role in fork stabilization per se. The conclusions are not inconsistent with the data nor are they rigorously supported by the data.

They move on to analysis of ROX1 and confirm that deletion of ROX1 reduces Pol e association with origin and flanking chromatin. Because ROX1 regulates RNR gene expression, dNTP levels were measured and determined to be unchanged (significance test?), suggesting a different mechanism.

They analyzed RNA levels in *rox1Δ* cells to identify genes whose misexpression might explain the replication phenotype; however, no enrichment for genes annotated as DNA replication or repair were identified. They sought suppressors of the HU-sensitivity resulting from deletion of ROX1 by crossing *rox1Δ* with the gene knockout collection of strains. They identify 431 genes that suppressed *rox1Δ*, which seems like a huge number (the writing is confusing: "431 gene deletions were identified that suppressed growth on HU in *rox1Δ* (Fold change > 1.5)" I think they mean suppressed the lack of growth in *rox1Δ*). They overlap these 431 with genes identified as upregulated by the RNA-seq analysis and find (only?) three genes: SUR2, TRX2, and DOG1. Only SUR2 deletion suppresses *rox1Δ* in W303 background, so focus moves to SUR2. How can there be so many suppressors of *rox1Δ* HU-sensitivity?

SUR2 regulates ceramide production, which in turn can regulate PP2 phosphatase activity, which is a cell cycle regulator. They show that presence of ceramide can prevent suppression of *rox1Δ* by SUR2 deletion, supporting the idea that increased ceramide levels in *rox1Δ* cells produced by upregulated SUR2 lead to the HU-sensitivity and hence, is suppressed by SUR2 deletion. Given the delay in budding (reflecting the G1-S transition) in *rox1Δ* cells, it appears that *Rox1* exerts its effect on replication indirectly by modulating PP2 activity. This may act through G1-S control or intra-S control. Both mechanisms may explain the reduced Pol e loading in *rox1Δ* cells, though this is not further examined. Do any other genes identified in the RNA-seq or suppressor screen (e.g.: in the GO categories Response to chemical and lipid metabolic process) potentially further support the involvement of the ceramide pathway described here?

Overall, the impacts of Repli-ID here are modest. Out of hundreds of candidates showing decreased Pol e association, only a few appeared worthy of follow up. *Lge1* was already implicated through *Bre1* so appears more confirmatory than novel.

Rox1 involvement is more novel, but the real insight here came from the suppressor screen based on HU sensitivity. Perhaps the identification of other candidate genes by this screen will provide motivation to further examine and possibly reveal some new insights as with Rox1, but it's not clear that there will be many more strong candidates in their list, which as far as I can tell is not provided, but should be the most valuable part of this study.

Significant revision is required to support the potential impact for the field and other researchers. Conclusions from the data need to consider alternative explanations and caveats.

Version 1:

Reviewer comments:

Reviewer #1

(Remarks to the Author)

In this study, the authors describe a new strategy (termed Repli-ID) designed to identify regulators of replication stress responses genome-wide in budding yeast. They go on to characterize two hits, LGE1 and ROX1. Repli-ID is based on Epi-ID and involves crossing a strain library of knockouts to a strain containing a cassette with ARS404 plus the KanMX gene flanked by two barcodes (one immediately adjacent to origin, one 1.6 kb from origin as cassette is integrated into the HO locus), an epitope tagged Pol epsilon plus sld3-38A dbf4-4A to enable ARS404 to "escape" intra-S phase checkpoint regulation so it can fire prior to a hydroxyurea arrest instead of later in the cell cycle. Investigators perform ChIPseq for an epitope tagged Pol epsilon in the presence of DNA replication stress mediated by hydroxyurea after release from a G1 arrest. Overall, Repli-ID appears to be a powerful strategy to identify genes that influence DNA replication either directly or indirectly, the data is of high quality and analyzed appropriately. The study will be of high interest to others in the field, many of whom will be interested in adopting various versions of this approach.

In this revision, the authors have greatly clarified how experiments were conducted, included several followup control/validation experiments, and expanded discussion and commentary on and acknowledgement of potential alternate interpretations to the data they present. This will greatly facilitate readers' understanding of the study and how the reported findings relate to published studies.

Enough detail is now provided to confirm methodology is sound and enough detail has been provided in the methods for the work to be reproduced.

Reviewer #2

(Remarks to the Author)

I acknowledge that the authors have addressed all the points raised and have made significant improvements to the manuscript.

Nevertheless, some points are incompletely addressed or require further explanation.

Point 14. Reviewers #2, #3 and #4 pointed out that H2Bub levels in the vicinity of ARS607 in wild-type cells do not seem to change before and after replication stress (Figure S3G). This is not clearly stated in the Results section, where the authors only mention that H2Bub levels were equally decreased in G1 and S+HU in lge1 Δ and bre1 Δ mutants compared to wild-type cells.

These results do not suggest that H2Bub is part of the replication stress response that stabilizes Pol epsilon. However, the authors claim in their discussion (page 16) that their mechanistic studies show that Lge1 directly drives H2Bub at stalled forks and enhances Bre1 activity.

This conclusion is not justified by the data because there is no difference in H2Bub levels between G1 and S+HU. An alternative interpretation is needed. For example, the authors do not mention in the discussion the delayed checkpoint activation in lge1 Δ and bre1 Δ mutants, which could affect the stability of Pol epsilon (through loss of chromatin remodeling?).

Point 16. Given the potential importance of the checkpoint defect in lge1 Δ and bre1 Δ mutants, these data need to be consolidated.

The authors have stated that they cropped the original blot, justifying the presence of dotted lines in Figure 4B. This needs to be indicated in the figure legend.

In addition, the authors provided the reviewer with the replicate of this experiment (Reviewer_only_Figure 4). This blot is inconclusive (2 bands at time 0, no clear increase in phospho-mobility shift observed) and needs to be repeated.

Point 15. The Bre1 ChIP efficiency near ARS607 in response to replication stress is still an issue. The authors have included a positive control (PMA1 gene) in Figure S4B, but this control was not included in response to HU, where a very weak recruitment of Bre1 in wild-type cells and the partial loss in lge1 Δ were observed, but questionable (Figure 4A).

Point 19. As the authors understood from Ferrari et al. (Mol Cell 2017), Irc21 promotes the production of ceramides and consequently the activation of PP2A.

If the defect of rox1 Δ in HU is due to the overexpression of SUR2 and the overproduction of ceramides, as proposed by the authors, this defect should be suppressed by the loss of IRC21.

This is not what the authors observed in their new results in the rox1 Δ irc21 Δ double mutant (Reviewer_only_Figure 5). In addition, the same figure shows that despite the overall increase in HU sensitivity, sur2 Δ still suppresses rox1 Δ HU sensitivity in the presence of cell-permeable C2 ceramide.

Finally, the results shown in Figure S6E hardly show that the addition of ceramides decreases Rad53 phosphorylation in wild-type cells (poor resolution of the phospho-mobility shift and more protein loading in the +HU-only condition) and do not confirm the results shown in Figure 6B. The authors did not describe or interpret the effect of ceramide addition in *rox1Δ* cells. It appears that ceramide addition is still able to decrease Rad53 phosphorylation in *rox1Δ* cells. Therefore, we believe that the authors need to find an alternative interpretation of their data.

Point 9. The authors have provided Pol2 ChIP data in the *sgs1Δ* mutant in the *sld3 dfb4* background (Reviewer_only_Figure 3) and show that Pol2 levels are globally decreased near ARS404 and ARS607. Since *sgs1Δ* has already been described to decrease Pol2 levels in HU, these new data validate their approach and should be included in the manuscript.

Reviewer #3

(Remarks to the Author)

Reviewer #4

(Remarks to the Author)

The authors have done a good job at responding to my critiques with new experiments and analysis. But I disagree strongly with their interpretation regarding the effect on initiation versus fork progression. I think they might be more circumspect about this point.

They state (twice in the rebuttal): "The new Mcm4 ChIP experiments (see previous point and new results in Fig. 3c) did not show a replication initiation/licensing defect in *Ige1Δ*. Moreover, S-phase entry, as monitored by budding index analysis, was normal (see Supplementary Fig. S3c)." They also state: "This is supported by new copy number analysis, showing that, although DNA synthesis is initiated, it is reduced in *Ige1Δ* (see new results in Supplementary Fig. S3d)." Licensing and initiation are two different things. Budding index does not indicate S-phase entry only progression through START. I still think their data point equally strongly to an initiation defect rather than fork stability as they observe reduced Pol epsilon loading even without HU and a delay in any copy number increase at the origin. These data are completely consistent with an initiation defect. Whereas there's progression of the MCMs in the WT, this doesn't appear to happen in *Ige1Δ*. Given the copy number data it would seem that *Ige1Δ* cells are replicating very poorly. I think DNA content analysis is still sorely missing; presumably more analysis of the total DNA for the copy number analysis would address this. The Discussion in the text (p. 8-9) is similarly problematic, suggesting that proper licensing mean initiation is normal.

It is indeed compelling that Bre1 is only seen to associate with the origin in the presence of HU but how do the authors explain that effects on H2BUB are already apparent at T=0 (e.g. Fig. S4b)?

I also think that all ChIP-seq datasets should show analysis of how well experimental replicates correlate (R2) genome-wide as supplementary information.

Version 2:

Reviewer comments:

Reviewer #2

(Remarks to the Author)

The authors have now included all the controls and corrections we had suggested, making their manuscript acceptable for publication.

We congratulate the authors for their hard work, which we felt was necessary to improve the quality of this study.

Reviewer #3

(Remarks to the Author)

Reviewer #4

(Remarks to the Author)

The authors have added new analysis of copy-number and flow cytometry to address the criticism about initiation versus fork stability, but they have performed the experiment in a way that continues to obscure the issue by including HU, and

unusually 10mM HU, so this remains ambiguous. At least they acknowledge initiation as a possible defect though I think the statement in the abstract: “ Mechanistically, we show that Lge1 affects replication initiation and fork stability by promoting Bre1-dependent H2B mono-ubiquitylation.” is an overstatement. Their results suggest that Lge1 affects replication initiation and/or fork stability by...”

REVIEWER COMMENTS

We thank the reviewers for their constructive comments and useful suggestions. Based on these we have adapted the manuscript textually, performed several additional experiments and analyses, and included multiple new figure panels. These changes have solidified the main conclusions. We have addressed all comments point-by-point below.

Reviewer #1 (Remarks to the Author):

Replication-IDentifier links epigenetic and metabolic pathways to the replication stress response

Horst, Kollenstart, Batte, Keizer, Vreeken, Pandey, Chabes, Attikum

In this study, the authors describe a new strategy (termed Repli-ID) designed to identify regulators of replication stress responses genome-wide in budding yeast. They go on to characterize two hits, LGE1 and ROX1. Repli-ID is based on Epi-ID and involves crossing a strain library of knockouts to a strain containing a cassette with ARS404 plus the KanMX gene flanked by two barcodes (one immediately adjacent to origin, one 1.6 kb from origin as cassette is integrated into the HO locus), an epitope tagged Pol epsilon plus sld3-38A dbf4-4A to enable ARS404 to "escape" intra-S phase checkpoint regulation so it can fire prior to a hydroxyurea arrest instead of later in the cell cycle. Investigators perform ChIPseq for an epitope tagged Pol epsilon in the presence of DNA replication stress mediated by hydroxyurea after release from a G1 arrest. Overall, Repli-ID appears to be a powerful strategy to identify genes that influence DNA replication either directly or indirectly, the data is of high quality and analyzed appropriately. The study will be of high interest to others in the field, many of whom will be interested in adopting various versions of this approach. However, in its current form, the manuscript has a few issues, mostly at the level of more detail is needed in explanation of experiments plus care with "semantics" - the reader needs to be able to really understand the experimental designs so they can more easily draw their own conclusions from the data.

1. To facilitate readers' understanding of this study and how their findings relate to published studies, Fig. 1/legend or text should provide more detail on experimental method.

We thank the reviewer for pointing out the limited explanation of the experimental design. We have provided more detail on the experimental design by adapting Figure 1, expanding the legend of Figure 1, adding a description to the first paragraph of the sub-section "Towards Repli-ID: a new method to identify regulators of replication fork stability/progression" of the Results section, and clarifying further experimental details in the "Repli-ID screen" section of the Methods.

2. Throughout manuscript, clarification is needed as to what site is being assessed. Is the screen based on the efficiency of localization of Pol epsilon to the UPTag/0 kb from origin only? If so, Fig. 1a is displaying a compilation of mutants that potentially cause differences in either (non-catalytic activity dependent) role of Pol epsilon in functioning in the preloading complex, a role of Pol epsilon in converting the CMG to a competent form for origin firing, exchange of Pol alpha for Pol epsilon during initiation, or the presence of Pol epsilon in forks

that initiated at a different origin, but move through this region that then "de-license" ARS404. Several of these roles of Pol epsilon occur prior to the existence of a fork that is "progressing" as emphasized by the authors and therefore (many?) identified mutants may instead be affecting "assembly" of a fork, not "stability" as emphasized by the others. But this is not made clear, and it is not clear if those possibilities have been fully considered by the authors when presenting/interpreting the data. Although it might be "simpler" to tell a story in terms of fork stability/progression, it would be helpful for the authors to also discuss their findings with these possibilities in mind.

We apologize for the confusion about the site that was assessed during Repli-ID. We used the previously described DownTag, which is a 20 bp barcode that flanks a KanMX selection marker in a cassette that was used to replace the *HO* gene (Yan et al., Nat Methods, 2008). ARS404 is located upstream of the *HO* gene. Consequently, following integration of the DownTag-KanMX cassette at the *HO* gene, the DownTag is located 53 bp downstream of ARS404. We have indicated this in a new version of Figure 1 (panel a).

We acknowledge that Pol2 does not only affect fork stability and progression, but also impacts origin loading and licensing. We have adapted our manuscript to include these possibilities. We mention this in the first paragraph of the subsection "Repli-ID identifies known regulators of replication fork stability/progression" by adding "likely due to impaired replication fork stability/progression and/or reduced S-phase entry" in the Results section and origin loading, licensing, S-phase entry is discussed in the second paragraph of the Discussion section as possible explanations for Pol2 enrichment defect.

Finally, we acknowledge that replication forks that initiated at a different origin may move through the ARS404-containing region and "de-license" this origin. We refer to our response to point 4 of this reviewer (see below), in which we elaborate more extensively on this possible phenomenon.

3. To facilitate readers' understanding of this study and how their findings relates to published studies, Oligo table should note original names and source of design of oligonucleotides (e.g. that ARS 404 fw ARS404 rv correspond to Fred van Leeuwen's group's primers for the UPTag barcode). Distance from initiation site should be noted. This was a bugger to sort out.

We apologize this was not directly clear from our manuscript, especially as the ARS404 fw and ARS404 rv correspond to the primers from Fred van Leeuwen's group near DownTag barcode directly adjacent to the origin (HOtermQfw and HOtermQrv). We adapted the names of these primers in Supplementary Table S2. The distance from ARS404 to the DownTag barcode is indicated in the new version of Figure 1 (panel a).

4. Presumably ARS404's timing and efficiency was originally mapped by Raguhuraman plus Bonita Brewer and others. What is known about it's timing as well as the location of other origins that are documented, but not annotated (e.g. not in SGD), and in the vicinity of ARS? e.g. Foss elife 88987.4 2024. Presumably changes in the Pol epsilon ChiP signals could alternately be influenced by mutants that affect adjacent origin firing, so it would be helpful for the authors to also discuss their findings in this context. If a mutant enables alternate origins to start firing, this may show up in Fig. 1a upper right quadrant.

ARS404 is known as a late firing origin (Crabbé et al, Nat. Struct. Mol. Biol., 2010), which fires after 60 minutes in checkpoint-deficient *rad53-11* cells, but not in wild-type cells. The DNA

replication origin database (OriDB) published by Siow et al. (Nucleic Acids Res., 2012) reports different replication times for ARS404 in conditions without HU: 36.5 minutes (according to Raghuraman et al, *Science*, 2001), 27.4 minutes (according to Yabuki et al., *Genes Cells*, 2002), and 17.5 minutes (according to Alvino et al., *Mol. Cell. Biol.*, 2007). As a comparison, for the early origin ARS607 the reported times are 12.8, 20.4 and 10 minutes, respectively.

Finally, the upstream and downstream ARS403 and ARS405 are around 30 and 39 kb away from ARS404, respectively. Fork speed, which under normal conditions has been estimated at around 2 kb/minute (Theulot et al., *Nat. Commun.*, 2022), is around 10-fold reduced in the presence of HU (around 0.2 kb/minute) (Poli et al., *EMBO J.*, 2012). Since we monitored Pol2 levels at 40 and 80 minute after release in S-phase in the presence of HU, the forks could have travelled maximally 16 kb. Based on this, we believe it is unlikely that firing of the adjacent origins ARS403 and ARS405 delicensed ARS404 during the Repli-ID screen. However, while our study was ongoing, Foss et al. (eLife, 2024) reported the identification of 1600 origins, including several sites outside of known origins (referred to as alternate origins), from which replication can be initiated. We found 15 alternate origins in the region between ARS403 and ARS405 (see Reviewer only Figure 1). We do not know if replication initiated from these alternate origins in mutants identified in our screen, which overexpressed mutant forms of Sld3 and Dbf4 (*sld3-38A dbf4-4A*). However, five of these, indicated in blue, were within a 16kb region on either side of ARS404, a distance that forks could have travelled maximally at 80 minutes under HU conditions. This suggests that we cannot exclude an indirect effect of these alternate origins on Pol2 levels at the DownTag near ARS404 in these mutants. This has now been mentioned in the fourth paragraph of the Discussion section of our revised manuscript.

Reviewer only Figure 1: Visualisation of alternate origins around ARS404, created with data from Foss et al. (eLife, 2024). Shown is a Genome browser view of a 78kb region (top), including the location of genes (middle) and origins of replication (bottom). ARS403, ARS404 and ARS405 are known origins. Vertical lines (grey bar) indicate alternate origins (Foss et al., eLife, 2024). Blue lines indicate alternate origins from which are within a 16kb distance on either side of ARS404.

5. Add the Pol2 myc tagged *sld3-38A dbf4-4A* strain to strain table.

The Pol2-Myc tagged *sld3-38A dbf4-4A* strain has been added to the strain table (Supplementary Table S2).

6. p 6. "Moreover, 154 mutants showed reduced Pol \square levels at 40 minutes (Log2 (fold change) < -1.25), but not at 80 minutes, possibly reflecting impaired fork progression, whereas 202 mutants showed this effect at 80 minutes (Log2 (fold change) < -1.25), but not at 40 minutes, likely due to enhanced fork progression"

Please explain logic in more detail and which site is being referred to (0 kb versus distal site primers or both). The logic for the interpretation is not clear. For the first case, if authors are analyzing 0 kb from origin, this could instead reflect a timing delay in origin firing/ recruitment of Pol epsilon to origin in the mutant vs wt, independent of the presence of *sld3-38A dbf4-4A*

and equal Pol epsilon at 80 min could simply reflect the origin fired later in the mutant and have nothing to do with "progression".

Only the DownTag barcode directly adjacent to ARS404 was used in the Repli-ID screen (see Figure 1). To improve the clarity of our manuscript and circumvent misunderstanding about the site that is being referred to, we decided to remove ARS404+1.6kb (UPTag barcode) from all ChIP-qPCR figures in the manuscript. The Results section in the manuscript, and the qPCR primer table (Supplementary Table S2) have been adapted accordingly, meaning that we now refer to Pol2 levels at the DownTag only in both the Repli-ID screen and ChIP-qPCR experiments.

We thank the reviewer for suggesting the alternative interpretation of decreased and increased Pol2 levels at 40 and 80 minutes, respectively. We now mention this in the last paragraph of the subsection "Towards Repli-ID: a new method to identify regulators of replication fork stability/progression in yeast" of the Results section that "As decreased Pol ϵ at ARS404 at both timepoints is more likely to reflect an unstable fork, including both timepoints enables us to distinguish slower travelling or delayed forks (e.g. from delayed origin firing) from unstable forks." Furthermore, we now mention in the first paragraph of the subsection "Repli-ID identifies known regulators of replication fork stability/progression" that the decrease Pol ϵ levels were "likely due to impaired replication fork stability/progression and/or reduced S-phase entry" in the Results section and origin loading, licensing, S-phase entry is discussed in the second paragraph of the Discussion section as possible explanations for Pol2 enrichment defect.

7. One thing to consider for "...a large fraction of mutants with impaired RNAPII transcription was found to have low levels of Pol epsilon.." Immediately upstream of ARS404 are the overlapping genes on opposite strands SSB1/YDL228C. If the transcription mutants are affecting SSB1 vs. YDL228C promoter usage (YDL228C's promoter overlaps the ARS), this has the potential to influence a competition between the TXN machinery and Pol epsilon (or chromatin accessibility facilitated by txn machinery) at the site assessed by the screen if 0 kb from origin was used. Testing the SSB1 vs. YDL228C transcript ratios in the transcription mutants should clarify this possibility.

The two genes, *SSB1* and *YDL228C*, are located upstream of ARS404, while the DownTag barcode is located 53 bp downstream of this origin of replication. This means that replication forks that passes the DownTag barcode will not be affected by transcription of these genes. Only forks travelling towards these two genes could be affected. Importantly, the DownTag barcode is located between ARS404 and the KanMX marker. This means that only forks that have passed the DownTag barcode could be affected by transcription of the KANMX marker. Thus, we believe that transcription did not impact Pol2 levels measured at the DownTag barcode in our Repli-ID screen.

8. Flow cytometry or additional budding index data should be provided to confirm discussed mutant (e.g. highlighted in Fig. 1) phenotypes do not reflect a defect in release from alpha factor G1 arrest, which would indirectly lead to reduced origin firing/association of Pol epsilon at time of cell harvest, but be unrelated to DNA replication per se. This appears to be the case for the rox1 line of inquiry in Fig. 5, not (noted in end of results section) a "new factor that impacts Pol epsilon progression". An alternate interpretation is the reduction of Pol epsilon ChIP signal in Fig 5e is probably primarily reflecting that only a subset of cells released from

the G1 arrest during the experiment. Would Pol epsilon normally be recruited to origins in cells still in G1? This is very different than "ensuring proper progression and stability of Pol epsilon", which implies a fully assembled fork has left an origin after START. Care with/Change of phrasing is recommended. In the long run, outside of the scope of this manuscript, it may be useful to include a secondary screen with Repli-ID to differentiate between factors that influence "release from G1 arrest" and those that act at the level of replication.

We acknowledge that it is unlikely that in G1-phase cells Pol2 binds at origins of replication. Indeed, this is what we observed for ARS607 in several Pol2 ChIP-qPCR experiments (see e.g. Fig. 3B and 5A). Therefore, cells indeed need to be released from G1 before origin firing/association of Pol2 can happen. Hoose et al. (PLoS Genet., 2012) identified 152 mutants showing an increase in the fraction of G1-phase cells. We compared these mutants to the 423 mutants identified in the Repli-ID screen and found 21 mutants that overlapped (3.5%; see new Fig. S2c), indicating that Repli-ID also enriches for mutants showing a defect in the release from G1-arrest. This result has been added to the manuscript (see new Fig. S2c) and described in the second paragraph of the subsection " Repli-ID identifies known regulators of replication fork stability/progression". Moreover, in the last paragraph of the Results section of our manuscript, the phrasing "ensuring proper progression and stability of Pol epsilon" has been changed to "ensuring proper Pol epsilon levels". Furthermore, origin loading, licensing, S-phase entry is discussed in the second paragraph of the Discussion section as possible explanations for Pol2 enrichment defect.

Among the identified mutants with decreased Pol2 levels depicted in Fig. 1a, *yta7Δ*, *nsr1Δ* and *pml1Δ* showed an increase in the fraction of G1-phase cells. We validated the Pol2 phenotype in *nsr1Δ* in another strain background, namely W303 (see Fig. 2D). However, the consistent decrease in Pol2 levels across different strain background, it did not confer HU sensitivity (see Supplemental Fig. S2E). This suggests that the Pol2 phenotype of *nsr1Δ* is due to a defect in the release from G1 phase rather than impaired DNA replication.

9. For the sup. figures in excel, more detail/a key is need for worksheets. e.g. Sup Fig S3c. Readers would be helped with a definition of the unit for the unitless numbers. The number of significant digits shown should reflect the whether the assay can really be accurately measured to four significant digits. Or not.

We added units to the unitless numbers in each table. The numbers for ChIP-qPCR, RT-qPCR and copy number are rounded to 2 decimal numbers, since the Ct value were obtained with 2 decimal numbers. The budding index, viability assay and GO-term analysis are percentages with SEM, which were rounded to 1 and 2 decimal numbers, respectively as recommended by the IES-NCES (https://nces.ed.gov/statprog/2002/std5_3.asp).

Reviewer #2 (Remarks to the Author):

In this article, SC van der Horst, L Kollenstart and colleagues present a new technical approach, termed Repli-ID, to screen for mutants affecting the stabilization of DNA polymerase epsilon on chromatin around replication origins after depleting dNTP pools with hydroxyurea. This technique is based on Pol2 ChIP at a barcoded origin of replication followed by next-generation sequencing. They identified mutants affecting the binding of Pol2 to chromatin upon HU treatment and undertook the mechanistic characterization of two candidates, Lge1 and Rox1. They conclude that Lge1 and Rox1 are indirectly required for the efficient activation of the S-phase checkpoint through epigenetic and metabolic pathways, respectively, and that Repli-ID could be used in future studies to better understand the response to replication stress.

Overall, the reviewers felt that, although this technique appears of interest for the discovery of new players in the replication stress response, the manuscript lacks clarity in many aspects. The introductory and discussion parts need a better inclusion of already published data. The choices of the technical approach parameters are not well justified or not well described. There are flaws in the data analysis and the conclusions are not always supported by the data.

Thus, the reviewers think that a major revision is required and have the following recommendations to improve the manuscript and promote its publication:

(There is no page numbering in the manuscript, rendering the review process very tedious).

Global improvement of the introduction part:

1. Better describe the functions of the DDR.

What is the purpose of increasing the transcription of RNR genes? Are other genes' transcription induced? Why suppressing late origin firing? What means "late"? Other functions?

Better describe the mechanistic activation of the DDR.

How ssDNA stretches are exposed? Current models?

Why the authors cite REF#5 Cobb et al., 2003 if they do not describe the role of Sgs1 in facilitating the activation of Rad53?

We have changed the Introduction of the manuscript and now better describe the functions of the DDR in the introduction, by 1) mentioning the purpose of increasing the transcription of RNR genes (to upregulate dNTP pools), 2) indicating that DDR genes are transcriptionally induced, and 3) explaining the reason for the suppression of late origin firing. Furthermore, we more extensively describe the mechanistic activation of the DDR (specifically the activation of Mec1) and explain how ssDNA is exposed according to the current models. Finally, we briefly mention the role of Sgs1 in Rad53 activation when citing Cobb et al. (EMBO. J., 2003).

2. The authors claim in the abstract that Repli-ID has been elaborated to identify new regulators of the replication stress response. However, they describe in the introduction that

Repli-ID is meant to measure replication fork progression or stability, which is more accurate considering that they immunoprecipitated the DNA polymerase epsilon. To identify new regulators of the DDR, one could expect a more direct ChIP readout as the recruitment to chromatin of Mec1, Ddc2, or Rad53. Please correct the abstract.

The abstract has been corrected accordingly.

3. Concerning the choice to ChIP Pol2 around replication origins under HU, there is a complete lack of justification of this choice and a lack of description of the known literature. For example, REF#5, Katou et al., 2003, De Piccoli et al., 2012 and others.

We apologize for not having cited these papers and added them to the manuscript. We have also added further justification of the Repli-ID set-up in the first sub-section of the Results section of our manuscript. In brief, our aim was not only to identify factors that play a role in DNA replication itself, but also factors that impact this process under stress conditions, using HU exposure as a means to deplete dNTPs and perturb DNA replication, the latter of which is monitored by Pol2 ChIP-qPCR (as was done in e.g. Cobb et al, EMBO J, 2003; Bjergbaek et al., EMBO J., 2004; Tittel-Elmer et al., EMBO J., 2009; Shimada et al, Life Sci Alliance, 2021). The identification, validation and mechanistic follow-up studies on some of the identified factors should reveal new insights into how cells deal with DNA replication stress.

Interpretation of Pol2 ChIP-qPCR data under HU has been under debate for years and Pol2 ChIP-seq data have identified disparities between the binding pattern of Pol2 around very early replication origins like ARS607 used in this work and other origins (De Piccoli et al., 2012). These data may prevent the generalization of the conclusions made in this manuscript.

We are aware of the disparities between Pol2 binding at different origins across the genome. However, the goal of our study was not to study Pol2 binding across the genome in different yeast mutants. Our Repli-ID screen set out to identify regulators of replication forks using ARS404 as a model locus. We successfully identified mutants in which Pol2 was reduced at ARS404. We validated these findings for several mutants from the screen, not only at ARS404, but also at ARS607 and ARS305. These findings suggest that our screen identified mutants that may generally impact Pol2 levels at origins of replication. However, we cannot rule out that the identified mutants may have a different impact on replication forks at different origins across the genome. Furthermore, in the Discussion section we mention that the lower Pol2 levels may also be due to replisomes that moved away from sites of DNA synthesis (De Piccoli et al, Mol Cell, 2012).

Results section, screen validation and results:

4. The authors do not justify why they choose to study the binding of Pol2 near ARS404, which, as such, seems irrelevant since this replication origin is inhibited under HU.

We aimed at using the freely available barcode library of yeast strains (Yan et al., Nat. Methods, 2008), rather than barcoding all mutants from the yeast knockout collection ourselves. In the strains from the barcode library, the barcodes were integrated at the *HO* locus, which is located downstream of the late firing origin ARS404 (see new Figure 1). ARS404 is indeed inhibited under HU conditions. However, by crossing we introduced a

cassette for overexpression of *sld3-38A dbf4-4A* in these strains (Zegerman and Diffley, Nature, 2010; Mantiero et al., EMBO J., 2011), allowing the firing of ARS404 in the presence of HU (see Fig. S1b and S1d). This allowed us to study Pol2 binding at ARS404. We adjusted the first paragraph of the sub-section “Towards Repli-ID: a new method to identify regulators of replication fork stability/progression in yeast” of the Results section of the manuscript to better justify our experimental setup.

They describe this result in Supplementary Figure S1b, which also describe the binding of Pol2 at the early origin ARS607. However, the data of Pol2 ChIP at ARS607 is not consistent with what is expected (progression along time from ARS607 to ARS607 + 4kb) and with similar data presented in Figures 3 and 4.

The main point of Fig. S1b was to show that ARS607, but not ARS404, fires in wild-type cells, confirming that ARS404 is a late firing origin. In contrast, Fig. S1d shows that in galactose conditions, when *sld3-38A dbf4-4A* are overexpressed (from a galactose-inducible promoter), both ARS607 and ARS404 fire. Pol2 abundance in Fig. S1b (and perhaps also in Fig. S1d) would indeed have been expected to be somewhat higher at ARS607 20 minutes after release in HU. However, this is possibly due to a less efficient alpha-factor wash/release in these initial experiments of the project.

All over the manuscript, the reviewers noticed huge variations in the “fold increase” of Pol2 ChIP in wild-type cells between experiments. They believe that the double normalization method used for ChIP experiments (first normalization over the beads only signal (Figure 2) or IgG signal (Supplementary Figure S1) and second normalization over a non-replicated region ARS607+14kb) is critically impacting the enrichment of the proteins of interest, thus affecting conclusions.

Raw IP/input ratios (without normalization) may be a better way to present their data or use only one normalization parameter.

In Fig. 2a, Fig. 2c and Fig. S1b-e BY4741 strains were used that overexpress *sld3-38A dbf4-4A* in galactose-containing medium. In all other figures W303 strains were used that were grown in glucose-containing medium. Differences in medium and carbon source, as well as in strain background may explain some of the differences in Pol2 abundance. Besides this, Pol2 ChIP-qPCR experiments were done over years of time with different batches of formaldehyde, alpha-factor, beads and antibodies, leading to experimental variation and impacting Pol2 abundance across experiments. However, in all experiment the Pol2 abundance in mutants was compared to that in wild-type under the same experimental conditions, ruling out that any impact of a mutant on Pol2 abundance was due experimental variation.

We used the beads only control (or IgG) to remove noise/non-specific binding of DNA to the beads, which is a commonly used method in the field (e.g Cobb et al, EMBO J., 2003). The internal normalization to ARS607+14kb, a region that does not become replicated in presence of HU, removes the inter-sample differences caused by e.g. pipetting errors or differences in IP efficiency. This is also a commonly used method in the field (e.g. Han et al., Cell, 2013; Shimada et al., Life Sci. Alliance, 2021; Oh et al., Cell Rep., 2018).

Moreover, Pol2 ChIP data analysis would benefit a lot from the parallel analysis of the copy number increase by using the qPCR data of input samples after release in S phase normalized

by the qPCR data of input samples in G1 (Time 0). This would allow a parallel assessment of the progression of replication forks under the experimental conditions used.

Our experience is that assessing the copy number increase using qPCR data from ChIP input samples (from G1- and S-phase cells) results in highly variable outcomes (unpublished data). We therefore established a copy number assay that employs qPCR on genomic DNA isolated from G1- and S-phase cells. A calibration curve is established on serially diluted genomic DNA from asynchronous cells and used to calculate the copy number based on qPCR values from S-phase cells versus G1-phase cells. We have previously used this method for copy number analysis of wild-type, *chl1* and *mrc1* mutant strains (Batte, Van der Horst et al., Life Sci. Alliance, 2022). Here, we have used this method to show a lowering in copy numbers in *lge1* and *rox1* mutant strains when compared to wild-type (see new Fig. S3d and S5a), which is in agreement with the decreased Pol2 abundance observed in these mutants (Fig. 3b and 5a).

5. The use of ARS404 in Repli-ID forces the authors to perform the screen in the *sld3-38A dbf4-4A* genetic background. It is not clear why they decided to overexpress the mutant alleles in the presence of wild-type alleles rather than using a strain only expressing the mutant alleles constitutively. It seems to complicate the experimental procedure by working in a culture medium with raffinose supplemented with galactose to overexpress the mutants. Please explain.

Using this inducible system, *sld3-38A* and *dbf4-4A* were only overexpressed 30 minutes before the release into S-phase in the presence of HU. This way, we can exclude possible secondary effects from aberrant DNA replication, including the induction of DNA damage, caused by their overexpression in previous cell cycles (Zegerman and Diffley, Nature, 2010).

Moreover, they authors should indicate that DNA replication in the *sld3-38A dbf4-4A* genetic background has detrimental consequences: global lower efficiency of origin firing, spontaneous activation of the DDR (gamma-H2A signal), increased DNA damage in response to HU (Rad52 foci formation), increase in topological stress. The authors should cite the relevant literature, for example, Morafraille et al., 2019, and discuss the implications of using the *sld3-38A dbf4-4A* mutant background in Repli-ID screens.

Morafraille et al. (Genes & Dev., 2019) and Zegerman and Diffley (Nature, 2010) observed that cells overexpressing *sld3-38A dbf4-4A* are fully competent in essential functions in replication initiation and checkpoint activation. However, Morafraille et al. also showed that the increase in replication initiation causes the accumulation of DNA damage markers γ H2A and Rad52 due to topological stress. Since we examined Pol2 binding at relatively early timepoints in S-phase (40 and 80 minutes) and close to ARS404, which is far from other commonly known origins (30.4kb (ARS403) and 39.7kb (ARS405) away from ARS404), the topological stress will be minimized, although alternate origins could have an impact on topological stress (see point 4 of reviewer 1). We have added these points to fourth paragraph of the Discussion section of the manuscript.

6. The reviewers understood that the authors justify the choice of using the 40min and 80min time points in HU for the screen because they observed maximum Pol2 ChIP enrichment values at the replication origin at 40min and away from the origin at 80min (Supplementary Figure S1c). However, they claim in their conclusion that the 80min time point is ideal for

sample collection during Repli-ID screens because Pol2 ChIP enrichment values “decline” at ARS404 at 80min (Supplementary Figure S1d). Please clarify your justification.

This is correct and we have added the following justification to the first sub-section of the Results section of our manuscript:

“The strongest recruitment of Pol ϵ at ARS404 was observed 40 minutes after release in S-phase (Fig. S1e). After 80 minutes, Pol ϵ declined at ARS404, suggesting progression of the replication fork away from the origin into the neighbouring chromatin. As decreased Pol ϵ near ARS404 at both timepoints is more likely to reflect an unstable fork, including both timepoints enables us to distinguish slower travelling forks (e.g. from delayed origin firing) from unstable forks. Therefore, these timepoints are ideal for sample collection during Repli-ID screens.”

Moreover, the data should be better described. For example, Pol2 signals accumulate at the origin at early time points, then progressively decrease while increasing away from the origin at later time points, indicating replication fork progression.

This has been changed accordingly throughout the Results section of the manuscript.

7. The authors describe that they built the Repli-ID library by crossing the Pol2-9Myc sld3-38A dbf4-4A with the barcoder mutant library (Figure 1a) but this is not depicted in the Figure 1a.

We have added this to Figure 1b.

The distance from the ARS404 at which the barcode sequence is inserted is not indicated. Thus, it is difficult to understand if Repli-ID is assessing the binding of Pol2 at the replication origin or away from the origin. The fact that Pol2 hardly accumulated away from ARS404 in the experimental conditions used (low Pol2 enrichment at ARS404+1.6 kb) questions the ability of the Repli-ID technique to assess fork progression.

We have now indicated the distance from ARS404 at which the barcode sequence was inserted (53 bp; see new Fig. 1a). Given that the locus is only 53 base pairs away from ARS404, it is possible that we have been examining Pol2 loading or S-phase entry rather than actual progression. We now mention in the first paragraph of the subsection “Repli-ID identifies known regulators of replication fork stability/progression” that the decrease Pol ϵ levels were “likely due to impaired replication fork stability/progression and/or reduced S-phase entry” in the Results section and origin loading, licensing, S-phase entry is discussed in the second paragraph of the Discussion section as possible explanations for Pol2 enrichment defect.

Indeed, at the 40-minute timepoint, which we used for validation (previous Fig. 2c), there was no significant increase of Pol2 binding at ARS404+1.6kb (previous Fig. S1d-e and previous Fig. 2c). Similar observations were made at ARS607+4kb. However, we observed a 3-5 fold increase in Pol2 enrichment at ARS404+1.6kb after 60 and 80 minutes in wild-type cells, which is indicative of replication fork progression away from ARS404 (previous Figure S1d-e). To improve the clarity of our manuscript and circumvent misunderstanding about the site that is being referred to, we decided to remove ARS404+1.6kb (the UPTag barcode) from all ChIP-qPCR figures in the manuscript (see point 6 of reviewer 1).

8. Having performed the screen, the authors identified 423 deletion mutants in which Pol2 levels were decreased at both time points. They conclude that these mutants are likely affected in fork stability/progression under HU stress. They cannot exclude that these mutants can be

affected in replication initiation independently of replication stress and should test, at least in some candidates, if Pol2 binding to the barcode sequence is affected in the absence of HU.

We randomly selected three hits from the Repli-ID screen, *rec8Δ*, *fpr4Δ* and *apt1Δ*, for which we validated reduced Pol2 levels near ARS404 using ChIP-qPCR (see Fig. 2c). We performed ChIP-qPCR experiments at 20 degrees, which slows down S-phase progression in the absence of HU, allowing us to capture Pol2 (which would otherwise progress to quickly during the course of the experiment and in the absence of HU). In *rec8Δ* cells, Pol2 levels were similar to that observed in wild-type cells after 30 minutes in S-phase (see new results in Fig. S2d). In contrast, at this timepoint Pol2 levels were decreased in *fpr4Δ* and *apt1Δ* cells when compared to that in wild-type, suggesting that replication initiation in these mutants, but not in *rec8Δ*, may be affected (see new results in Fig. S2d). We have mentioned these new results in the first paragraph of the subsection “Repli-ID identifies novel factors that affect replication fork stability/progression” in the Results section and discussed them in the second paragraph of the Discussion. Furthermore, as mentioned above, we now mention in the first paragraph of the subsection “Repli-ID identifies known regulators of replication fork stability/progression” that the decrease Pol ϵ levels were “likely due to impaired replication fork stability/progression and/or reduced S-phase entry” in the Results section and origin loading, licensing, S-phase entry is discussed in the second paragraph of the Discussion section as possible explanations for Pol2 enrichment defect.

They also identified 128 mutants in which Pol2 levels were increased at both time points. They have not interpreted these data.

We have added an interpretation of the data to the first paragraph of the sub-section “Repli-ID identifies known regulators and replication fork stability/progression” of the Results section, as well as to the third paragraph of the Discussion, by mentioning that this phenotype may indicate stalled, but not unstable replication forks.

They also identified 154 mutants in which Pol2 levels were decreased at 40min but not at 80min, suggesting an impaired fork progression, and 202 mutants in which Pol2 levels were not decreased at 40min but decreased at 80min, suggesting enhanced fork progression.

The authors should discuss further these data and propose alternative interpretations. For instance, decreased signals at a replication origin at both time points are reminiscent of what has been observed in *mec1* and *rad53* mutants, defective in checkpoint activation. Pol2 signals were detected only 15 kb away from some very early replication origins in these mutants (De Piccoli et al., 2012), reflecting replication fork progression before dNTP depletion.

We have addressed this accordingly in the second paragraph of the Discussion.

In light of the checkpoint activation defect observed in both *lge1Δ* and *rox1Δ* mutants, the authors should reconsider their data interpretation.

The *lge1Δ* and *rox1Δ* mutants showed partial defects in Rad53 activation, suggesting they are not fully checkpoint-deficient (see Fig. 4b and 6b). Accordingly, we observed that, in contrast to that in *mec1* and *rad53* mutants (Crabbé et al, Nat Struct Mol Biol, 2010), late origins still did not fire in *lge1Δ* after 60 minutes in HU (see Fig. S4d). We therefore feel that in terms of replication fork progression the *lge1Δ* and *rox1Δ* may also not fully recapitulate the phenotype of checkpoint-deficient *mec1* and *rad53* mutants.

9. The strongest hits showing Pol2 decrease at ARS404 are mutants of factors involved in DNA replication (Pol32, Mgs1 and Pif1) but none of them has been shown to impact the stability of Pol2 under HU. On the contrary, the *sgs1Δ* mutant did not come out as a strong hit but has already been shown to decrease Pol2 levels at replication origins under HU by ChIP-qPCR (Cobb et al., 2003). This would have validated the new findings of the screen.

The authors should propose an explanation for the absence of *sgs1Δ* in their strongest hits, although it is present in the Repli-ID library and test it individually by ChIP-qPCR at ARS404 and ARS607 in the *sld3-38A dbf4-4A* genetic background.

We retrieved the *sgs1Δ* strain from the Repli-ID library and performed a spot dilution test to assess the HU sensitivity. As expected, *sgs1Δ* cells from the Repli-ID library were hypersensitive to HU. This sensitivity was comparable to that of an *sgs1Δ* mutant (oHA-1075) generated in our laboratory (see Reviewer only Figure 2).

Reviewer only Figure 2: Spot dilution assay with the indicated strains. Five-fold serial dilutions were spotted on rich YPAD medium without or with the indicated concentrations of HU.

We also performed a Pol2 ChIP-qPCR with the *sgs1Δ* strain from the Repli-ID library. As expected, we observed reduced Pol2 levels near ARS404 and ARS607 (Reviewer only Figure 3).

Reviewer only Figure 3: ChIP-qPCR analysis of Pol ϵ -9xMyc at ARS607 and ARS404 in the indicated strains at different timepoints after release from G1 into S-phase in the presence of 200 mM HU. Data represent the mean relative fold enrichment + SEM of Myc signal over beads only signal of four independent experiments. Values were normalized to a non-replicated region (ARS607+14kb).

The outcomes of the spot dilution test and Pol2 ChIP-qPCR experiment indicate that the *sgs1Δ* strain in the Repli-ID library is correct. We hypothesize that it did not appear as a hit in our Repli-ID because *sgs1Δ* cells are sensitive to suppressor mutations (Schmidt and Kolodner, Proc. Natl. Acad. Sci. USA, 2006), a problem we have previously observed during genetic interaction screens that included this mutant (Guérolé et al, Mol. Cell, 2012; Sun et al, J. Cell Sci., 2020).

10. Among the candidate mutants showing reduced Pol2 levels at both time points, the authors curated the results by assessing Pol2 levels in individual mutants by ChIP-qPCR. The authors should clearly describe that the first step of curation was performed in the *sld3-38A dbf4-4A* background (Figure 2c) and the second step in mutants generated de novo in wild-type W303 background (Supplementary Figure S2c). The latter panel S2c should be included in the main Figure 2. In this experiment, the authors must justify why they decided to assess Pol2 levels at the time point 20min and not 40min or 80min as in the Repli-ID screen. They also have to explain to the readers why they decided to include ARS305 and ARS501.

We have explained this now in the the sub-section “Repli-ID identifies novel factors that affect replication fork stability/progression” of the Results section of the manuscript. The first step of curation was performed in the *sld3-38A dbf4-4A* background and the second step in mutants generated de novo in wild-type strains of the W303 background. Supplementary Fig. S2c has been moved to Figure 2 (panel d).

In W303 cells, we observed the highest enrichment at ARS607 after 20 minutes (see Fig. 3b) and therefore decided to use this timepoint. ARS305 is another early firing origin, which we included to further validate the data from ARS404. ARS501 is a late firing origin, which does not fire after 20 minutes in HU, and was therefore included as a negative control. We have also clarified this now in the aforementioned sub-section of the Results section.

11. The authors observed that *rox1Δ* and *lge1Δ* mutants are sensitive to high doses of HU and use this result to justify further characterization of these mutants. Exposure to high doses of HU not only depletes dNTPs but also leads to the production of reactive oxygen species (ROS; Huang et al., 2016), which can explain the mutants’ sensitivity independently of a replication defect. The authors should thus test HU sensitivity in the presence of a ROS scavenger (N-acetyl-l-cysteine 10mM or alpha-ketoglutarate 5mM) to demonstrate that *rox1Δ* and *lge1Δ* mutants have defects related to the depletion of dNTP pools and the stalling of replication forks by HU.

We have tested the HU sensitivity of *rox1Δ* and *lge1Δ* in the presence of the ROS scavenger N-acetyl-l-cysteine (NAC). No rescue of the HU sensitivity phenotype in presence of NAC was observed, which indicates that the HU phenotype we observed was not due to the production of excessive ROS, but to the depletion of dNTP pools and stalling of replication forks by HU (see new Supplementary Fig. S3f and Supplementary Fig. S6d). We mentioned this at the end of the first paragraph of the subsection “Lge1 affects DNA replication under stress conditions by promoting H2B mono-ubiquitylation” and at the end of the subsection “Rox1 regulates ceramide levels via Sur2 to control the response to replication stress” of the Results section in our manuscript.

Results section, *lge1* Δ characterization:

12. The authors state that a role for Lge1 in the response to replication stress has not been described yet. This is maybe true, but they should also clearly indicate that a role for Bre1, co-factor of Lge1 for histone H2B ubiquitylation, has been deeply investigated in response to HU-induced replication stress. They cited the relevant literature (REF#24) without recalling the main phenotypes caused by the absence of Bre1 or the ubiquitylation of histone H2B (*htb-K123R* mutant) on DNA replication.

The authors should acknowledge these previous findings and interpret their results in light of these previous findings, which also showed that the absence of H2B ubiquitylation led to a checkpoint activation defect.

This has been changed accordingly in the first paragraph of the sub-section “Lge1 affects DNA replication under stress conditions by promoting H2B mono-ubiquitylation” of the Results section in the manuscript.

13. The authors found that *lge1* Δ and *bre1* Δ mutants have increased dNTP levels but argue that this cannot impair replication progression and explain the low levels of Pol2 at replication origins under HU. However, mutants with high dNTP levels have been described to progress further away from origins than wild-type cells in HU because of a delayed exhaustion of dNTPs. A consequence of this extended progression could result in the dilution of the ChIP signal, explaining why Pol2 signals are lower in *lge1* Δ than in wild-type cells. This would also indicate that ChIP-qPCR values should not be normalized by ARS607+14kb.

Here again, exploiting the DNA copy number increase data from the ChIP data should indicate the degree of replication progression independently of Pol2 binding to chromatin. This readout is much more informative than the budding index.

We performed this copy number analysis and found the copy number to be reduced in *lge1* Δ cells when compared to that in wild-type cells (see new Fig. S3d), consistent with reduced Pol2 levels (Fig. 3b). These new results affirm a DNA replication defect in *lge1* Δ cells (Fig. S3d), and indicate that the normalisation to ARS607+14kb, which is a commonly used method in the field (e.g. Han et al., Cell, 2013; Shimada et al., Life Sci. Alliance, 2021; Oh et al., Cell Rep., 2018), is valid. The internal normalization to ARS607+14kb in ChIP-qPCR experiments is important as it removes the inter-sample differences caused by e.g. pipetting errors or differences in IP efficiency.

14. The loss of Lge1 or Bre1 resulted in strongly decreased levels of H2Bub in G1 and S phase +HU. The authors concluded that Lge1- and Bre1-dependent H2B ubiquitylation facilitate DNA replication at HU-stalled forks. This conclusion is not justified by the data. H2Bub levels are high in all regions close to ARS607 without DNA replication or without HU in G1-arrested wild-type cells, likely due to the presence of transcribed genes. After the release of cells in S phase +HU, the pattern of H2Bub does not change along time, whereas Pol2 progressively travel from ARS607 to ARS607+8kb. Thus, the progression of DNA replication under HU cannot be linked to H2B ubiquitylation.

In *lge1* Δ cells, H2Bub levels are globally lost (Figures 3c, 3d and Supplementary Figure S3a) in G1-arrested cells and S phase +HU. Whether the absence of H2Bub impacts the stability

of Pol2 under HU is just a correlation. It could be confirmed by measuring Pol2 levels in *bre1Δ* or *htb-K123R*.

We measured Pol2 levels in *bre1Δ* and observed reduced Pol2 abundance, a phenotype that resembles that of *lge1Δ* cells (Fig. 3b) This suggest that it is indeed the loss of H2B ubiquitylation that impacts Pol2 levels (see new Fig. 3e).

15. The efficiency of Bre1 ChIP is doubtful (Figure 3e). Normalized fold increase values are very low (around 0.5 above the background) in wild-type cells. A positive control validating the ChIP efficiency would have been welcome. The figure legend indicates that normalization was done on the beads only signal but source data indicate TELVI (?). Raw IP/input ratios may be a better indicator of the efficiency of Bre1 ChIP.

The efficiency of the Bre1 ChIP was indeed low, but very similar to that observed previously (Fig. 3b in Liu *et al.*, Proc. Natl. Acad. Sci. USA, 2021). We included a positive control, the *PMA1* locus, previously used by Song and Ahn (J. Biol. Chem., 2010), at which we observed up to a 7-fold enrichment of Bre1 over the background (see new Fig. S4a). Bre1 signals were first normalized to that of the beads only signal (to remove signals caused by background binding of DNA to the beads). Next, signals were normalized to that at a telomere region, TelVI (0.5 kb away from telomere on right arm of chromosome 6), at which no binding of Bre1 is expected due to a lack of histones. The normalisation procedure has been clarified in more detail now in the figure legend of Fig. 3d.

The authors propose that the loss of H2Bub in *lge1Δ* could be due to an upregulated recruitment of Ubp8. However, they do not test this hypothesis experimentally in *ubp8Δ* cells. This may be worth mentioning only in the discussion.

We tested this hypothesis experimentally by assessing the H2Bub levels in *lge1Δ* and *lge1Δubp8Δ*. However, H2Bub levels in *lge1Δubp8Δ* were not restored to that in *lge1Δ* (see new result in Fig. S4c). This suggests that the loss of H2Bub in *lge1Δ*, while Bre1 is still partially present at the fork, is not due to upregulated Ubp8 recruitment but rather due to loss of Bre1 functionality when Lge1 is absent. We have changed this accordingly in the second paragraph of the sub-section “Bre1 recruitment to stalled replication forks is partially Lge1-dependent” of the Result section of the manuscript.

16. The authors study the kinetics of Rad53 phosphorylation as an indicator of the checkpoint response (Figure 3f). The source data indicate that the western blot picture has been cut to select time points of interest, which may explain the presence of dotted lines on the picture. What means “X” on the source picture? How many times this experiment has been performed?

In the experiment shown in Figure 3f, we had also included samples for timepoint 60 minutes. These are shown in the source picture in which they are marked by “X”. Unfortunately, a smear of Rad53 bands was observed for the wild-type (wt) sample, whereas a bubble appeared on the blot for the *lge1Δ* sample. Therefore, we decided to cut the blot and remove the samples for timepoint 60 minutes.

We have performed this experiment two times. The first experiment is shown in Figure 3f, the second in Reviewer only_Figure 4. Although the impact of Lge1 or Bre1 loss on Rad53 activation is rather similar in the two experiments, the blots of the first experiment were in our opinion of better quality, hence their inclusion in the main figures (Fig. 3f).

Reviewer only_Figure 4: Western blot analysis of FLAG-Rad53 in the indicated strains at different timepoints after G1-arrest and release in S-phase in 200 mM HU.

These data indicate that *lge1Δ* cells have a delayed checkpoint activation, as previously observed in *htb-K123R* mutant (REF#24). As stated by the authors, efficient checkpoint activation is critical for the recovery of cells from HU-induced replication stalling and tested it in *lge1Δ* and *bre1Δ* mutants. However, their data did not indicate that the checkpoint activation was not “efficient” but “delayed”.

We thank the reviewer for raising this point. Checkpoint activation was indeed “delayed” rather than “not efficient” in *lge1Δ* and *bre1Δ* mutants. We have changed the phrasing on this point in the subsection “Lge1 controls the intra-S-checkpoint and fork recovery via H2B ubiquitylation” of the Results section in our manuscript.

Timely activation of the S-phase checkpoint is critical for the inhibition of late origin firing. The authors should therefore investigate if late origins are fired in the *lge1Δ* mutant as this may be related to the loss of viability. They observed that Pol2 was not recruited at ARS501 in *lge1Δ* (Supplementary Figure 2c) but late origins may fire later than 20min after release in S phase+HU. They also observed an impairment in recovery from HU arrest in *lge1Δ* and *bre1Δ* mutants to the same extent as in *mrc1Δ* (not described in the results). *mrc1Δ* mutant is a canonical example of a checkpoint deficient mutant that has lost the ability to inhibit late origin firing in HU.

We have performed new Pol2 ChIP-qPCR experiments to examine the presence of Pol2 at several late firing origins up to 60 minutes in S-phase in the presence of HU. However, we did not observe Pol2 binding at the late firing origins ARS501, ARS404 and ARS316 after 0, 20, 40 and 60 minutes, while Pol2 bound at the early firing origin ARS607 after 20 minutes, declining at the later timepoints as observed earlier (see new Supplemental Fig. S4d). These new results suggest that the HU sensitivity of *lge1Δ* cells is not caused by aberrant firing of late origins, which we now mention in the subsection “Lge1 controls the intra-S-checkpoint and fork recovery via H2B ubiquitylation” of the Results section in our manuscript.

Results section, *rox1Δ* characterization:

17. As indicated earlier, HU exposure also leads to ROS and *rox1Δ* phenotypes may be more linked to ROS accumulation than to dNTP depletion. Therefore, *rox1Δ* and *rox1Δ sur2Δ* sensitivity to HU should be re-evaluated in the presence of ROS scavengers.

We tested the HU sensitivity of *rox1Δ* and *rox1Δsur2Δ* in the presence of the ROS scavenger N-acetyl-L-cysteine (NAC). No rescue of the HU sensitivity phenotype in presence of NAC was observed, which indicates that the HU phenotype we observed was not due to the production of excessive reactive oxygen species, but to the depletion of dNTP pools and stalling of replication forks by HU (see new Supplementary Fig. S6d). We mentioned this at the end of

the subsection “Rox1 regulates ceramide levels via Sur2 to control the response to replication stress” of the Results section in our manuscript.

18. When looking at RNR genes expression, the authors should indicate that the increase of *RNR3* expression is expected in response to HU because it is a checkpoint response-induced gene.

The RNR expression levels in *rox1Δ* in Fig. S5b were normalized to that in wild-type, which was set to 1, for unperturbed and HU-treated conditions, respectively. Although a strong HU-dependent induction of *RNR3* expression was observed in *rox1Δ*, the impact of HU on *RNR3* expression could not be appreciated because of the normalisation. Therefore, we have now included the unnormalized data (see new Figure S5c), showing an HU-dependent increase in *RNR3* expression for wild-type strains, agreeing with previous reports (Huang et al., Cell, 1998; Zhou et al., Cell, 1993). This has also been mentioned in the second paragraph of the subsection “Rox1 does not impact DNA replication by affecting cellular dNTP levels” of the Results section in our manuscript.

19. In Figure 5a; the authors test how the addition of ceramide impacts the sensitivity of *rox1Δ* and *sur2Δ* mutants to HU. The description of the results is incomplete. It is worth mentioning that ceramide addition increased the sensitivity to HU of wild-type and *rox1Δ* cells, before describing that ceramide addition prevented the suppression of *rox1Δ* HU sensitivity by *sur2Δ*.

We now mentioned in the subsection “Rox1 regulates ceramide levels via Sur2 to control the response to replication stress”, of the Results section in our manuscript that ceramide addition increased the HU sensitivity wild-type and *rox1Δ* cells.

Ceramides activate PP2A through Irc21 (REF#42). Thus, it would be worth checking if the addition of ceramides in the absence of IRC21 still prevents the suppression and if *irc21Δ* could suppress the HU sensitivity and the defect in Rad53 phosphorylation in response to HU of *rox1Δ* cells.

According to Ref#42 (Ferrari et al., Mol. Cell., 2017), ceramides do not activate PP2A via Irc21. This paper rather showed that Irc21 is involved in PP2A activation by promoting the production of ceramides, thereby inhibiting the DNA damage response. Based on this, it may act similarly to Sur2 in suppressing the HU sensitivity of *rox1Δ*. To test this, we generated single, double, and triple deletions mutants of *ROX1*, *SUR2* and *IRC21* and performed drop test assays on HU. However, the *irc21Δ* single mutant was already hypersensitive to HU, making the epistasis analysis difficult if not impossible (see Reviewer only_Figure 5).

Reviewer only Figure 5: Spot dilution assay with the indicated strains. Five-fold serial dilutions were spotted on rich YPAD medium without or with the indicated concentrations of HU and ceramide.

20. In Figure 5c, the authors use the budding index as an indicator of the G1/S transition. It may not be a good indicator because the lipid metabolism is affected in *rox1Δ* cells and HU could also lead to lipids oxidation. Both could impede a normal bud growth independently of the G1/S transition.

We showed in the manuscript that in the absence of HU the G1/S transition of *rox1Δ* is rather comparable to that of wild-type cells *rox1Δ* (see Fig. S6f), indicating that the impact on lipid metabolism in *rox1Δ* cannot be the cause of the impaired G1/S transition in the presence of HU. Furthermore, the HU effect on lipid oxidation may be limited, since we added HU just after the release from G1-phase into S-phase.

21. When assessing Pol2 levels in *rox1Δ* and *sur2Δ* mutants (Figure 5e), the authors chose the time point 60min without any justification. Enrichment values are very different from those at the same positions in wild-type cells at 60min (Figure 3b), questioning the validity of the data.

Moreover, Pol2 levels are globally higher in *sur2Δ* than in wild-type cells, raising the possibility that the rescue in *rox1Δ sur2Δ* may be a compensatory effect.

We thank the reviewers for spotting this, as we only now realized that we made a mistake in the figure annotation. The Pol2 levels were measured at 20 minutes instead of 60 minutes in HU. We chose this timepoint as we observed the biggest impact of *ROX1* loss on Pol2. We corrected this error in the new version of the figure legend of Fig. 6e.

In the absence of ceramide, the levels of Pol2 were indeed somewhat higher in *sur2Δ* when compared to that in wild-type cells, while in the presence of ceramide the Pol2 levels were somewhat lower in *sur2Δ* when compared to that in wild-type cells. However, neither of these differences was statistically significant ($p > 0.05$).

Reviewer #3 (Remarks to the Author):

Reviewer #4 (Remarks to the Author):

Van der Horst et al. 2024

The authors establish a method they call Repli-ID (derived from Epi-ID) to enable high-throughput screening of barcoded mutant collections for genes affecting replication fork stability. The method uses ChIP of Pol e, the leading strand replicative DNA polymerase, near a barcoded origin as a proxy for replication fork stability in hydroxyurea (HU), which depletes dNTPs and slows/stalls DNA Polymerase. Relative dissociation (or accumulation) of Pol e from replicating chromatin is inferred to indicate some defect in fork stabilization/checkpoint signaling. Enriched or diminished number of specific barcodes indicates potential gene(s) involved.

While the overall concept of analyzing Pol e stability on DNA as a measure of fork stability in different mutants makes sense, a strangely baroque and incompletely explained aspect of this screen is the use of ARS404 as the barcoded locus for analysis. Presumably this is to use an existing barcoder strain library, but ARS404 doesn't fire in HU in this context (late/dormant), so they overexpress *Dbf4* and *Sld3* mutants to bypass the intra-S checkpoint. This raises some concern given their intended purpose to identify regulators of fork stability, which is dependent on the intra-S checkpoint. Bypassing the checkpoint can create replication stress, which might sensitize the assay and lead to identification of genes with indirect effects. Why not barcode ARS607?

We aimed at using the freely available barcode library of yeast strains (Yan et al., Nat. Methods, 2008), rather than barcoding all mutants from the yeast knockout collection ourselves. In the strains from the barcode library, the barcodes were integrated at the *HO* locus, which is located downstream of the late firing origin ARS404 (see new Figure 1). ARS404 is indeed inhibited under HU conditions. However, by crossing we introduced a cassette for overexpression of *sld3-38A dbf4-4A* (Zegerman and Diffley, Nature, 2010) in these strains, allowing the firing of ARS404 in the presence of HU (see Fig. S1b and S1d). This allowed us to study Pol2 binding at ARS404. To barcode ARS607, we would have to redesign the barcoder KO library with CRISPR/Cas9 in order to place all barcodes near ARS607, which would be laborious. We adjusted the discussion section of the manuscript accordingly to better justify our choice.

We have tested the HU sensitivity of *sld3-38A dbf4-4A* overexpressing strains on galactose (see new Fig. S1c). Only a very slight HU sensitivity, if at all, was observed for strains that constitutively overexpress *sld3-38A dbf4-4A*. However, during the Repli-ID screens, *sld3-38A dbf4-4A* were only overexpressed 30 minutes before the release of cells into S-phase in the presence of HU. This way, we can exclude possible secondary effects from aberrant DNA replication, including the induction of DNA damage, caused by their overexpression in previous cell cycles (Zegerman and Diffley, Nature, 2010).

Nevertheless, the results of a screen of ~2900 (or 4500?) knock-out strains resulted in identification of 423 strains with reduced Pol e association and 128 strains with increased signal, though stringency appears low. The 423 genes represented in the strains with reduced Pol e are said to be highly enriched for certain GO annotations like DNA repair, but statistical analysis appears absent. It also seems that certain categories not expected to be related to fork stability like “transmembrane transport” are similarly enriched so the significance of this analysis as validation is not established. Most replication genes are essential and thus are not represented in the knock-out collection; however, three non-essential replication/repair factors were identified, as might be anticipated, suggesting at least nominal functionality of the screen.

Repli-ID was performed using a library that consists of ~4700 yeast knockout strains that each carry a unique barcode near ARS404, express Myc-tagged Pol e, and overexpress *sld3-38A dbf4-4A*. Using this library Pol2 abundance was determined following ChIP and next-generation sequencing of the barcodes. Barcodes were counted, and depletion or enrichment of each barcode was determined as described by Vlaming et al. (eLife, 2016), with the exception that barcodes with read numbers below 0.0025% of the total reads in an input sample were excluded, as they were considered absent from the plate. Additionally, median-normalized barcode scores from the ChIPs were divided by the corresponding input scores. The ratios from two replicate screens were averaged for t = 40 minutes and t = 80 minutes, and overlapping ORFs were identified. This analysis produced Repli-ID results for 2,905 mutants (Supplementary Data 1). We have now described this analysis in the subsection “Repli-ID analysis” of the Methods, and mention in the first paragraph of the subsection “Repli-ID identifies known regulators of replication fork stability/progression” of the Results and in the legend of Figure 2 that Repli-ID provides Pol2 abundance outcomes for 2905 mutants.

We used the GoSLIM tool of SGD to find enrichment for GO-annotated processes. However, this tool does not include statistical analysis. To provide statistical analysis, we re-analyzed the data with PANTHER GO-Slim and found endosomal transport, vacuolar transport and chromatin organisation as significant hits (FDR<0.05). Chromatin organisation, which is known to impact DNA replication (Vogelauer et al., Mol. Cell, 2002; MacAlpine et al., Cold Spring Harb. Perspect. Biol., 2013), included interesting hits amongst which Dpb4, a subunit of DNA Pol epsilon contributing to the DNA polymerase complex stability, and Yta7, which is shown to promote nucleosome disassembly and replication (Chacin et al., Nat. Commun., 2021). Vacuolar transport and Endosomal transport may be explained by defects in vacuole biogenesis, resulting in cell cycle mis-regulation (Jin et al, Elife, 2015). The new GO-term analysis has been added the figures (see new Fig. 2b) and described in the second paragraph of the sub-section “Repli-ID identifies known regulators of replication fork stability/progression” in the Results section in the manuscript.

The GOterm analysis of the RNA-seq and suppressor screen have also been repeated using the PANTHER GO-Slim analysis. For RNA-seq, we found homeostatic process and response to oxidative stress to be significantly enriched. For the suppressor screen no significantly enriched GO-terms were found. Also this new GO-term analysis has been included in new Supplementary figure S6a, and described in first and second paragraph of the sub-section “Rox1 controls the response to replication stress by repressing Sur2” in the Results section in the manuscript.

They validated the accuracy of the screen by re-testing 12 knockout strains by Pol e ChIP and most (10) were reproducible (i.e., reduced Pol e association), supporting the ability of the

screen to detect reduce Pol e in HU. It is not clear how these 12 were selected for further testing, which is important for validation and inferring the overall usefulness of the screen. It is implied that these have been linked in some way to replication/genome stability but the connection is vague and possibly indirect. If this is true, the remaining ~400 genes have no previous link to replication, and while these might represent bona fide new regulators, no further analysis of those is performed. Is this correct? Why?

We randomly selected the 12 mutants from hits of the Repli-ID screen that showed reduced Pol2 levels and that have not previously been linked to DNA replication. Using these criteria, we expected to identify new regulators of DNA replication. We now mention this accordingly in the first paragraph of the sub-section “Repli-ID identifies novel factors that affect replication fork stability/progression” of the Results section in our manuscript. Indeed, our further in-depth follow-up analysis revealed Lge1 and Rox1 as new players in the response to replication stress. A similar analysis of the remaining >400 genes is beyond the scope of this manuscript, if feasible at all.

Moreover, reconstruction of six of the 12 knockout strains in a different strain background only validated three (LGE1, ROX1, and NSR1), suggesting significant strain-dependent effects or questionable strain identities in the knockout collection. Because *lge1*Δ and *rox1*Δ were HU-sensitive, they focused further analysis on these two.

Lge1 is a known co-factor of Bre1 that together are required for ubiquitylation of H2B, which is reported to play a role in establishment of cohesion during DNA replication (Zhang et al 2017 not cited) and premeiotic DNA synthesis (Jordan et al 2007 not cited). Whereas H2Bub is known to be required for recovery from replication stress, Lge1 has not been directly implicated previously, but would seemingly be strongly implicated by association.

The first paragraph of the sub-section “Lge1 affects DNA replication under stress conditions by promoting H2B mono-ubiquitylation” of the Results section of the manuscript has been adapted accordingly, also citing these two publications.

Analysis of *lge1*Δ cells by ChIP of Pol e shows a striking decrease in Pol e loading onto ARS607, suggesting an initiation defect. There is no analysis without HU so there’s no distinction here between an initiation defect versus fork stability defect due to HU. I think analysis without HU is required. Also, they should demonstrate that reduced replication is not due to a defect in licensing by analyzing MCM levels.

We performed Pol2 ChIP-qPCR experiments at 20 degrees, which slows down S-phase progression in the absence of HU, allowing us to capture Pol2 (which would otherwise progress too quickly during the course of the experiment and in the absence of HU). We found that Pol2 levels were dramatically reduced in *lge1*Δ cells after 25 minutes in S-phase without HU (see new results new Fig. S3b), which was similar to the effect observed in the presence of HU (see Fig. 3b). We also performed Mcm4 ChIP-qPCR experiments at 30 degrees in the presence of HU. In contrast to Pol2 (Fig. 3b), Mcm4 levels were not dramatically altered in *lge1*Δ, albeit perhaps somewhat reduced at the later timepoints (15 and 20 minutes) when compared to that in wild-type, suggesting that replication initiation/licensing in this mutant is not affected (see new results in Fig. 3c). We have mentioned this in the first paragraph of the sub-section “Lge1 affects DNA replication under stress conditions by promoting H2B mono-ubiquitylation” of the Results section of the manuscript.

In WT cells H2Bub is present on chromatin prior to replication (and generally higher at origin distal regions) and changes little during S-phase in HU. In *lge1Δ*, levels of H2Bub are lower across the region, throughout the time course; similar in *bre1Δ* and *bre1Δ lge1Δ*. While deletion of LGE1 reduces H2Bub and reduces Pol e loading or stability, there is no functional connection between these effects. A simple interpretation is that H2Bub present at origin is required for efficient initiation, but this is not demonstrated. Elimination of Lge1 and/or Bre1 function through a degron approach may enable more rigorous test of Lge1-Bre1 requirement in replication. Analysis of bulk replication by flow cytometry or other approaches is sorely lacking.

The new Mcm4 ChIP experiments (see previous point and new results in Fig. 3c) did not show a replication initiation/licensing defect in *lge1Δ*. Moreover, S-phase entry, as monitored by budding index analysis, was normal (see Supplementary Fig. S3c). This suggest that the reduced Pol2 levels observed in *lge1Δ* and in *bre1Δ* reflect Pol2 stability/progression rather than an initiation problem. This is supported by new copy number analysis, showing that, although DNA synthesis is initiated, it is reduced in *lge1Δ* (see new results in Fig. 3d). We elaborate on the implications on these new findings on the link between H2Bub and Pol2 loading in the Discussion.

Though there is no obvious change in H2Bub levels linked to replication across the region during the time-course. Bre1 association with the origin region appears to increase (~2-fold) during replication, possibly even anticipating the increased Pol e association that occurs during the time-course, which seems inconsistent with its recruitment to stalled forks. This increase in Bre1 signal is partially dependent on LGE1. If this increased association is functionally significant, it remains unclear whether this is a normal or replication stress response.

We performed ChIP-qPCR experiments at 20 degrees, which slows down S-phase progression, in the absence of HU. While we found increased levels of Bre1 near ARS607 under HU conditions (see Fig. 4a), we were unable to detect Bre1 at this locus in unperturbed conditions (see new result in Supplementary Fig. S4b). This was not due to an issue with our ChIP-qPCR approach as we also included a positive control, the *PMA1* locus, previously used by Song and Ahn (J. Biol. Chem., 2010), at which we observed up to a 7-fold enrichment of Bre1 over the background (see new result in Supplementary Fig. S4b). Collectively, these findings suggest that the Bre1 recruitment is a replication stress-induced phenomenon.

They show reduced Rad53 phosphorylation/activation. Given the possible evidence of reduced replication initiation, the reduced Rad53 activation may simply reflect lower initiation levels, rather than a direct role in replication stress signaling. Viability assays show sensitivity to HU in the *lge1Δ*, *bre1Δ* and *bre1Δ lge1Δ* cells; mechanistic significance is unclear.

The new Mcm4 ChIP-qPCR experiments (see above and see new results in Fig. 3c) did not show a replication initiation/licensing defect in *lge1Δ*. Moreover, S-phase entry, as monitored by budding index analysis, was normal. In contrast, reduced Pol2 levels were observed in *lge1Δ/bre1Δ*, suggesting a Pol2 stability/progression rather than an initiation problem. This is supported by new copy number analysis, showing that, although DNA synthesis is initiated, it is reduced in *lge1Δ* (see new results in Supplementary Fig. S3d). Taken together, these findings suggest that the reduced Rad53 phosphorylation/activation in *lge1Δ* does not reflect

lower replication initiation levels, but rather reduced signalling of unstable/arrested replication forks, causing reduced fork recovery, as indicated by the viability assay (see Fig. 4c).

Overall, the identification of LGE1 is seen as validation of the screen given prior evidence of its role in replication as well as the well-established role of Bre1 in replication fork stability; however, the additional analysis of *lge1*Δ does not provide compelling new evidence of a role in fork stabilization per se. The conclusions are not inconsistent with the data nor are they rigorously supported by the data.

Our data obtained in *lge1*Δ mutants are indeed consistent with previously published data in *bre1*Δ and *htb-K123R* mutants (Trujillo et al., Mol. Cell, 2012). This not only contributes to the validation of the Repli-ID screen, but also reveals that Lge1 is an important regulator of the Bre1/Rad6 complex during the response to derailed DNA replication. This role may be distinct from that at transcription sites where Bre1 recruitment occurs in a manner independent of Lge1 (Kim et al., FEBS Lett., 2018).

They move on to analysis of ROX1 and confirm that deletion of ROX1 reduces Pol e association with origin and flanking chromatin. Because ROX1 regulates RNR gene expression, dNTP levels were measured and determined to be unchanged (significance test?), suggesting a different mechanism.

A two-tailed unpaired Student's t test assuming unequal variances has been done to test for significance (see figure legend of Fig. S5), but none of the dNTPs changed in a statistically significant manner in *rox1*Δ when compared to that in wild-type. We do not indicate non-significant results in the figures, as this will reduce the clarity of the figures. However, to clarify the lack of a statistically significant difference, we now mention in the second paragraph of the sub-section "Rox1 does not impact DNA replication by affecting cellular dNTP levels" in the Results section that "dNTP levels did not change significantly in the absence of Rox1".

They analyzed RNA levels in *rox1*Δ cells to identify genes whose misexpression might explain the replication phenotype; however, no enrichment for genes annotated as DNA replication or repair were identified. They sought suppressors of the HU-sensitivity resulting from deletion of ROX1 by crossing *rox1*Δ with the gene knockout collection of strains. They identify 431 genes that suppressed *rox1*Δ, which seems like a huge number (the writing is confusing: "431 gene deletions were identified that suppressed growth on HU in *rox1*Δ (Fold change > 1.5)" I think they mean suppressed the lack of growth in *rox1*Δ.). They overlap these 431 with genes identified as upregulated by the RNA-seq analysis and find (only?) three genes: SUR2, TRX2, and DOG1. Only SUR2 deletion suppresses *rox1*Δ in W303 background, so focus moves to SUR2. How can there be so many suppressors of *rox1*Δ HU-sensitivity?

"I think they mean suppressed the lack of growth in *rox1*Δ." The reviewer is right and we have changed this accordingly in the text.

The suppressor screen was done once and may contain false-positive hits (e.g. due to contaminations or spontaneous suppressor). Repeating this screen and performing additional statistical analysis will likely help to get rid of some false-positive hits, thereby reducing the number of suppressors of the *rox1*Δ HU-sensitivity. However, this is beyond the scope of the current manuscript.

SUR2 regulates ceramide production, which in turn can regulate PP2 phosphatase activity, which is a cell cycle regulator. They show that presence of ceramide can prevent suppression of *rox1Δ* by SUR2 deletion, supporting the idea that increased ceramide levels in *rox1Δ* cells produced by upregulated SUR2 lead to the HU-sensitivity and hence, is suppressed by SUR2 deletion. Given the delay in budding (reflecting the G1-S transition) in *rox1Δ* cells, it appears that Rox1 exerts its effect on replication indirectly by modulating PP2 activity. This may act through G1-S control or intra-S control. Both mechanisms may explain the reduced Pol e loading in *rox1Δ* cells, though this is not further examined. Do any other genes identified in the RNA-seq or suppressor screen (e.g.: in the GO categories Response to chemical and lipid metabolic process) potentially further support the involvement of the ceramide pathway described here?

Factors involved in the sphingolipid and ceramide pathway identified in the suppressor screen performed in *rox1Δ* were *SUR4/ELO3*, *ISC1*, *SIT4*, *SLC1* and *TSC3*. However, none of these genes is upregulated in the RNA-seq experiment screen performed in *rox1Δ*, indicating they are not transcriptionally regulated by Rox1. Conversely, in the RNA-seq experiment in *rox1Δ*, *LAC1* and *RTS3* were upregulated, but loss of these genes did not suppress the HU phenotype of *rox1Δ* in the suppressor screen.

Overall, the impacts of Repli-ID here are modest. Out of hundreds of candidates showing decreased Pol e association, only a few appeared worthy of follow up. Lge1 was already implicated through Bre1 so appears more confirmatory than novel. Rox1 involvement is more novel, but the real insight here came from the suppressor screen based on HU sensitivity. Perhaps the identification of other candidate genes by this screen will provide motivation to further examine and possibly reveal some new insights as with Rox1, but it's not clear that there will be many more strong candidates in their list, which as far as I can tell is not provided, but should be the most valuable part of this study.

All results from the Repli-ID and suppressor screens were provided as source data, but the results from the Repli-ID screen have now also been included in a new Supplementary Data 1 file. The suppressor screen was performed in *rox1Δ*, yielding insights into Rox1-related pathways in the cellular response to HU-induced replication fork stalling. We therefore believe that the data from this screen, while being important for unravelling Rox1 function, will be of limited value with regards to its use in combination with the Repli-ID screen. However, the Repli-ID screen alone already allowed us to identify 423 mutants with altered Pol2 levels at ARS404 (Fig. 2a). The impact of several of these mutants on Pol2 could be validated not only at ARS404, but also at ARS607 (Fig. 2c). We therefore expect that further validation and follow-up studies of such hits (by us or other laboratories in the future) will provide additional new mechanistic insights in the response to replication fork perturbation. These studies may also include mutants in which Pol2 levels are increased at stalled replication forks, which is a phenomenon that, to our knowledge, remains poorly understood.

Significant revision is required to support the potential impact for the field and other researchers. Conclusions from the data need to consider alternative explanations and caveats.

REVIEWER COMMENTS

We thank the reviewers for their constructive comments and useful suggestions. Based on these we have adapted the manuscript textually, performed additional experiments, and included new figure panels. These changes have further solidified the conclusions. We have addressed all comments point-by-point below. Changes in the revised manuscript are in marked in red text.

Reviewer #1 (Remarks to the Author):

In this study, the authors describe a new strategy (termed Repli-ID) designed to identify regulators of replication stress responses genome-wide in budding yeast. They go on to characterize two hits, LGE1 and ROX1. Repli-ID is based on Epi-ID and involves crossing a strain library of knockouts to a strain containing a cassette with ARS404 plus the KanMX gene flanked by two barcodes (one immediately adjacent to origin, one 1.6 kb from origin as cassette is integrated into the HO locus), an epitope tagged Pol epsilon plus *sld3-38A dbf4-4A* to enable ARS404 to "escape" intra-S phase checkpoint regulation so it can fire prior to a hydroxyurea arrest instead of later in the cell cycle. Investigators perform ChIPseq for an epitope tagged Pol epsilon in the presence of DNA replication stress mediated by hydroxyurea after release from a G1 arrest. Overall, Repli-ID appears to be a powerful strategy to identify genes that influence DNA replication either directly or indirectly, the data is of high quality and analyzed appropriately. The study will be of high interest to others in the field, many of whom will be interested in adopting various versions of this approach.

In this revision, the authors have greatly clarified how experiments were conducted, included several followup control/validation experiments, and expanded discussion and commentary on and acknowledgement of potential alternate interpretations to the data they present. This will greatly facilitate readers' understanding of the study and how the reported findings relate to published studies.

Enough detail is now provided to confirm methodology is sound and enough detail has been provided in the methods for the work to be reproduced.

We thank the reviewer for the positive evaluation of our revised manuscript.

Reviewer #2 (Remarks to the Author):

I acknowledge that the authors have addressed all the points raised and have made significant improvements to the manuscript. Nevertheless, some points are incompletely addressed or require further explanation.

Point 14. Reviewers #2, #3 and #4 pointed out that H2Bub levels in the vicinity of ARS607 in wild-type cells do not seem to change before and after replication stress (Figure S3G). This is not clearly stated in the Results section, where the authors only mention that H2Bub levels

were equally decreased in G1 and S+HU in *lge1* Δ and *bre1* Δ mutants compared to wild-type cells.

To better indicate the lack of change in H2Bub levels in wild-type cells between G1 and S+HU, we have added the following sentence to the 4th paragraph of the subsection “Lge1 affects DNA replication under stress conditions by promoting H2B mono-ubiquitylation” of the Results: “In wild-type cells, high levels of H2Bub were detected in G1-phase (t₀), which remained largely unchanged at 20, 40 and 60 minutes after release in HU (Supplementary Fig. S3g)”.

These results do not suggest that H2Bub is part of the replication stress response that stabilizes Pol epsilon. However, the authors claim in their discussion (page 16) that their mechanistic studies show that Lge1 directly drives H2Bub at stalled forks and enhances Bre1 activity.

We have toned down this conclusion:

- in the Results section by saying that the lack of H2Bub does not contribute to, but associates with loss of Pol ϵ under HU conditions (see 4th paragraph of the subsection “Lge1 affects DNA replication under stress conditions by promoting H2B mono-ubiquitylation”).

- in the Results section by saying that Lge1 sustains Bre1-dependent H2B mono-ubiquitylation near origins of replication rather than at stalled forks, both by stabilizing Bre1 and enhancing its activity (see last paragraph of the subsection “Lge1 controls the intra-S-checkpoint and fork recovery via H2B ubiquitylation”).

- in the Discussion by saying “Lge1 drives Bre1-dependent H2B mono-ubiquitylation at Lysine 123, which may promote replication fork stability and recovery” rather than “Lge1 drives Bre1-dependent H2B mono-ubiquitylation at Lysine 123 directly at stalled replication forks to promote their recovery” (see 6th paragraph).

This conclusion is not justified by the data because there is no difference in H2Bub levels between G1 and S+HU. An alternative interpretation is needed. For example, the authors do not mention in the discussion the delayed checkpoint activation in *lge1* Δ and *bre1* Δ mutants, which could affect the stability of Pol epsilon (through loss of chromatin remodeling?).

We already stated in our revised manuscript that Lge1 may promote fork stability by promoting H2Bub-dependent checkpoint activation, agreeing with the reviewer and a previous report (Lin et al., *PLoS Genet.*, 2014) (last sentence of the last paragraph of the subsection “Lge1 controls the intra-S-checkpoint and fork recovery via H2B ubiquitylation” of the Results). However, to better clarify this point, we have changed the phrasing of this sentence by saying “Lge1 sustains Bre1-dependent H2B mono-ubiquitylation near origins of replication both by stabilizing Bre1 and enhancing its activity, thereby facilitating an efficient intra-S-checkpoint response crucial for fork stability and recovery following replication stress”. Importantly, we have also accommodated this point by adjusting the model, which now indicates that fork stability may be affected by H2Bub-dependent checkpoint activation (see new Fig. 4d).

Point 16. Given the potential importance of the checkpoint defect in *lge1* Δ and *bre1* Δ mutants, these data need to be consolidated.

The authors have stated that they cropped the original blot, justifying the presence of dotted lines in Figure 4B. This needs to be indicated in the figure legend.

We have now indicated that the dotted line indicates that the blot is a cropped blot (see new legend Fig. 4b).

In addition, the authors provided the reviewer with the replicate of this experiment (Reviewer_only_Figure 4). This blot is inconclusive (2 bands at time 0, no clear increase in phospho-mobility shift observed) and needs to be repeated.

We have repeated the experiment to examine the Rad53 phospho-mobility shift in the absence of Lge1 but not Bre1, as the role of Bre1 in Rad53 activation has been previously reported (Giannattasio et al., J. Biol. Chem., 2005; Lin et al., PLoS Genet., 2014) and corroborated in our current manuscript (see Figure 4b). The new experiment reveals a reduced Rad53 phospho-mobility shift in absence of Lge1, particularly 30 and 60 minutes after the release from G1 into S-phase + HU (see new Supplementary Fig. S4d), consistent with the representative blot we already showed in Figure 4b of the revised manuscript, and supporting the conclusion that Rad53 activation is delayed in absence Lge1.

Point 15. The Bre1 ChIP efficiency near ARS607 in response to replication stress is still an issue. The authors have included a positive control (PMA1 gene) in Figure S4B, but this control was not included in response to HU, where a very weak recruitment of Bre1 in wild-type cells and the partial loss in lge1 Δ were observed, but questionable (Figure 4A).

This control was already included in our analysis of the response to HU, as described in the text (see subsection “Bre1 recruitment to stalled replication forks is partially Lge1-dependent” of the Results) and shown in Supplementary Fig. S4a. However, for clarity, we have revised the text to better highlight this control, which demonstrates that the loss of Lge1 affects Bre1’s association with the *PMA1* gene both in the absence and presence of HU.

Point 19. As the authors understood from Ferrari et al. (Mol Cell 2017), Irc21 promotes the production of ceramides and consequently the activation of PP2A.

If the defect of rox1 Δ in HU is due to the overexpression of SUR2 and the overproduction of ceramides, as proposed by the authors, this defect should be suppressed by the loss of IRC21.

This is not what the authors observed in their new results in the rox1 Δ irc21 Δ double mutant (Reviewer_only_Figure 5).

Ferrari et al. (Mol. Cell, 2017) demonstrated that deletion of *IRC21* could rescue the HU sensitivity of checkpoint mutants, but in their experiments a lower HU concentration of 3 mM was used. In contrast, in our experiments 100 mM HU was used, a condition under which the *irc21 Δ mutant itself exhibited hypersensitivity. This makes it difficult, if not impossible, to detect a rescue of the HU sensitivity of the *rox1 Δ mutant by the loss of *IRC21*.**

Furthermore, it remains unclear whether Irc21 and Sur2 operate within the same pathway regulating ceramide production. If they function independently, the excessive ceramide production driven by Sur2 overexpression (in the absence of Rox1) could obscure the effect of IRC21 deletion on ceramide levels. Further research is required to clarify how Irc21 and Sur2-dependent ceramide production influence the cellular response to HU. This point has now been included in the penultimate paragraph of the Discussion.

In addition, the same figure shows that despite the overall increase in HU sensitivity, sur2 Δ still suppresses rox1 Δ HU sensitivity in the presence of cell-permeable C2 ceramide.

This figure shows only a very slight suppression, which is much less than observed in the absence of ceramide. In a *sur2Δ* mutant, ceramide production is still reduced. Therefore, it is possible that the concentration of ceramide is insufficient to fully counteract the effect of Sur2 deletion in the spot dilution assay shown in this figure.

Finally, the results shown in Figure S6E hardly show that the addition of ceramides decreases Rad53 phosphorylation in wild-type cells (poor resolution of the phospho-mobility shift and more protein loading in the +HU-only condition) and do not confirm the results shown in Figure 6B. The authors did not describe or interpret the effect of ceramide addition in *rox1Δ* cells. It appears that ceramide addition is still able to decrease Rad53 phosphorylation in *rox1Δ* cells.

Therefore, we believe that the authors need to find an alternative interpretation of their data.

The addition of ceramide to *rox1Δ* cells slightly reduces Rad53 phosphorylation levels (see Fig. 6b and Supplementary Fig. S6e). Similarly, in *rox1Δ* cells, ceramide also has a modest effect on the HU sensitivity (Fig. 6a), the budding index (Fig. 6d), and Pol2 abundance at the replication fork (Fig. 6e). We hypothesize that in the absence of Rox1, ceramide levels are elevated due to high Sur2 expression, leading to issues with the G1/S phase transition and defective checkpoint activation. However, adding cell-permeable C2 ceramide further increases the ceramide concentration, which may exacerbate both the G1/S phase transition and checkpoint defects. We have now included this explanation in the first and second paragraph of the subsection "Rox1 affects DNA replication by regulating checkpoint activation and S-phase entry through Sur2/ceramide control" of the Results.

Point 9. The authors have provided Pol2 ChIP data in the *sgs1Δ* mutant in the *sld3 dfb4* background (Reviewer_only_Figure 3) and show that Pol2 levels are globally decreased near ARS404 and ARS607.

Since *sgs1Δ* has already been described to decrease Pol2 levels in HU, these new data validate their approach and should be included in the manuscript.

The Pol2 ChIP-qPCR data obtained with the *sgs1Δ* mutant have now been included in the manuscript (see 3rd and 4th paragraph of the Results section and new Supplementary Fig. S1e).

Reviewer #3 (Remarks to the Author):

Reviewer #4 (Remarks to the Author):

The authors have done a good job at responding to my critiques with new experiments and analysis. But I disagree strongly with their interpretation regarding the effect on initiation versus fork progression. I think they might be more circumspect about this point.

They state (twice in the rebuttal): “The new Mcm4 ChIP experiments (see previous point and new results in Fig. 3c) did not show a replication initiation/licensing defect in *lge1Δ*. Moreover, S-phase entry, as monitored by budding index analysis, was normal (see Supplementary Fig. S3c).” They also state: “This is supported by new copy number analysis, showing that, although DNA synthesis is initiated, it is reduced in *lge1Δ* (see new results in Supplementary Fig. S3d).”

Licensing and initiation are two different things. Budding index does not indicate S-phase entry only progression through START. I still think their data point equally strongly to an initiation defect rather than fork stability as they observe reduced Pol epsilon loading even without HU and a delay in any copy number increase at the origin. These data are completely consistent with an initiation defect. Whereas there’s progression of the MCMs in the WT, this doesn’t appear to happen in *lge1Δ*. Given the copy number data it would seem that *lge1Δ* cells are replicating very poorly. I think DNA content analysis is still sorely missing; presumably more analysis of the total DNA for the copy number analysis would address this. The Discussion in the text (p. 8-9) is similarly problematic, suggesting that proper licensing mean initiation is normal.

We have revised the manuscript to emphasize that our data suggest defects in both replication initiation and fork stability in *lge1Δ* cells and changed our interpretation of the budding index results (see abstract, first and second paragraph of the subsection “Lge1 affects DNA replication under stress conditions by promoting H2B mono-ubiquitylation” of the Results, and sixth paragraph of the Discussion).

We have also analyzed DNA content by propidium iodide (PI) staining and flow cytometry. This revealed that *lge1Δ* cells progress slower through S-phase, consistent with the replication initiation and fork stability defect. We have added this new result to the revised manuscript (see new Fig. 3d and Supplementary Fig. S3c, and the first paragraph of the subsection “Lge1 affects DNA replication under stress conditions by promoting H2B mono-ubiquitylation” of the Results).

It is indeed compelling that Bre1 is only seen to associate with the origin in the presence of HU but how do the authors explain that effects on H2BUB are already apparent at T=0 (e.g. Fig. S4b)?

We believe the reviewer is referring to the effect of Lge1 loss on H2Bub levels in G1-phase cells (t=0 min). If this interpretation is correct, we suggest that H2Bub levels in wild-type G1 cells are mediated by Bre1, the sole known E3 ligase for H2Bub. However, Bre1 might not be detected in ChIP-qPCR experiments, likely due to its low abundance or its transient and difficult-to-crosslink interaction with chromatin. Consequently, the loss of Lge1 reduces functional Bre1 levels, which remain undetectable by ChIP-qPCR in G1-phase cells.

I also think that all ChIP-seq datasets should show analysis of how well experimental replicates correlate (R2) genome-wide as supplementary information.

The correlation plots for the two replicates of our ChIP-seq dataset are shown in Supplementary Fig. S2a and S2b of the revised manuscript. Spearman correlation coefficients (R) with corresponding p-values, indicating a strong correlation between the Repli-ID replicates at t=40 (R=0.90) and t=80 minutes (R=0.91), are mentioned in the figure and the first paragraph of the Results subsection, "Repli-ID identifies known regulators of replication fork stability/progression."